# CellNavi predicts genes directing cellular transitions by learning a gene graph-enhanced cell state manifold

Tianze Wang[1,2,6], Yan Pan[1,3,6], Fusong Ju[1,6], Shuxin Zheng[1,5,6], Chang Liu[1,6], Yaosen Min[1,5], Qun Jiang[1,3], Xinwei Liu[1,4], Huanhuan Xia[1,5], Guoqing Liu[1], Haiguang Liu[1] & Pan Deng [1]✉

A select few genes act as pivotal drivers in the process of cell state transitions. However, finding key genes involved in different transitions is challenging. Here, to address this problem, we present CellNavi, a deep learning-based framework designed to predict genes that drive cell state transitions. CellNavi builds a driver gene predictor upon a cell state manifold, which captures the intrinsic features of cells by learning from large-scale, high-dimensional transcriptomics data and integrating gene graphs with directional connections. Our analysis shows that CellNavi can accurately predict driver genes for transitions induced by genetic, chemical and cytokine perturbations across diverse cell types, conditions and studies. By leveraging a biologically meaningful cell state manifold, it is proficient in tasks involving critical transitions such as cellular differentiation, disease progression and drug response. CellNavi represents a substantial advancement in driver gene prediction and cell state manipulation, opening new avenues in disease biology and therapeutic discovery.

Understanding the genetic drivers of cellular transitions is crucial for elucidating complex biological processes and disease mechanisms[1–3]. However, identifying these drivers remains inherently challenging due to the vast number of genes involved in transitions and their complex interdependencies, contrasted with limited experimental capacity and incomplete biological knowledge. Therefore, in silico methods capable of predicting driver genes across diverse contexts are highly desirable.

Traditionally, efforts to pinpoint critical driver genes have primarily relied on network-based methodologies, with a particular focus on gene regulatory networks (GRNs)[4–8]. Although GRN-centric approaches have made notable progress, they also encounter limitations that hinder their broader use. For example, deducing accurate GRNs within heterogeneous cell populations, which is more relevant to translational research, remains a challenge[9,10]. Moreover, GRN models tend to prioritize transcription factors and may overlook non-transcriptional drivers of cellular transitions. This limits our understanding of complex cellular processes such as disease progression, immune modulation and pharmacological responses.

To this end, we developed CellNavi, a deep learning framework designed to predict driver genes and navigate cellular transitions. CellNavi constructs a driver gene predictor (DGP) on top of a learned manifold that parameterizes valid cell states. This manifold is modelled by mapping raw cell state representations onto a lower-dimensional coordinate space, where the dimensions correspond to intrinsic features of cell states, and the distance reflects the biological similarity between cells. To build this manifold, CellNavi is trained on large-scale, high-dimensional single-cell transcriptomic data, along with prior directional gene graphs that reveal the underlying structure of cell

[1]Microsoft Research AI for Science, Beijing, China. [2]Paul G. Allen School of Computer Science and Engineering, University of Washington, Seattle, WA, USA. [3]Department of Automation, Tsinghua University, Beijing, China. [4]Department of Physics, Beijing Normal University, Beijing, China. [5]Present address: Zhongguancun Institute of Artificial Intelligence and Beijing Zhongguancun Academy, Beijing, China. [6]These authors contributed equally: Tianze Wang, Yan Pan, Fusong Ju, Shuxin Zheng, Chang Liu. ✉e-mail: dengpan@bjzgca.edu.cn

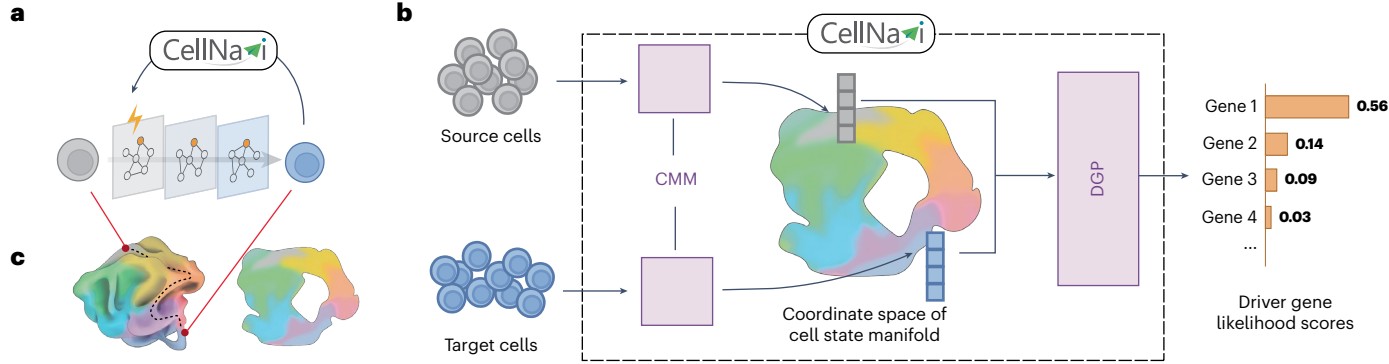

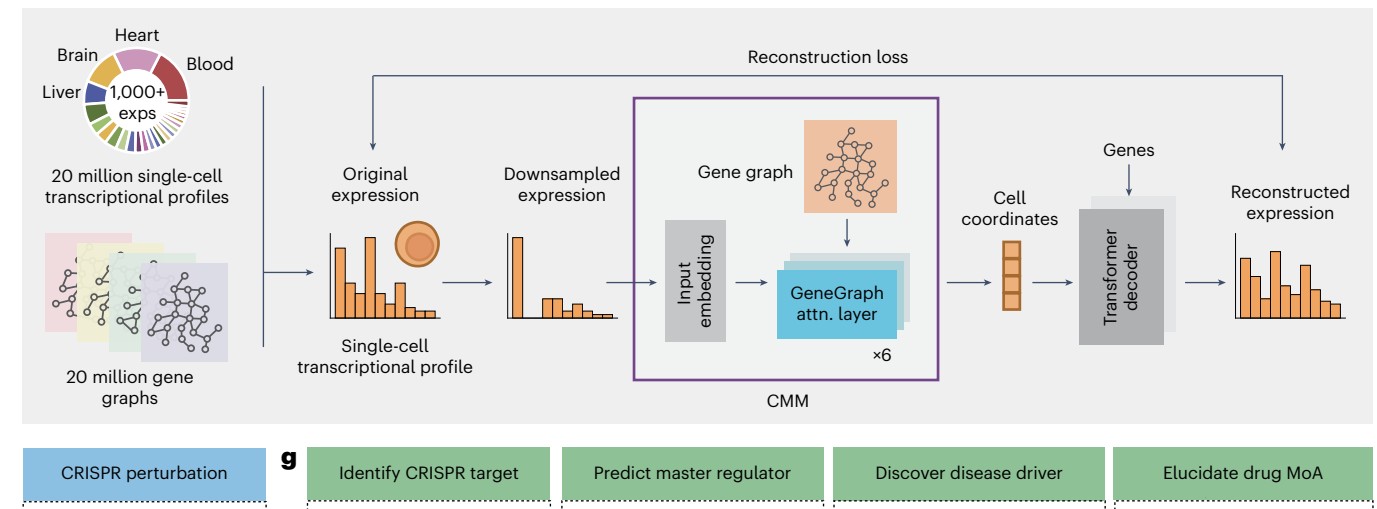

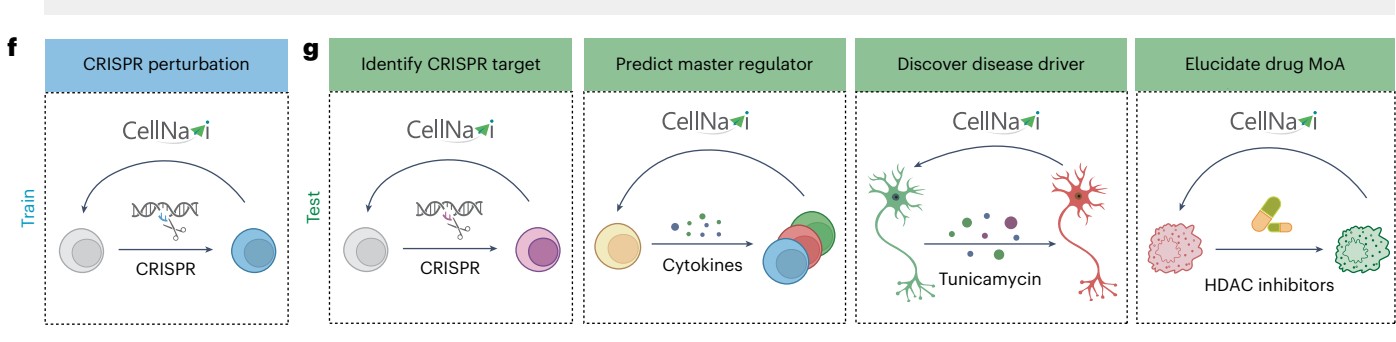

**Fig. 1 | Overview of CellNavi. a**, A conceptual illustration of CellNavi's task. Given a pair of source and target cells undergoing a transition induced by stimuli, CellNavi predicts the driver gene responsible for this transition. **b**, The workflow of CellNavi. The CMM maps the source and target cells onto a coordinate space of the cell manifold. The DGP then uses the cell coordinates produced by the CMM to rank the candidate genes by likelihood scores. **c**, An illustration of the cell manifold and its coordinate space. **d**, Data used for the CMM training. exps,

experiments. **e**, Training of the CMM. The CMM consists of six GeneGraph Attention (attn.) layers designed to incorporate graph-based information. During training, single-cell transcriptomic profiles are randomly sampled from the curated HCA dataset and used as input. Cell embeddings generated by the model are then used by a transformer decoder to reconstruct gene expression profiles. **f**, Data used for the DGP training. **g**, Application scenarios and test cases of CellNavi. MoA, mechanism of action. Schematic elements created with BioRender.com.

states. By projecting cellular data onto this biologically meaningful space with reduced dimensionality and enhanced biological relevance, CellNavi provides a universal framework that generalizes across diverse cellular contexts, allowing robust driver gene predictions even in previously unexplored cell types or conditions.

Our results show that CellNavi excels at predicting driver genes across a wide range of biological transitions, demonstrating strong performance in quantitative tasks curated in both immortalized cell lines and primary cells. It identifies crucial regulators in T cell differentiation and uncovers key genes associated with neurodegenerative diseases. Notably, CellNavi infers mechanisms of action for drug compounds without the need for drug-specific training, underscoring its potential in drug discovery. In summary, CellNavi offers a powerful framework for deciphering cell state transitions and their underlying

mechanisms, holding profound promise for advancing cell biology and disease research.

## Overview of CellNavi

CellNavi is designed to predict driver genes for given cellular transitions, where the transcriptomic data of the source and target cells represent the initial and final states of these transitions (Fig. 1a–c).

CellNavi comprises two main components: the cell manifold model (CMM), which captures and represents cell states, and the DGP, which identifies key genes driving these transitions based on learned cell representations (Fig. 1b).

The CMM is built to capture valid cell states across diverse biological contexts. While transcriptomes are often used to represent cell states, valid cell states do not span the entire high-dimensional transcriptomic

space but instead form a lower-dimensional manifold (Fig. 1c). To model this, the CMM maps transcriptomic vectors to a lower-dimensional coordinate space that represents the intrinsic features of cell states, while preserving the relative similarities between cells (dimensionality considerations are discussed in Supplementary Note 1).

We first curated a dataset of approximately 20 million single-cell transcriptomic profiles sourced from the Human Cell Atlas (HCA)[11] (Fig. 1d) and adapted a transformer architecture based on attention mechanisms, known for its ability to discern complex patterns in large-scale data[12–17], to train the CMM (Fig. 1e). The training involved a self-supervised downsampling reconstruction task (Methods and Supplementary Note 2). To prioritize cell rather than gene-level representations, we developed a decoder module to reconstruct gene expression profiles from the cell coordinates—representations of cells within the coordinate space of the cell state manifold—generated by the CMM (Fig. 1e and Extended Data Fig. 1; Methods). This approach aligns cells across varying sequencing depths (Extended Data Fig. 2a) and recapitulates developmental trajectories from single cells (Extended Data Fig. 2b–d), indicating that it captures both intra- and intercellular features.

However, relying solely on transcriptomic data may overlook the intricate gene–gene interactions that are crucial for describing and distinguishing cell states. To address this, we incorporated 20 million cell-specific gene graphs into the CMM training process (Fig. 1d,e). These graphs encode directional connections derived from a prior network that spans over 30,000 human genes and their associated signalling pathways[18] (Methods). More specifically, in these gene graphs, each edge represents a causal relationship between two genes, with the direction indicating the regulatory influence from one gene to the other (Methods). These graphs provide richer information about the complex dependencies among genes, which extend beyond simple transcriptomic data, hence better implying intrinsic variables spanning the valid cell space. To leverage these gene graphs, we replaced the standard transformer encoder layer in the CMM using a GeneGraph attention layer (Extended Data Fig. 1b). These layers, inspired by attention variants tailored for graph data[19], can process gene networks, thus enabling the model to integrate critical gene–gene relationships. With these designs, the model is driven to cultivate a manifold that systematically represents cell states and effectively reflects the relationships between cells, forming an informative foundation for driver gene prediction.

Building upon this manifold, we developed the DGP to predict genes driving specified cellular transitions (Methods and Extended Data Fig. 3a). The DGP is trained on clustered regularly interspaced short palindromic repeats (CRISPR) screen data, which link genetic perturbations to consequent changes in cell states[20–25]. We designated unperturbed controls and CRISPR-perturbed cells as source and target pairs, respectively, and utilized validated perturbed genes as labels for joint training (fine-tuning) of the CMM and DGP (Fig. 1f). Specifically, for each cell pair, their transcriptomic profiles are transformed into cell coordinates by the CMM, which are then processed by the DGP to generate a likelihood score vector indicating the probability that various candidate genes are orchestrating the transitions (Extended Data Fig. 3a).

We demonstrate that CellNavi, fine-tuned on CRISPR screen data—typically conducted on cultured cells or homogeneous populations and focusing on immediate genetic perturbations—can be extended to more complex transitions in heterogeneous tissues and primary cells (Fig. 1g and Extended Data Fig. 3b). By leveraging a biologically meaningful manifold, CellNavi generalizes knowledge gained from CRISPR screens beyond their original scope, to cellular transitions that are challenging to investigate using regular CRISPR methodologies. However, we acknowledge that CellNavi's performance in specific contexts may benefit from additional fine-tuning on relevant CRISPR datasets. Incorporating expanded experimental data may further enhance its applicability across diverse biological settings with minimal adaptation.

## Quantitative evaluation of CellNavi

To assess the capabilities of CellNavi, we first evaluated its performance on CRISPR perturbation datasets, where driver gene information is well established for transitions from source (unperturbed) to target (perturbed) cells.

We initially applied CellNavi to the Schmidt dataset, a CRISPR activation screen profiling 69 genetic perturbations[26]. This dataset captures distinct expression profiles and molecular phenotypes across both resting and restimulated T cells, within and between different cell types, before and after perturbations (Extended Data Fig. 4a). We fine-tuned our model on restimulated T cells and tested it on resting T cells (Fig. 2a). This set-up allowed us to evaluate CellNavi's ability to generalize across heterogeneous primary cells and predict driver genes in new cell states.

For each source–target cell pair, CellNavi prioritizes candidate genes based on their predicted likelihood scores. Across 23,047 source–target cell pairs, CellNavi achieves a top-1 accuracy of 0.621 and a top-5 accuracy of 0.733 (Fig. 2b), while maintaining strong performance across additional metrics (Fig. 2b,c and Extended Data Fig. 4b). Interestingly, substantial variation in top-1 accuracy was observed across perturbed genes, independent of sample size (Fig. 2d). Correlation analysis between gene-wise performance and the Local Inverse Simpson's Index (LISI)[27] suggests that CellNavi's accuracy is influenced by the degree of perturbation heterogeneity: perturbations with low average LISI values, indicative of a more distinct and homogeneous response, were associated with higher accuracy (top-1 accuracy >0.8, Fig. 2e).

To demonstrate CellNavi's effectiveness, we compared it with two alternative methods: SCENIC/SCENIC+[4,5], a training-free approach that infers GRNs from transcriptomic data with a focus on master regulators, and GEARS[28], an in silico perturbation approach, which targets a partially inverse problem of cellular transition prediction (Methods). Both SCENIC and GEARS exhibited markedly lower performance compared to CellNavi (Fig. 2b–d and Extended Data Fig. 4b). In addition, SCENIC, the network-based approaches, faced challenges in identifying regulons at the single-cell level (Extended Data Fig. 4c) and therefore struggled to make predictions in many cases. To investigate whether this is a broad challenge for GRN inference methods, we evaluated three alternative GRN inference approaches: GENIE3[29], GRNBoost2[30] and RENGE[31] (Methods). These methods similarly exhibited poor performance in single-cell contexts (Supplementary Table 1).

**Fig. 2 | Quantitative assessment of CellNavi. a**, A schematic of the quantitative evaluation framework. CRISPR-perturbed cells and their unperturbed controls are used for model training and evaluation, with data split by cell states to enable more rigorous testing. **b**, Top-1 accuracy, top-5 accuracy and F1 score for driver gene prediction in the Schmidt dataset, comparing CellNavi with alternative methods. The dashed line indicates the performance of a random guess. **c**, Area under the receiver operating characteristic curve (AUROC) scores for driver gene prediction in the Schmidt dataset, comparing CellNavi with alternative methods. **d**, Average top-1 accuracy for each gene. Left y axis: top-1 accuracy of different methods for each gene. Right y axis: the number of training (light blue) and test (steel blue) samples. **e**, Negative correlation between CellNavi's top-1 accuracy and the average LISI score across genes (Pearson correlation coefficient −0.451). A LISI score of 1 indicates indistinguishable perturbation effects, while a score of 0 suggests a distinct perturbation pattern. Dot colours represent the top-1 accuracy for individual genes. **f**, Top-1 accuracy, top-5 accuracy and F1 score for driver gene prediction in the Norman dataset (single perturbation), comparing CellNavi with alternative methods. The dashed line indicates the performance of a random guess. **g**, AUROC scores for driver gene prediction in the Norman dataset (single perturbation), comparing CellNavi with alternative methods. **h**, The distribution of predicted rankings for perturbed gene pairs. 'Perturbation 1' represents genes ranked higher, and 'Perturbation 2' represents genes ranked lower. n = 4,916. Source data for (**b,c,f,g**) are available in Supplementary Table 1.

CellNavi does not simply predict driver genes from expression changes. We conducted an ablation study by systematically removing the expression of perturbed genes from the input. Although this led to a decrease in performance, CellNavi still maintained substantial predictive accuracy, far surpassing expectations of random prediction (Extended Data Fig. 4d,e). In addition, DGE analysis revealed that the rankings of differentially expressed genes were poorly correlated with the actual perturbed genes (Fig. 2b–d and Extended Data Fig. 4b). These results suggest that CellNavi identifies driver genes beyond those detectable by expression shifts alone.

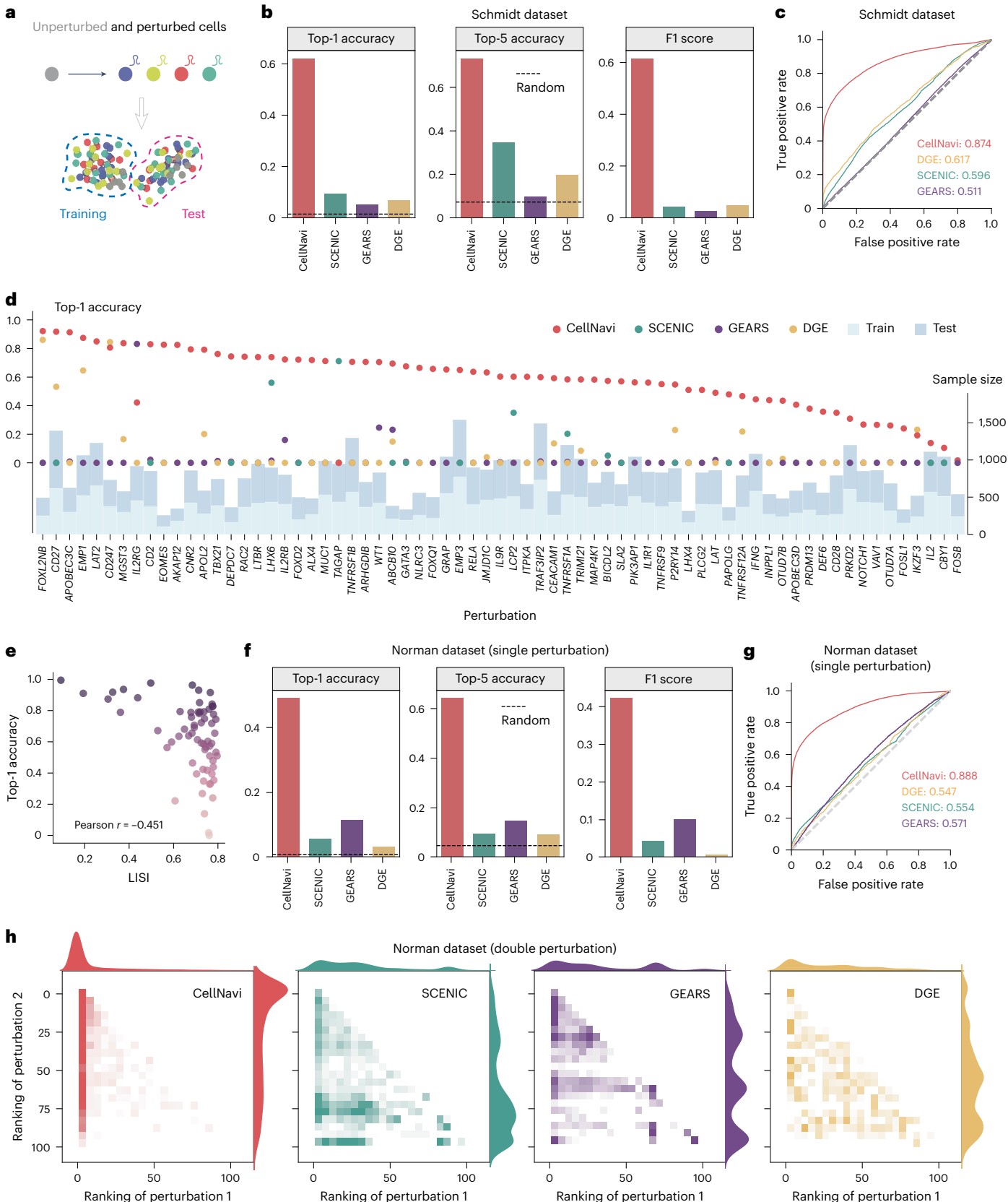

We further tested CellNavi on the Norman dataset[32], which features a CRISPR interference screen on the K562 cell line. This dataset encompasses 105 single-gene and 131 gene pair perturbations, allowing us to assess CellNavi's performance on transitions driven by both single and multiple genes. Using the unsupervised Leiden algorithm[33], we stratified the cells by cluster, holding out one cluster for testing and training on the remaining ones (Fig. 2a and Extended Data Fig. 4f). To ensure rigorous evaluation, we excluded all multigene perturbations from training.

CellNavi maintained strong performance on single driver gene prediction in the Norman dataset (Fig. 2f,g, Extended Data Fig. 4g and Supplementary Table 1). To evaluate multigene scenarios, we focused on the predicted rankings of perturbed genes. CellNavi ranked the first and second perturbed genes at averages of 7.9 and 31.2 out of 105 candidates, respectively, greatly outperforming all other tested methods (Fig. 2h).

Several recent studies have indicated that linear models can outperform deep learning methods in cell modelling tasks[34–37]. To investigate this, we evaluated multiple linear models for driver gene prediction under various conditions. Our results showed that CellNavi consistently outperformed these linear models by a substantial margin across settings (Supplementary Note 3). Furthermore, we applied cross-validation to ensure robust and unbiased evaluation and found that CellNavi demonstrated consistently superior performance across these conditions (Supplementary Tables 2 and 3). Altogether, these results, spanning diverse datasets and metrics, highlight CellNavi's strong capability to identify genes driving cellular changes, even in previously uncharacterized cell states.

## Evaluating model components and graph configurations

To assess the contributions of the CMM and DGP components, and to evaluate whether pretraining with the CMM improves generalization across biological contexts, we designed two ablated methods. The first combined the DGP with raw gene expression vectors instead of outputs from the CMM (no-CMM). The second replaced the DGP with a simpler multinomial logistic regression model (no-DGP). In addition to the Norman single perturbation split, which utilizes a cluster-based holdout strategy (out-of-domain split), we curated an alternative evaluation approach using random holdout to simulate a scenario without generalization (in-domain split). Removing either CMM pretraining (no-CMM) or DGP fine-tuning (no-DGP) led to reduced performance; however, for out-of-domain split, the absence of CMM pretraining (no-CMM) caused a greater drop in performance compared to the in-domain split scenario (Extended Data Fig. 5). These results highlight that CMM pretraining is essential for generalization across biologically diverse contexts, while DGP fine-tuning further optimizes task-specific predictions.

We also evaluated the impact of the NicheNet gene graph on CellNavi's predictions. Replacing NicheNet with GRNs inferred using GENIE3, GRNBoost2 or RENGE resulted in reduced performance (Extended Data Table 1), underscoring the advantage of integrating pathway-level information beyond GRNs, particularly in modelling perturbation-induced transitions. Furthermore, we tested graph configurations with varying levels of connectivity, including fully connected graphs, sparsified graphs with edges reduced to 1/10 or 1/20 of the original graph, and random graphs with the same sparsity as NicheNet (Methods). All alternative configurations led to further performance declines relative to biologically meaningful graphs constructed using diverse GRN inference methods (Extended Data Table 1). Collectively, these results emphasize the importance of leveraging biologically meaningful and comprehensive gene graphs, such as NicheNet, to ensure predictive robustness and accuracy.

## CellNavi identifies key genes in T cell differentiation

We next applied CellNavi to the Cano-Gomez dataset[38], which profiled T cell differentiation by stimulating naive and memory CD4+ T cells in vitro with anti-CD3/anti-CD28 and cytokines. During this process, external signals, such as antigens and cytokines, activate key genes modulating genetic circuits and gene expression programs, allowing T cells to adopt specialized functions. We assessed whether CellNavi could identify such key genes underlying transitions.

For this dataset, we constructed source–target cell pairs using Th0 cells as the source and cytokine-induced cells as targets. As cells differentiated into various effector T cell subtypes after stimulation[26,38–43], we first compiled a comprehensive marker gene set and computed a 'transition score' to quantify differentiation into these subtypes for each cell. Notably, marker genes associated with IL-2hi, IFNγhi and T helper 2 ($T_H2$) cells were strongly enriched (Extended Data Fig. 6), and transition scores towards these cell types demonstrated clear patterns (Fig. 3a–c and Methods). We then examined CellNavi's ability to identify driver genes across these effector T cell groups. Corresponding cell pairs were input into a CellNavi model trained on the Schmidt dataset, which encompasses extensive immune-related gene programs. Finally, we curated a literature-based list of established driver genes for phenotypic transitions towards specific effector cell types[26,42–49] (Supplementary Table 4) and evaluated CellNavi's performance in prioritizing these genes.

CellNavi accurately ranked *CD28* and *VAV1*, key drivers of IL-2hi cells, as the top candidates in the IL-2hi group defined by the transition score (Fig. 3d). Similarly, high rankings were observed for *CD27* and *IL9R* in IFNγ-high cells, and *GATA3* in $T_H2$ cells (Fig. 3d). We further analysed the average rankings of these established driver genes across the different effector cell groups. As expected, the relevant driver genes consistently ranked higher in their corresponding cell groups where they are known to drive differentiation. Notably, *CD28*, *VAV1*, *CD27* and *IL9R* achieved average rankings of 2.6, 2.9, 5.3 and 8.5, respectively, in their associated cell groups, greatly outperforming their rankings in unrelated groups (Fig. 3e). These results demonstrate CellNavi's effectiveness in identifying key genes that govern distinct differentiation pathways while distinguishing between cell fates. However, *CD28*'s dual role in IL-2 and IFNγ regulation was not fully captured by the model. In addition, although *GATA3* ranked highly in Th2 cells, its average ranking was not as strong as expected. Upon further inspection of the $T_H2$ cluster, we observed that *GATA3* was ranked first in an aggregated subset of cells, while its ranking was more dispersed across the entire $T_H2$ group (Fig. 3f), suggesting heterogeneity within the cluster.

Next, we examined the likelihood scores assigned by CellNavi to driver genes across different cell groups. For known driver genes, CellNavi consistently assigned higher likelihood scores within their corresponding cell groups compared to other groups (Fig. 3g), suggesting that these scores accurately prioritize key driver genes. In addition, the scores could be used to distinguish cell states undergoing specific transitions (Fig. 3h,i and Methods), offering an alternative approach for cell state characterization.

## CellNavi predicts key genes during pathogenesis

We then investigated whether CellNavi could predict key genes involved in disease progression, using an in vitro model system of neurodegenerative diseases, specifically the Fernandes dataset[50]. This system comprises induced pluripotent stem (iPS) cell-derived dopaminergic neurons subjected to tunicamycin treatment. Tunicamycin induces endoplasmic reticulum (ER) stress and Parkinson's disease (PD)-like symptoms by inhibiting N-linked glycosylation[51], a process that affects a broad spectrum of proteins post-translationally, without perturbing any single gene directly.

Before this analysis, CellNavi was trained on single-cell CRISPR screen data on iPS cell-derived neurons from a different study, the Tian dataset[52]. While both studies investigate neurodegenerative diseases using human iPS cell-derived neurons, they differ in the source of iPS cells and the differentiation protocols, resulting in the generation of distinct neuron types[50,52,53] (Fig. 4a).

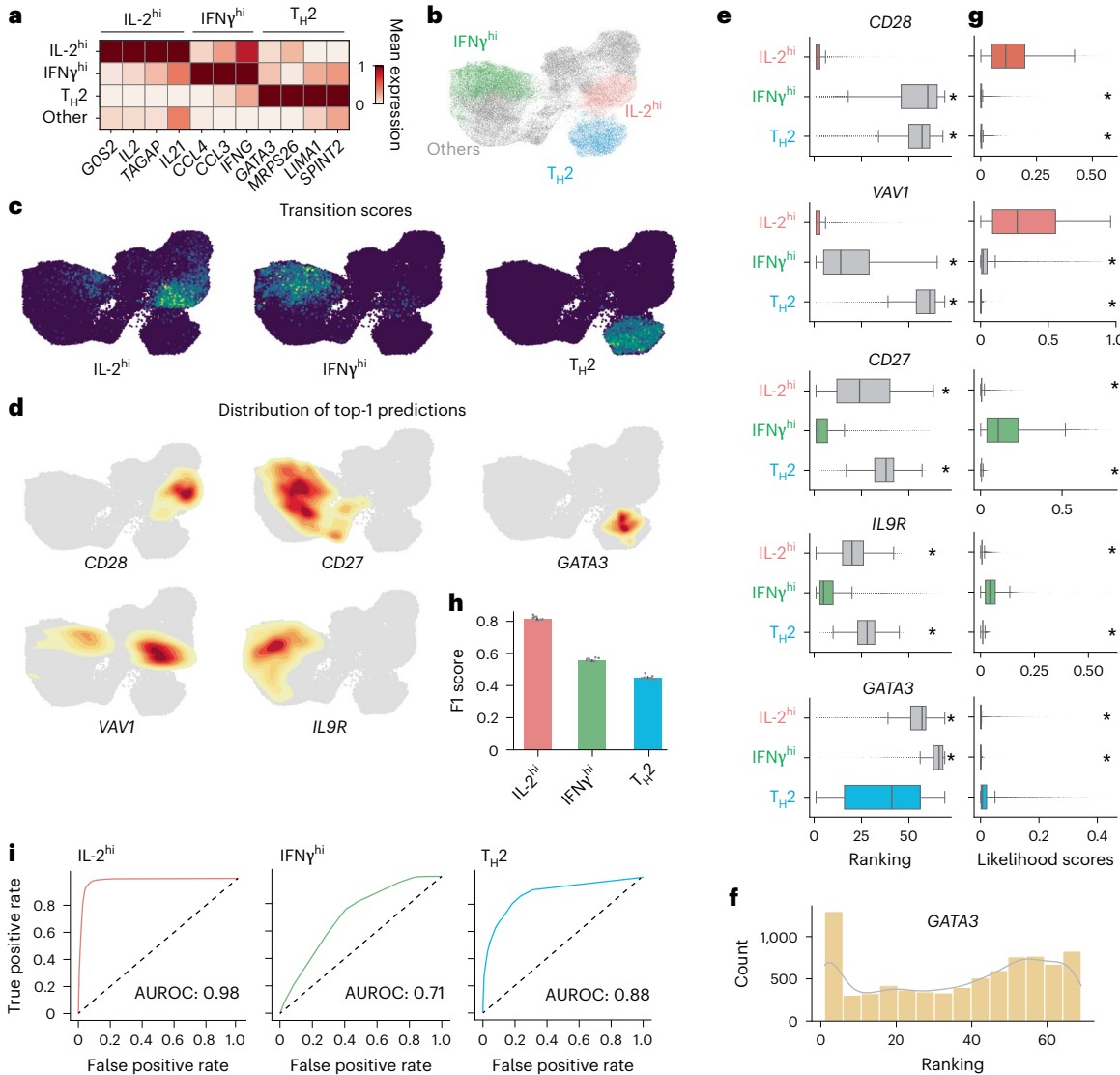

**Fig. 3 | CellNavi identifies key genes involved in T cell differentiation.**
**a**, Changes in expression levels of canonical marker genes corresponding to specific T cell groups. **b**, Uniform Manifold Approximation and Projection (UMAP) visualization of source–target T cell pairs, coloured by effector T cell groups classified on the basis of transition scores. Each data point represents a source–target cell pair representation generated by CellNavi. **c**, Transition scores calculated using IL-2$^{hi}$, IFNγ$^{hi}$ and T$_H$2-related marker genes referenced in (**a**). **d**, Distributions of established driver genes predicted by CellNavi for IL2-high cells (*CD28* and *VAV1*), IFNγ-high cells (*CD27* and *IL9R*) and Th2 cells (*GATA3*). **e**, Predicted rankings of established driver genes across different cell groups. Centre line, median; box limits, upper and lower quartiles; whiskers, 1.5×

interquartile range; points, outliers. $n = 23,342$. $P$ values were calculated with two-sided Mann–Whitney $U$ test. *$P < 1 \times 10^{-6}$. Exact $P$ values are provided in the source data file. **f**, The distribution of predicted rankings for *GATA3* in Th2 cells. **g**, Predicted likelihood scores for established driver genes in different cell groups. Centre line, median; box limits, upper and lower quartiles; whiskers, 1.5× interquartile range; points, outliers. $n = 23,342$. $P$ values were calculated with two-sided Mann–Whitney $U$ test. *$P < 1 \times 10^{-6}$. Exact $P$ values are provided in the source data file. **h**, F1 scores for predicting effector T cell types using likelihood scores. Centre: mean. Error bar: standard error, calculated from tenfold cross-validation (Methods). $n = 10$. **i**, AUROC scores for predicting effector T cell types using likelihood scores (Methods).

After training, we input approximately 47,000 source–target cell pairs from the Fernandes dataset into CellNavi, using untreated cells as sources and cells exposed to tunicamycin as targets. We asked CellNavi to prioritize 184 candidate genes, including 5 known ER stress response genes. CellNavi successfully pinpointed *EIF2S1*, *BAX* and *HSPA5*, which achieved median rankings of 3, 7 and 16, respectively, among the candidate genes (Fig. 4b). However, *HYOU1* and *VCP* ranked lower. One possible explanation is that these genes play more nuanced roles in the ER stress response or are involved in pathways not prominently activated under the specific experimental conditions of this study.

We next examined the top 20 predicted genes for each cell pair. While a total of 31 genes were significantly enriched (Fig. 4c), *FAM57B*,

*EIF2S1*, *NDUFS8*, *BAX* and *CYCS* consistently ranked highest across the majority of cells. Notably, *EIF2S1* and *BAX* are well-established ER stress regulators, while *NDUFS8* and *CYCS* are linked to mitochondrial stress, which is often closely associated with ER stress[54]. In parallel, Fernandes et al. previously identified six subtypes of iPS cell-derived neurons from transcriptomic data and our top 20 predictions revealed subtype-specific gene preferences. For instance, our model suggests that *FARP1*, *CELF1*, *HYOU1* and *APEX1* may play more critical roles in progenitor cells (Fig. 4c). Lastly, except for *HSPA5* and *HYOU1*, most predicted genes showed modest expression changes (Fig. 4d and Extended Data Fig. 7), consistent with previous observations that CellNavi identifies key regulators beyond those detectable by expression shifts alone.

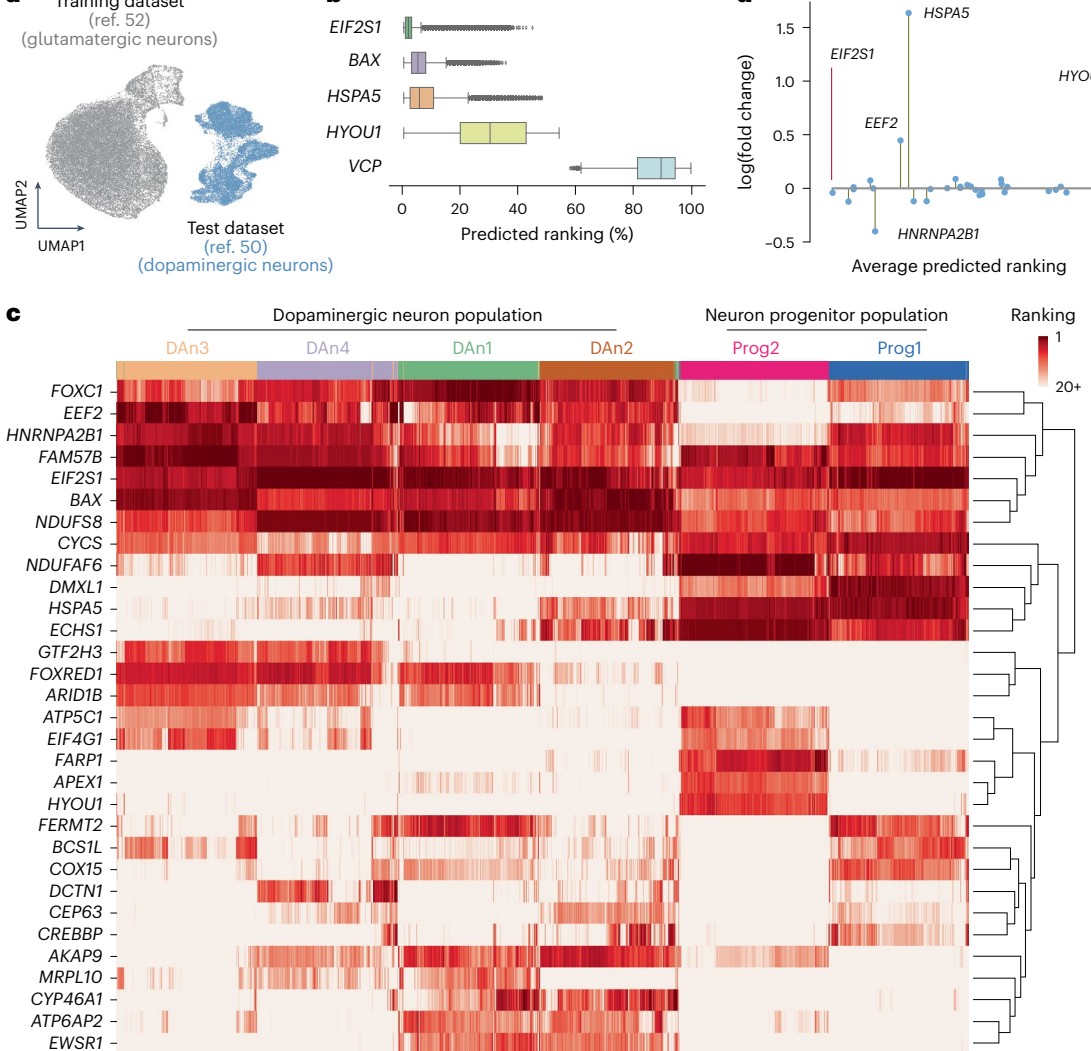

**Fig. 4 | CellNavi predicts key genes involved in neurodegenerative pathogenesis. a**, UMAP visualization of transcriptomic profiles from neurodegenerative disease-related datasets. Grey, iPS cell-derived glutamatergic neurons (Tian dataset[52]) used for model training. Blue, iPS cell-derived dopaminergic neurons (Fernandes dataset[50]). **b**, Predicted rankings for ER stress response-associated genes, based on likelihood score vectors generated by CellNavi. Centre line, median; box limits, upper and lower quartiles; whiskers, 1.5× interquartile range; points, outliers. $n = 47,437$. **c**, The distribution of the top 20 predicted genes across all cell pairs. Rows represent cell types as defined by the original publication[50]. Darker colours indicate higher rankings, and lighter colours indicate lower rankings. Hierarchical clustering was performed using Ward's method. **d**, Expression changes for the top 20 predicted genes. The $x$ axis shows the average ranking of each gene across cell pairs, while the $y$ axis indicates the fold change in expression between target cells and source cells.

## CellNavi reveals mechanisms of action for drug compounds

Understanding the mechanisms of action of novel drug candidates may enhance drug safety and efficacy, reduce development costs and accelerate drug discovery process. However, conventional drug screening paradigms often fall short in elucidating the cellular-level effects that drive biological functions and therapeutic outcomes.

Here, we applied CellNavi to predict key genes modulated by histone deacetylase (HDAC) inhibitors, a class of antitumour drugs with promising therapeutic potential in cancer treatment[55]. HDACs are enzymes integral to post-translational protein modifications and interact with various oncogenic pathways to promote tumour progression[56,57]. The intricate downstream pathways influenced by HDAC presents a considerable challenge in fully understanding mechanisms through which HDAC inhibitors exert their effects within cells.

For this purpose, we applied CellNavi to a chemical screen that quantified the transcriptomic response of K562 cells to 17 distinct HDAC inhibitors (referred to as the Srivastan dataset)[58]. In this set-up, vehicle-treated cells were designated as sources, while cells exposed to the HDAC inhibitors served as targets. The predicted likelihood score indicated whether a gene was modulated during drug treatment, with higher scores suggesting a more prominent role during treatment with specific HDAC inhibitors. Notably, CellNavi was trained exclusively on genetic perturbations[25].

While the transcriptomic data depicted a mixed response across the inhibitors (Extended Data Fig. 8a), the likelihood score vectors effectively clustered the inhibitors into distinct clusters (Fig. 5a,b and Extended Data Fig. 8b). Further analysis revealed diversity in the top-ranked driver genes (Fig. 5c). Specifically, cells treated with mocetinostat, tucidinostat, entinostat and tacedinaline (grouped in cluster 3) exhibited high scores for mitochondrial-related genes such as *MRPS31* and *NDUFB7*. By contrast, most other compounds prioritized genes related to RNA splicing and transcription regulation, such as *PRPF3* and *POLR2A*.

Gene Ontology (GO) enrichment analysis of the top 50 genes predicted for each inhibitor revealed a consistent pattern (Fig. 5d and

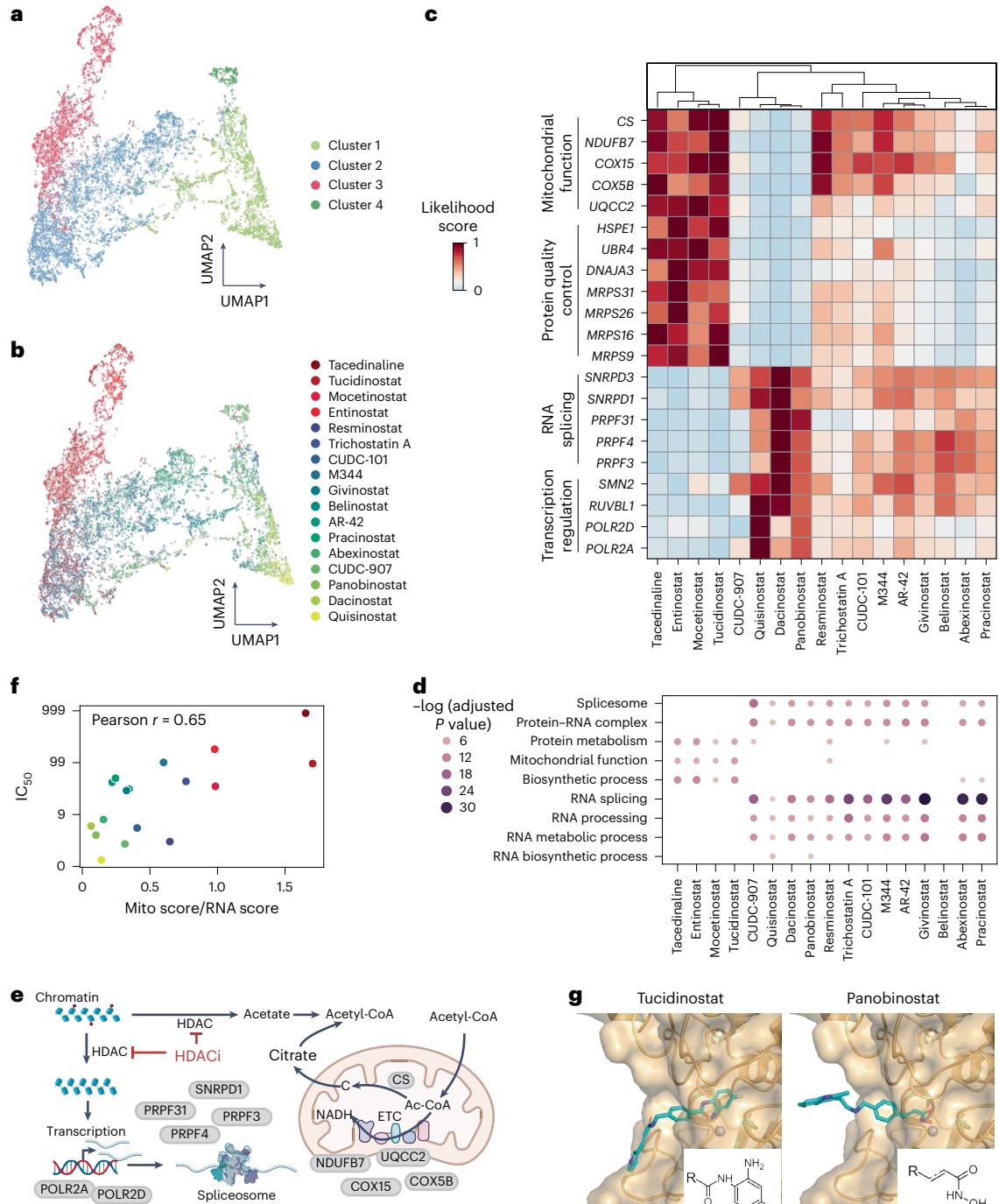

**Fig. 5 | CellNavi reveals diverse downstream gene programs affected by HDAC inhibitors. a,b**, UMAP visualization of cells treated with HDAC inhibitors. Each cell is represented as a 2,057-dimensional vector consisting of likelihood scores predicted by CellNavi for each candidate driver gene. Cells are coloured by clusters identified using the Leiden algorithm (**a**) and by HDAC inhibitor type (**b**). **c**, Average likelihood scores for top-ranked genes in each treatment group, with hierarchical clustering performed using Ward's method. **d**, GO enrichment analysis for each treatment group. The size and darkness of the dots correlate negatively with the adjusted *P* value (one-sided Fisher's exact test with Benjamini–Hochberg correction for multiple comparisons). See Supplementary Fig. 1 for a complete list of GO enrichment results. **e**, A schematic representation of HDAC inhibitor mechanisms. Proteins encoded by top-ranked driver genes

are shown in grey, and red dots on chromatin indicate histone acetylation. ETC, electron transport chain; Ac-CoA, acetyl-CoA. Diagram created with BioRender. com. **f**, A scatter plot showing the correlation between $IC_{50}$ values and the functional selectivity predicted by CellNavi. Each dot represents a compound, coloured according to the clusters in (**b**). Mito score, averaged likelihood scores for genes involved in mitochondrial functions. RNA score, averaged likelihood scores for genes involved in RNA regulation. **g**, Binding modes of tucidinostat and panobinostat at the active site of the zinc-dependent HDAC2 enzyme, with the enzyme represented as a surface representation and the drug compounds in stick representation. Shared warhead (or structural) motifs of different compound classes are highlighted in the bottom right corner. See Supplementary Fig. 2 for a complete list of molecular structures.

Supplementary Fig. 1): compounds in cluster 3 were enriched for genes involved in biosynthetic processes, mitochondrial function and protein metabolism, whereas compounds in other clusters were enriched in gene programs related to RNA splicing, processing and metabolism. These findings align with the known effect of deacetylation inhibition, which lowers cytoplasmic acetate levels and alters acetyl-CoA concentrations, a key metabolite involved in cellular metabolism[58]. Moreover, the results suggest that certain HDAC inhibitors may preferentially target chromatin regions regulating RNA processing genes, which are crucial for tumour cell proliferation[59–61] (Fig. 5e).

Intriguingly, we observed a correlation between the selectivity of downstream gene programs and the half-maximal inhibitory concentration ($IC_{50}$) values reported in the literature[58] (Fig. 5f). Specifically, compounds with lower $IC_{50}$ values tend to influence RNA-related pathways, whereas those with higher $IC_{50}$ values were associated with mitochondrial functions. To further explore the molecular basis of this divergence, we examined the interactions between human HDAC2 and either panobinostat (enriched for RNA-related genes) or tucidinostat (enriched for mitochondrial-related genes). Although molecular docking revealed no major differences in their potential interactions with the zinc-dependent HDAC protein, the aniline group in tucidinostat allowed it to embed more deeply into the HDAC2 pocket (Fig. 5g). Interestingly, all four compounds in cluster 3 shared similar warheads, a feature absent in other compounds (Fig. 5g and Supplementary Fig. 2). This structural feature introduces a steric effect that may influence the efficacy of compounds[62] and lead to divergent downstream response, a phenomenon known as functional selectivity[63–66]. However, the mitochondrial preference and lower potency of compounds like tucidinostat may also result from higher lipophilicity, which can promote off-target or non-specific effects. Nonetheless, these findings highlight CellNavi's potential to elucidate the intricate mechanisms of action underlying drug interventions, highlighting an approach to optimize drug efficacy and specificity for targets involving complex downstream signalling pathways.

## CellNavi generalizes to novel cell types

Lastly, we evaluated the generalization capability of CellNavi. We focused on a CRISPR interference screen across HEK293FT and K562 cell lines[67]. The cell types are markedly different in origin and characteristics—HEK293FT cells are derived from human embryonic kidney cells, while K562 cells are derived from human chronic myelogenous leukaemia (Fig. 6a). In this experiment, CellNavi was trained on HEK293FT cells, with all K562 cells held out as the test set (Methods).

For the 16 perturbations targeting the cleavage and polyadenylation regulatory machinery (Fig. 6a), CellNavi achieved a macro F1 score of 0.432 on top-1 predictions (Fig. 6b). The model misclassified some genes encoding components of the CPSF and CSTF complexes, probably due to their similar post-perturbation transcriptomic profiles (Fig. 6c). However, the model performed well in predicting *CPSF6* and *NUDT21*, which exhibit highly similar transcriptomic profiles after perturbation. Interestingly, despite distinct post-perturbation transcriptomic profiles for *RPRD1A* and *RPRD1B* perturbations, the model confused these genes in many cases. As the protein products of these genes form heterodimers to dephosphorylate the RNA polymerase II C-terminal domain[68], the model may be prioritizing functional interactions and shared pathways over expression differences, leading to the misinterpretation of these genes.

By comparing the similarities between cell groups stratified by true versus predicted perturbations, we found that both intra- and interperturbation correlations for predicted labels closely mirrored those of the true labels (Fig. 6c,d and Extended Data Fig. 9). This suggests that cells grouped by predicted perturbations exhibit gene expression signatures highly similar to those grouped by true perturbations. Although prediction accuracy may partly benefit from conserved perturbation effects across cell types, CellNavi remains

effective even when applied to cell types markedly different from those used in training, demonstrating robust generalization across diverse cellular contexts.

## Discussion

Understanding the regulatory mechanisms that govern cell identity and transitions stand a central challenge in cell biology[6,69–72]. In this study, we introduce CellNavi, a deep learning framework designed to identify driver genes—key factors that orchestrate complex cellular transitions—across diverse biological contexts. By modelling cell states on a biologically informed manifold constructed from large-scale single-cell transcriptomic data and gene graph priors, CellNavi achieves accurate and generalizable predictions across multiple tasks and datasets.

Describing cell states on a manifold that captures their biological dimensions has been a long-lasting endeavour[32,73–76]. Here, we utilized a structured gene graph derived from NicheNet to facilitate cell state manifold learning via deep neural networks. NicheNet is a comprehensive gene–gene graph integrating both GRNs and intercellular signalling pathways. This prior improved the accuracy for driver gene prediction compared with alternative or randomized graphs (Extended Data Table 1). Also, integrating prior gene graphs allowed CellNavi to place greater emphasis on transcription factors, which are crucial for defining cell states and orchestrating transitions[3,10,69] (Supplementary Figs. 3 and 4). This explicit focus on regulatory elements provides CellNavi with a distinct advantage to model complex biological processes and highlights the value of graph-based learning in improving model interpretability and biological relevance. However, we caution that attention mechanisms do not equate to mechanistic interpretability. The explainability remains a critical challenge for deep learning models, including CellNavi. Future work should develop tools to visualize and interpret how graph structures and attention dynamics shape predictions of driver genes.

Our construction of cell-type-specific graphs involves removing edges for genes with zero expression, based on a simplified assumption that such genes are unlikely to participate in active regulation. Consistent with the previous practices in single-cell foundation models[12,13,17] and cell-type-specific protein representation[77] learning, we expect this filtering to help reduce noise and highlight biologically relevant interactions. Yet, we recognize that zero expression values may also stem from technical artifacts such as dropout or low sequencing depth, rather than true biological absence. Future studies should assess alternative strategies, such as imputation or single-cell-level network construction[78], to balance denoising and information retention.

Inherent noise in biological data presents a substantial challenge for modelling. To mitigate technical variability, such as dropout events and differences in sequencing depth, we used a downsampling recovery pretraining strategy with a mixed downsampling rate. This strategy aligns input data of varying depths and improves robustness in handling real-world datasets. Additional noise arises from variability in CRISPR perturbation efficiency, including fluctuating perturbation success rates and off-target effects caused by intrinsic cellular stochasticity. Although CRISPR screens provide a rich and diverse dataset for CellNavi training, this noise may lead to inconsistent labels and biased learning. To mitigate this, future efforts could pool data from multiple batches, sources and single guide RNAs to reduce biases associated with specific experimental conditions. In addition, integrating orthogonal perturbation data, such as chemical treatments, could complement CRISPR-based data and further enhance model robustness.

CellNavi represents a pioneering effort to benchmark the performance and generalization capacity of deep learning methods on driver gene identification task. While the results are promising, several limitations remain. First, the current pipeline requires fine-tuning on single-cell CRISPR screen data relevant to the system of interest. While our proof-of-concept test involving HEK293FT and K562 cells

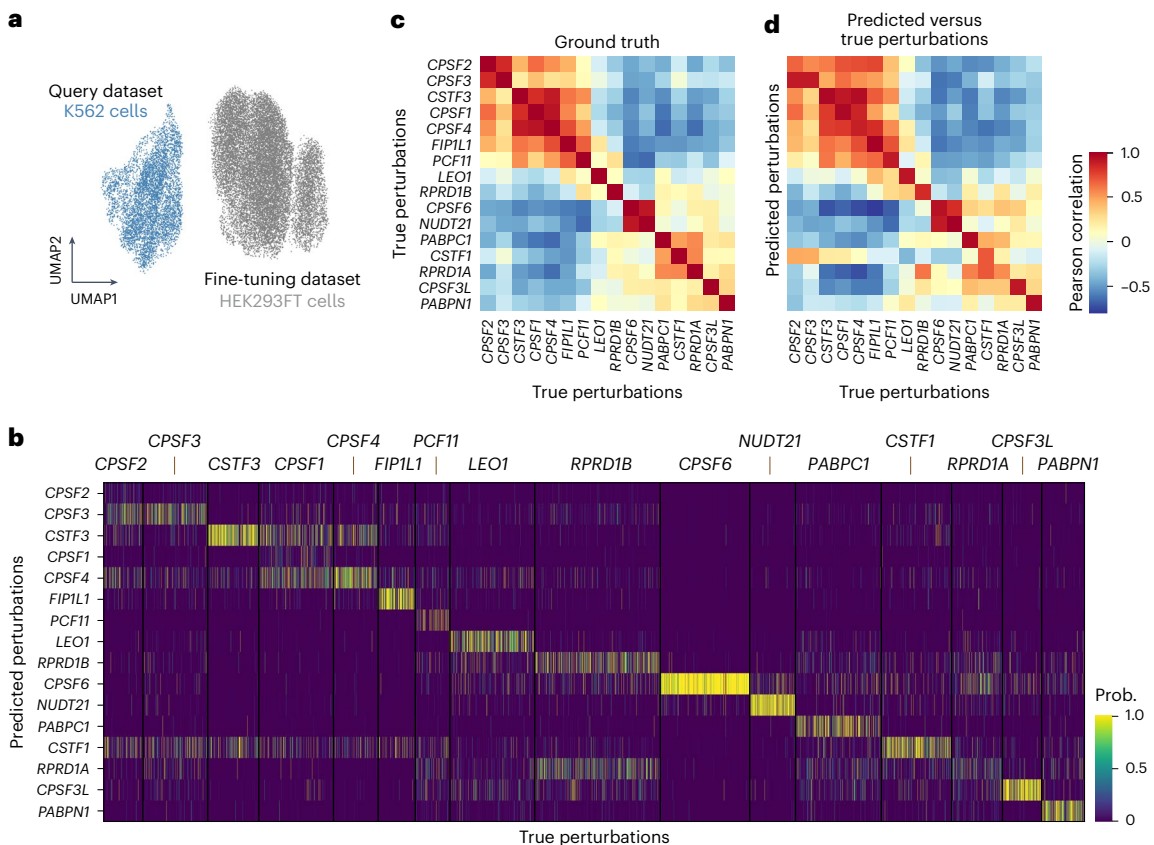

**Fig. 6 | CellNavi predicts driver genes in novel cell types. a**, UMAP visualization of transcriptomic profiles from ref. 67. Grey, HEK293FT cells used for model fine-tuning. Blue, K562 cells for model test. **b**, Predicted perturbations versus true perturbations in K562 cells. Each row represents a predicted perturbation, and each column represents a cell, whose true perturbation is labelled on top. Prob, probabilities of predicted perturbation. **c**, A heatmap showing average Pearson correlations over transcriptomic profiles between each pair of perturbations in K562 cells. **d**, A heatmap showing average Pearson correlations over transcriptomic profiles between predicted perturbations and true perturbations in K562 cells. Row, predicted perturbations. Column, true perturbations.

demonstrated promising results (Fig. 6), the extent to which Cell-Navi can generalize to entirely new cell types or experimental systems remains unclear. Addressing this will require testing across more diverse contexts and quantifying the 'distance' between systems to determine when fine-tuning is necessary. A long-term goal is to reduce the dependence on such datasets by developing models that generalize with minimal experimental effort.

Second, CellNavi cannot yet generalize to novel genes, which limits its broader applicability. Expanding this capacity would require capturing gene networks and representations that enable extrapolation beyond the training dataset. While single-cell CRISPR experiments encompassing a broader range of target genes and cell types are desirable, integrating generative models to infer missing relationships could further improve the model's capacity to handle novel genes.

Third, CellNavi lacks the ability to accurately model long-range transitions owing to its reliance on CRISPR perturbations and static snapshots of transcriptomic data. Many biological processes, such as differentiation and disease progression, unfold gradually through transient states not captured in steady-state data. Incorporating time-resolved single-cell data measurements could help construct dynamic manifolds that better reflect these processes.

Despite these challenges, CellNavi marks a major advance in modelling cell state transitions and identifying their genetic drivers. By combining biologically informed priors with advanced deep learning techniques, CellNavi achieves high accuracy and generalizability in diverse biological contexts. As we continue to refine and expand models like CellNavi, we are paving the way for novel treatments targeting the root causes of diseases with unprecedented specificity.

## Online content

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

## Methods

### Input embeddings

In CellNavi, we use single-cell raw count matrices as the only input. Specifically, the single-cell sequencing data are processed into a cell-by-gene count matrix, $\mathbf{X} \in \mathbb{R}^{N \times G}$, where each element $\mathbf{X}_{n,g}$ represents the expression of the $n$th cell and the $g$th gene (or read count of the $g$th RNA).

To better describe a gene's state in a cell, we involve both gene name and gene expression information in its input embeddings. Formally, the input embedding of a token is the concatenation of gene name embedding and gene expression embedding.

**Gene name embedding.** We use a learnable gene name embedding in CellNavi. The vocabulary of genes is obtained by taking the union set of gene names among all datasets. Then, the integer identifier of each gene in the vocabulary is fed into an embedding layer to obtain its gene name embedding. In addition, we incorporate a special token CLS in the vocabulary for aggregating all genes into a cell representation. The gene name embedding of cell $n$ can be represented as $\mathbf{h}_n^{(\mathrm{name})} \in \mathbb{R}^{(G+1) \times H}$:

$$\mathbf{h}_n^{(\mathrm{name})} = \left[ \mathbf{h}_{n,\mathrm{CLS}}^{(\mathrm{name})}, \mathbf{h}_{n,1}^{(\mathrm{name})}, \mathbf{h}_{n,2}^{(\mathrm{name})}, \dots, \mathbf{h}_{n,G}^{(\mathrm{name})} \right],$$

where $H$ is the dimension of embeddings, which is set to 256.

**Gene expression embedding.** One major challenge in modelling gene expression is the variability in absolute magnitudes across different sequencing protocols[13]. We tackled this challenge by normalizing the raw count expression for each cell using the shifted logarithm, which is defined as

$$\tilde{\mathbf{X}}_{n,g} = \log\left( L \frac{\mathbf{X}_{n,g}}{\sum_{g'} \mathbf{X}_{n,g'}} + 1 \right),$$

where $\mathbf{X}_{n,g}$ is the raw count of gene $g$ in cell $n$, $L$ is a scaling factor and we used a fixed value $L = 1 \times 10^4$ in this study, and $\tilde{\mathbf{X}}_{n,g}$ denotes the normalized count. Finally, a linear layer was applied on the normalized expression $\tilde{\mathbf{X}}_{n,g}$ to obtain the gene expression embedding. For the CLS token, we set it as a unique value for gene expression embedding. The gene expression embedding of cell $n$ can be represented as $\mathbf{h}_n^{(\mathrm{expr})} \in \mathbb{R}^{(G+1) \times H}$:

$$\mathbf{h}_n^{(\mathrm{expr})} = \left[ \mathbf{h}_{n,\mathrm{CLS}}^{(\mathrm{expr})}, \mathbf{h}_{n,1}^{(\mathrm{expr})}, \mathbf{h}_{n,2}^{(\mathrm{expr})}, \dots, \mathbf{h}_{n,G}^{(\mathrm{expr})} \right].$$

The final embedding of cell $n$ is defined as the concatenation of $\mathbf{h}_n^{(\mathrm{name})}$ and $\mathbf{h}_n^{(\mathrm{expr})}$:

$$\mathbf{h}_n = \mathrm{SUM}\left( \mathbf{h}_n^{(\mathrm{name})}, \mathbf{h}_n^{(\mathrm{expr})} \right) \in \mathbb{R}^{(G+1) \times H}.$$

### Cell manifold model

**Model architecture.** The CMM, is composed of six layers of a transformer variant that is designed specifically for processing graph-structured data (GeneGraph attention layers)[19]. The encoder takes the input embeddings to generate cell representations and uses only genes with non-zero expressions. To further speed up training, also as an approach of data augmentation, we performed a gene sampling strategy by randomly selecting at most 2,048 genes as input. It should be noted that the strategy is applied only during training; all non-zero genes are included at inference stage to avoid information loss. We use $\mathbf{h}_n^{(l)}$ to represent the embedding of cell $n$ at the $l$th layer, where $\mathbf{h}_n^{(l)}$ is defined as

$$\mathbf{h}_n^{(l)} = \begin{cases} \mathbf{h}_n, & l = 0, \\ \mathrm{GeneGraphAttnLayer}\left( \mathbf{h}_n^{(l-1)} \right), & l \in [1, 6]. \end{cases}$$

The multihead attention module in each GeneGraph attention layer consists of three components. In addition to a self-attention module, a centrality encoding module and a spatial encoding module are also incorporated to modify the standard self-attention module for graph data integration.

We start by introducing the standard self-attention module. Let $N_{\mathrm{heads}}$ be the number of heads in the self-attention module. In the $l$th layer, $i$th head, self-attention is calculated as

$$\mathbf{Q}_n^{(l,i)} = \mathbf{h}_n^{(l)} \mathbf{W}^{(\mathrm{qry},i)}, \mathbf{K}_n^{(l,i)} = \mathbf{h}_n^{(l)} \mathbf{W}^{(\mathrm{key},i)}, \mathbf{V}_n^{(l,i)} = \mathbf{h}_n^{(l)} \mathbf{W}^{(\mathrm{val},i)},$$

$$\mathbf{A}_n^{(l,i)} = \frac{\mathbf{Q}_n^{(l,i)} (\mathbf{K}_n^{(l,i)})^\top}{\sqrt{D}}, \mathrm{Attn}_n^{(l,i)} = \mathrm{softmax}\left( \mathbf{A}_n^{(l,i)} \right) \mathbf{V}_n^{(l,i)},$$

$$\mathbf{h}_n^{(l)'} = \mathrm{CONCAT}\left( \mathrm{Attn}_n^{(l,1)}, \cdots, \mathrm{Attn}_n^{(l,N_{\mathrm{heads}})} \right) \mathbf{W}^{(\mathrm{out})} \in \mathbb{R}^{(G+1) \times 2H},$$

where $\mathbf{W}^{(\mathrm{qry},i)}$, $\mathbf{W}^{(\mathrm{key},i)}$ and $\mathbf{W}^{(\mathrm{val},i)} \in \mathbb{R}^{2H \times D}$ are learnable matrices that project input embedding $\mathbf{h}_n^{(l)}$ of cell $n$ into $\mathbf{Q}_n^{(l,i)}$, $\mathbf{K}_n^{(l,i)}$ and $\mathbf{V}_n^{(l,i)}$, the symbol $\mathbf{W}^{(\mathrm{out})} \in \mathbb{R}^{(DN_{\mathrm{heads}}) \times 2H}$ is a learnable linear projection that refines the output of multihead attention, and $D$ is the feature dimension for each attention head that satisfies $DN_{\mathrm{heads}} = 2H$. The output of multihead attention $\mathbf{h}_n^{(l)}$ is then passed through a layer normalization layer and a multilayer perceptron (MLP) model, producing the final output $\mathbf{h}_n^{(l+1)}$ as the input to the next layer.

The standard attention mechanism processes features of each individual gene independently, whereas the gene graph incorporates relational information between genes. To incorporate the gene graph information into the model, the centrality encoding module projects the relational information into the regulatory activity feature of each single gene, and the spatial encoding module directly incorporates the relational information with the attention mechanism. More specifically, we define $\mathbf{z}_{\mathrm{deg}^-(\mathcal{G},g)}^-$ and $\mathbf{z}_{\mathrm{deg}^+(\mathcal{G},g)}^+$, learnable embeddings describing in-degree $\mathrm{deg}^-$ and out-degree $\mathrm{deg}^+$ of gene $g$ on the gene graph $\mathcal{G}$. We add these embeddings to the gene embeddings to update cell encoding:

$$\mathbf{h}_{n,g}^{(l)} = \mathbf{h}_{n,g}^{(l)'} + \mathbf{z}_{\mathrm{deg}^-(\mathcal{G},g)}^- + \mathbf{z}_{\mathrm{deg}^+(\mathcal{G},g)}^+.$$

This cell encoding update by the centrality encoding module is applied before the self-attention module.

The spatial encoding module aims to capture regulation relations between genes from the gene graph. For this purpose, we generate the distance matrix $\mathbf{S} \in \mathbb{N}^{G \times G}$, which contains the shortest distances between gene pairs on the gene graph $\mathcal{G}$. We assign each element in $\mathbf{S}$ as a learnable bias added to attention weights:

$$\mathbf{A}'_{g_1 g_2} = \mathbf{A}_{g_1 g_2} + b\left( \mathbf{S}_{g_1 g_2} \right),$$

where $b$ is a learnable scalar-valued function of the distance $\mathbf{S}_{g_1 g_2}$. It assigns a special value to genes that are not connected to the graph. We use $\mathbf{A}'$ in place of the original attention weights $\mathbf{A}$ in the standard self-attention module when computing self-attention in our model. In our implementation, we apply layer normalization and an MLP before computing multihead self-attention. The cell representation output from the CMM, $\mathbf{h}_{n,\mathrm{CLS}}^{(6)}$, is subsequently passed through a fully connected layer, where the dimensionality is increased from 256 to 2,048. This resulting value serves as the cell coordinate for cell $n$, denoted as $\mathbf{CRD}_n$.

**CMM pretraining task.** The CMM is expected to generate cell coordinates that parameterize the intrinsic features and variables (that are much less than the dimensions in the raw gene expression profile representation) of a cell state and maintain cell similarity in the vector space, to provide a concise and biologically relevant representation for the DGP to consume. To achieve this, we design a downsampling reconstruction pretraining task, which asks the CMM to produce a cell coordinates of a downsampled gene expression $\mathbf{X}_n^{(\mathrm{ds})}$ of a cell $n$, that

allows a separate decoder model to reconstruct the original gene expression $\mathbf{X}_n$ of that cell as accurate as possible. To achieve this, the CMM is enforced to capture the co-varying patterns among the raw gene expression dimensions, hence helping the CMM to extract the underlying intrinsic variables.

Specifically, for the downsampling process, we downsample the raw count expression of each gene via a binomial distribution. The downsampled expression $\mathbf{X}_{n,g}^{(\mathrm{ds})}$ of the $n$th cell and the $g$th gene is produced by

$$\mathbf{X}_{n,g}^{(\mathrm{ds})} \sim B\left(\mathbf{X}_{n,g}, \frac{1}{r^{(\mathrm{ds})}}\right),$$

where the $\sim$ denotes 'is distributed as', $\mathbf{X}_{n,g}$ is the raw count of gene $g$ in cell $n$, $r^{(\mathrm{ds})}$ is the downsample rate that is uniformly sampled from $[1, 20]$, and $B$ denotes the binomial distribution. The decoder is an MLP consisting of two linear layers. For each downsampled gene expression, the decoder concatenates the cell coordinates $\mathbf{CRD}_n$ of $\mathbf{X}_n^{(\mathrm{ds})}$ produced by the CMM and the embedding of that gene as the direct input to the MLP. The MLP output comes in the same shape as $\mathbf{X}_n$.

The learning objective for reconstructing the original gene expression profile $\mathbf{X}_n$ from the downsampled version $\mathbf{X}_n^{(\mathrm{ds})}$ is

$$\mathcal{L}_{\mathrm{recons}} = \frac{1}{N} \sum_{n=1}^{N} \|\mathrm{DEC}\left(\mathbf{CRD}\left(\mathbf{X}_n^{(\mathrm{ds})}\right)\right) - \mathbf{X}_n\|^2,$$

where $\|\cdot\|^2$ represents the squared 2-norm of a vector. Both the CMM and the decoder are optimized. After pretraining, the CMM is to be used for driver gene prediction, while the decoder is discarded.

### Driver gene predictor

The driver gene classifier is an MLP consisting of two linear layers. It is optimized to predict the perturbed genes from a pair of cell coordinates output by the CMM. To be more specific, transcriptomes of source cell $\mathbf{X}_{\mathrm{src}}$ and target cell $\mathbf{X}_{\mathrm{tgt}}$ are mapped to cell coordinates $\mathbf{CRD}_{\mathrm{src}}$ and $\mathbf{CRD}_{\mathrm{tgt}}$ with the CMM. For the direct input features, the DGP concatenates the two cell coordinates and then proceeds with an MLP, which outputs the logits of genes. We use the cross-entropy loss for training the DGP:

$$\mathcal{L}_{\mathrm{driver\_gene}} = \mathrm{CE}\left(\mathrm{DGP}\left(\mathrm{CONCAT}\left(\mathbf{CRD}\left(\mathbf{X}_{\mathrm{src}}\right), \mathbf{CRD}\left(\mathbf{X}_{\mathrm{tgt}}\right)\right)\right), g_{\mathrm{drv}}\right),$$

where $\mathrm{CE}\left(\mathbf{l}, g\right) = \frac{\mathbf{l}_g}{\log \sum_{g'} \exp(\mathbf{l}_{g'})}$ is the cross-entropy loss, and $g_{\mathrm{drv}}$ denotes the driver gene corresponding to $\mathbf{X}_{\mathrm{src}}$ and $\mathbf{X}_{\mathrm{tgt}}$. The loss is finally averaged over all $(\mathbf{X}_{\mathrm{src}}, \mathbf{X}_{\mathrm{tgt}}, g_{\mathrm{drv}})$ tuples in the dataset. The pretrained CMM used to produce $\mathbf{CRD}_{\mathrm{src}}$ and $\mathbf{CRD}_{\mathrm{tgt}}$ is also fine-tuned together with the DGP by this loss.

Additional training details for CellNavi are available in Supplementary Note 4.

### Baselines
**SCENIC and SCENIC+.** For each test dataset, SCENIC+ inferred a GRN, identified regulons $\mathbf{W}_r \in \mathbb{R}^{N_r \times N_g}$, and computed regulon activity $\mathbf{W}_a \in \mathbb{R}^{N_c \times N_r}$ in the cells, where $N_r$, $N_g$ and $N_c$ represent the number of identified regulons, genes and cells in the test dataset, respectively. $\mathbf{W}_r$ is a learnt matrix containing the weights of genes for different regulons, and $\mathbf{W}_a$ indicates the regulon activities for each cell. Then, we used $\mathbf{W}_g = \mathbf{W}_a \mathbf{W}_r$ to represent the regulatory importance of each gene in cells. Based on these values (elements in $\mathbf{W}_g$), genes in each cell were ranked, with higher values indicating a greater potential role in controlling cellular identity. We applied SCENIC+ to Norman et al. and Schmidt et al. datasets. Only genes present in the perturbation pools of these datasets were included in the ranking based on $W_g$. Hyperparameters of GRN inference, regulon identification and regulon activation were set to default. Cells with no regulon activated were removed from our analysis. SCENIC+ analysis was realized by pyscenic 0.12.1.

**Other GRNs.** We constructed GRNs using three alternative methods: GRNBoost2, GENIE3 and RENGE, following default parameters from prior studies where applicable. Due to computational memory constraints, we limited the analysis for GENIE3 and RENGE to the top 5,000 highly variable genes. For GENIE3 and GRNBoost2, we utilized the SCENIC implementation to infer GRNs. For RENGE, which is designed to infer GRNs using time-series single-cell RNA sequencing (RNA-seq) data, we adapted the method to work with static single-cell RNA-seq data. After constructing GRNs with these methods, we applied the same downstream analysis protocol as described for the SCENIC pipeline.

**In silico perturbation.** In silico perturbation methods, such as GEARS, are capable of predicting transcriptomic outcomes of genetic perturbations. We trained GEARS model on the datasets mentioned in the corresponding tasks. For evaluation, we computed the cosine similarity between the predicted transcriptomic profiles under various perturbations and the corresponding profiles of cells from the test datasets. Driver genes were predicted on the basis of the similarities, and high values in similarity indicate the potential to be driver genes. GEARS analysis was realized by cell-gears 0.1.1. Data processing and training followed the data processing tutorial (https://github.com/snap-stanford/GEARS/blob/master/demo/data_tutorial.ipynb) and training tutorials (https://github.com/snap-stanford/GEARS/blob/master/demo/model_tutorial.ipynb).

**DGE analysis.** DGE analysis is the most frequently used method to reveal cell-type-specific transcriptomic signature. Initially, cells from the test datasets were normalized and subjected to logarithmic transformation. Subsequently, we applied the Leiden algorithm, an unsupervised clustering method, to categorize the target cells into distinct groups. The number of clusters for each test dataset was set to range from 20 to 40, ensuring that cellular heterogeneity was maintained while providing a sufficient number of cells in each group for robust statistical analysis. We selected source cells to serve as a reference for comparison and performed DGE analysis on each target cell group against this reference. The Wilcoxon signed-rank test was used to determine statistical significance. Then, significant genes were ranked according to their log-fold changes in expression as potential driver genes. Both unsupervised clustering and DGE analysis were conducted using the package scanpy 1.9.6.

### The prior gene graph
The prior gene graph was constructed from NicheNet, where GRN and cellular signalling network were integrated. The gene graph is a directional graph. More specifically, for each gene node on the graph, the number of incoming edges corresponds to the genes that regulate it, while the number of outgoing edges represents the genes it regulates. In our approach, a connection was established between two genes if they were linked in either of the individual networks. The resulting integrated graph features 33,354 genes, each represented by a unique human gene symbol, and includes 8,452,360 edges that signify the potential interactions. The unweighted versions of NicheNet networks were used in our approach. For each cell, we remove the gene nodes with values of 0 in the raw count matrix of the single-cell transcriptomic profile, to construct the cell-type-specific gene graph. During pretraining, when downsampling is performed on single-cell transcriptomes, only the non-zero genes included as model input are retained to generate sample-specific graphs that guide the model's task.

To evaluate the impact of graph connectivity and structure, we generated alternative graph configurations as follows:

(1) Fully connected graph: A maximally connected graph where every pair of genes is connected by an edge of equal weight.
(2) Sparsified graphs: Graphs were created by downsampling the total number of edges from the original graph to 1/10 and 1/20

of its total edges, enabling an evaluation of how reduced connectivity affects performance.

(3) Random graphs: Randomized graphs were generated while preserving the number of nodes and certain structural properties of the original graph, such as self-loops. Edges were introduced probabilistically to maintain overall consistency with the original graph's sparsity and connectivity.

## Datasets

**Human Cell Atlas.** We downloaded all single-cell and single-nucleus datasets sourced from contributors or DCP/2 analysis in *Homo sapiens* up to March 2023, accumulating approximately 1.5 TB of raw data. We retained all experiments that included raw count matrices and standardized the variables to gene names using a mapping list obtained from Ensembl (https://www.ensembl.org/biomart/martview/574df5074dc07f2ee092b52c276ca4fc).

**Norman et al.** This dataset (GSE133344) measures transcriptomic consequences of CRISPR-mediated gene activation perturbations in K562 cell line. We filtered this dataset by removing cells with a total count below 3,500. After filtering, this dataset contained 105 perturbations targeting different genes, and 131 double perturbations targeting two genes simultaneously. We used unperturbed cells (with non-targeting guide RNA) as source cells and perturbed cells as target cells.

**Schmidt et al.** This dataset (GSE190604) measures the effects of CRISPR-mediated activation perturbations in human primary T cells under both stimulated and resting conditions. For our analysis, we excluded cells not mentioned in metadata and removed genes appeared in less than 50 cells. Gene expression levels of single guide RNA were deleted to avoid data leakage. We used unperturbed cells as source cells and perturbed cells as target cells. We also excluded cells without significant changes after perturbation following the procedure proposed by Mixscape tutorial via package pertpy. Default parameters were used for Mixscape analysis.

**Cano-Gamez et al.** This dataset (EGAS00001003215) comprises naive and memory T cells induced by several sets of cytokines. With cytokine stimulation, T cells are expected to differentiate into different subtypes. We took cells not treated by cytokines as source cells and cytokine-stimulated cells as target cells. This experimental set reflects the differential process of human T cells. For our analysis, clusters 14–17 were excluded because their source cells could not be reliably determined.

**Fernandes et al.** This dataset comprises heterogeneous dopamine neurons derived from human iPS cells. These neurons were exposed to oxidative stress and ER stress, representing PD-like phenotypes. We followed preprocessing procedures as mentioned in the original GitHub repo (https://github.com/metzakopian-lab/DNscRNAseq/blob/master/preprocessing.ipynb).

**Tian et al.** This dataset comprises iPS cell-derived neurons perturbed by more than 180 genes related to neurodegenerative diseases. CRISPR interference experiments with single-cell transcriptomic readouts were conducted by CRISPR droplet sequencing (CROP-seq). For our analysis, we removed genes that appeared in fewer than 50 cells. We used unperturbed cells as source cells and perturbed cells as target cells.

**Srivatsan et al.** This dataset (GSE139944) contains transcriptomic profiles of human cell lines perturbed by compounds. For our study, we utilized K562 cell line cells perturbed by HDAC inhibitors. We used unperturbed cells as source cells, and chemically perturbed cells as target cells. This set represents the process of cellular transition caused by drugs.

**Kowalski et al.** This dataset (GEO: GSE269600) measures the transcriptional consequences of CRISPR-mediated perturbations in HEK293FT and K562 cells. For our analysis, we excluded perturbations that consisted of fewer than 200 cells. Cells with minimal perturbation effects were removed from downstream analysis. We used cells from control groups as source cells, and perturbed cells as target cells.

## T cell differentiation analysis

**Identification of cellular phenotypic shift.** We computed the transition score to identify cellular phenotypic shifts on the transcriptomic level. We selected canonical marker genes associated with IFNG and IL2 secretion and Th2 differentiation. Then, we computed the transition score based on the mean expression level of these marker genes, that is, $\mathrm{CTS}_{ij} = \frac{1}{K_i} \sum_{k_i=1}^{K_i}(g_{j_1 k_i} - g_{j_0 k_i})$, where $\mathrm{CTS}_{ij}$ is the transition score of phenotypic shift type $i$ in source–target cell pair $j$, $g_{j_1 k_i}$ and $g_{j_0 k_i}$ are normalized gene expression levels of marker gene $k$ for cell-state transition type $i$ in target cell $j_1$ and its source cell $j_0$. The total number of marker genes for phenotypic shift type $i$ is represented with $K_i$. Then, classes of phenotypic changes were annotated on the basis of transition score. Transition scores are calculated via function tl.score_genes from scanpy package with default parameters.

**Cell type classification with predicted driver genes.** We selected a series of genes related to transition mentioned above from previous studies (Supplementary Table 4). We used the term 'likelihood scores' to describe the probability of a gene to be a driver factor predicted by the model, that is,

$$\mathrm{LS}_{jk}^{\mathrm{mod}} = p_{jk}^{\mathrm{mod}},$$

where $\mathrm{LS}_{jk}^{\mathrm{mod}}$ means the likelihood score for gene $k$ in source–target cell pair $j$ from model mod, and $p_{jk}^{\mathrm{mod}}$ represents the probability predicted by model mod. For our analysis, the mod could be CellNavi, or baseline models.

Then, we aggregate likelihood scores into 'prediction scores' to evaluate the performance of different models:

$$\mathrm{PS}_{ij}^{\mathrm{mod}} = \frac{1}{m_i} \sum_{k_i=1}^{m_i} \mathrm{LS}_{jk}^{\mathrm{mod}},$$

where $\mathrm{PS}_{ij}^{\mathrm{mod}}$ is the prediction score of cell-state transition type $i$ in source–target cell pair $j$ predicted by model mod. The number of candidate driver genes for each phenotypic changing type $i$ is $m_i$. Ideal prediction should reflect similar patterns as shown by the cellular transition score mentioned above. To evaluate it quantitatively, we trained decision tree classifiers with prediction scores as input to test whether predictions scores would faithfully demonstrate cell-state transition types. Classifiers were trained for each method independently, and tenfold cross-validation was conducted. Classifiers were implemented via shallow decision trees using the sklearn package.

## GO enrichment analysis

We used GO enrichment analysis to explore drugs' mechanisms of action. For each drug compound, the top 50 genes with highest scores predicted by CellNavi were used for GO enrichment analysis. The significant level was chosen to be 0.05, and the Benjamini–Hochberg procedure was used to control the false discovery rate. For implementation, we used package goatools for GO enrichment analysis.

## Molecular docking

We performed molecular docking for panobinostat and tucidinostat, with a reference protein structure obtained from the PDB entry 3MAX. The ligand structures from PDB entries 3MAX and 5G3W were used to guide the initial placement of panobinostat and tucidinostat, ensuring the pose correctness of the warheads and major scaffolds. Based on

such initial poses, local optimizations were performed with AutoDock Vina. PyMol was used for structure visualization.

**Reporting summary**

Further information on research design is available in the Nature Portfolio Reporting Summary linked to this article.

## Data availability

HCA data for CMM training were downloaded from the HCA data explorer (https://explore.data.humancellatlas.org/projects). The Norman et al.[32], Tian et al.[52] and Srivatsan et al.[58] datasets were downloaded from the scPerturb project[79] via Zenodo at https://doi.org/10.5281/zenodo.7041848 (ref. 80). The raw count data of Schmidt et al.[30] dataset were downloaded from the National Institutes of Health GEO with accession number GSE190604, and its metadata were downloaded via Zenodo at https://doi.org/10.5281/zenodo.5784650 (ref. 81). The Cano-Gamez et al.[38] dataset was downloaded from the Open Target Platform of this project (https://www.opentargets.org/projects/effectorness). The Fernandes et al.[50] dataset was downloaded from ArrayExpress with accession number E-MTAB-9154. The preprocessed Kowalski et al.[67] dataset was downloaded via Zenodo at https://doi.org/10.5281/zenodo.7619592 (ref. 82). For trajectory reconstruction, we used the dataset from GSE132188. For single-cell RNA-seq alignment across varying sequencing depths, we used data from GSE84133, specifically the Human3 sample. PDB 3MAX, 5G3W. Source data are provided with this paper.

## Code availability

Custom code developed in this study is available via GitHub at https://github.com/DLS5-Omics/CellNavi. Additional software packages for modelling and data analysis include the following: python = =3.8.19, torch = =2.4.0, pandas=2.2.2, numpy = =1.23.5, scikit-learn = =1.5.1, scipy = =1.14.1, networkx = =3.3, scanpy = =1.10.3. cell-gears = =0.1.2, pyscenic = =0.12.1, renge = =0.0.3, cell-gears=0.1.1, goatools = =1.4.12, pertpy (2024.04).

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

## Acknowledgements

We thank Z. Chen, S. Shao, C. Cao and Z. Tang for constructive suggestions and feedback on the manuscript and T.-Y. Liu for the supervision. We thank J. Bai for graphic designs and illustrations.

## Author contributions

Conceptualization, P.D., T.W., S.Z. and C.L.; data curation, T.W. and P.D.; methodology, Y.P., F.J., T.W., S.Z., C.L. and P.D.; model implementation—pretrain, F.J., Y.P., G.L. and H.X.; model implementation—fine-tuning and inference, Y.P., F.J., T.W., C.L. and Q.J.; result analysis, T.W., Y.P., F.J., P.D., Y.M. and X.L.; result interpretation, T.W., P.D., Y.P., F.J and Y.M.; writing—original draft, P.D., T.W., C.L., S.Z., F.J., Y.M. and Y.P.; writing—revision, P.D., C.L., S.Z., H.L., Y.P. and H.X.; supervision, H.L. All authors have read and approved the paper.

## Competing interests

P.D., F.J., C.L., G.L. and H.L. are paid employees of Microsoft Research. The other authors declare no competing interests.

## Additional information

**Extended data** is available for this paper at https://doi.org/10.1038/s41556-025-01755-1.

**Correspondence and requests for materials** should be addressed to Pan Deng.

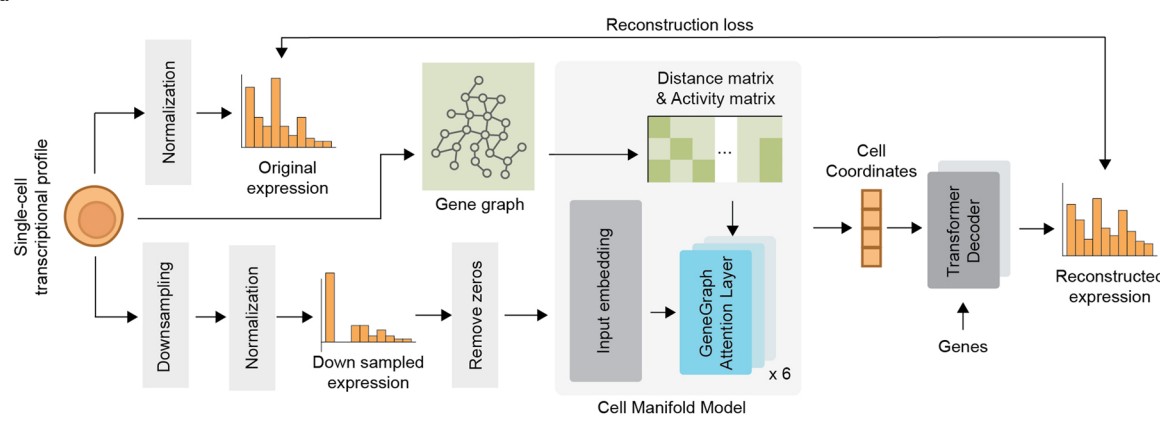

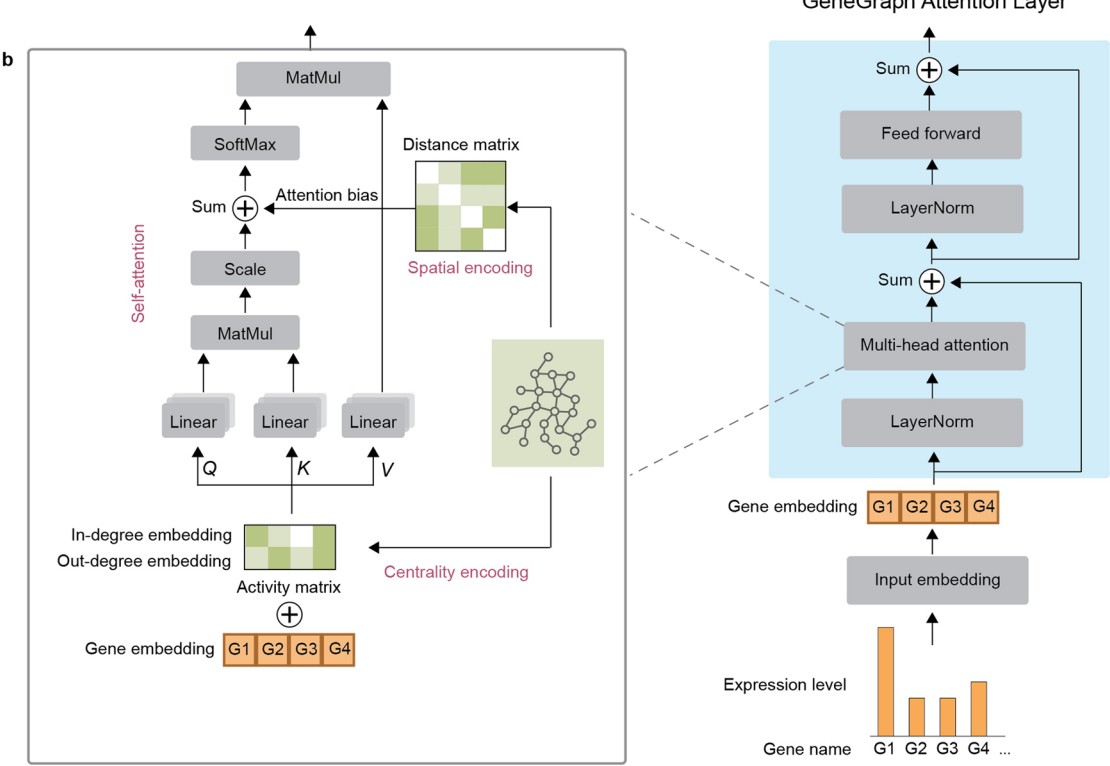

**Extended Data Fig. 1 | Details of the Cell Manifold Model (CMM) implementation. a**) The CMM is designed to reconstruct gene expression profiles by leveraging a Transformer variant composed of GeneGraph Attention Layers. **b**) GeneGraph Attention Layer integrates prior gene graph via its multi-head attention layer. To be noted, only sub gene graphs with nodes in the input sample (all non-zero genes) are used for attention calculation.

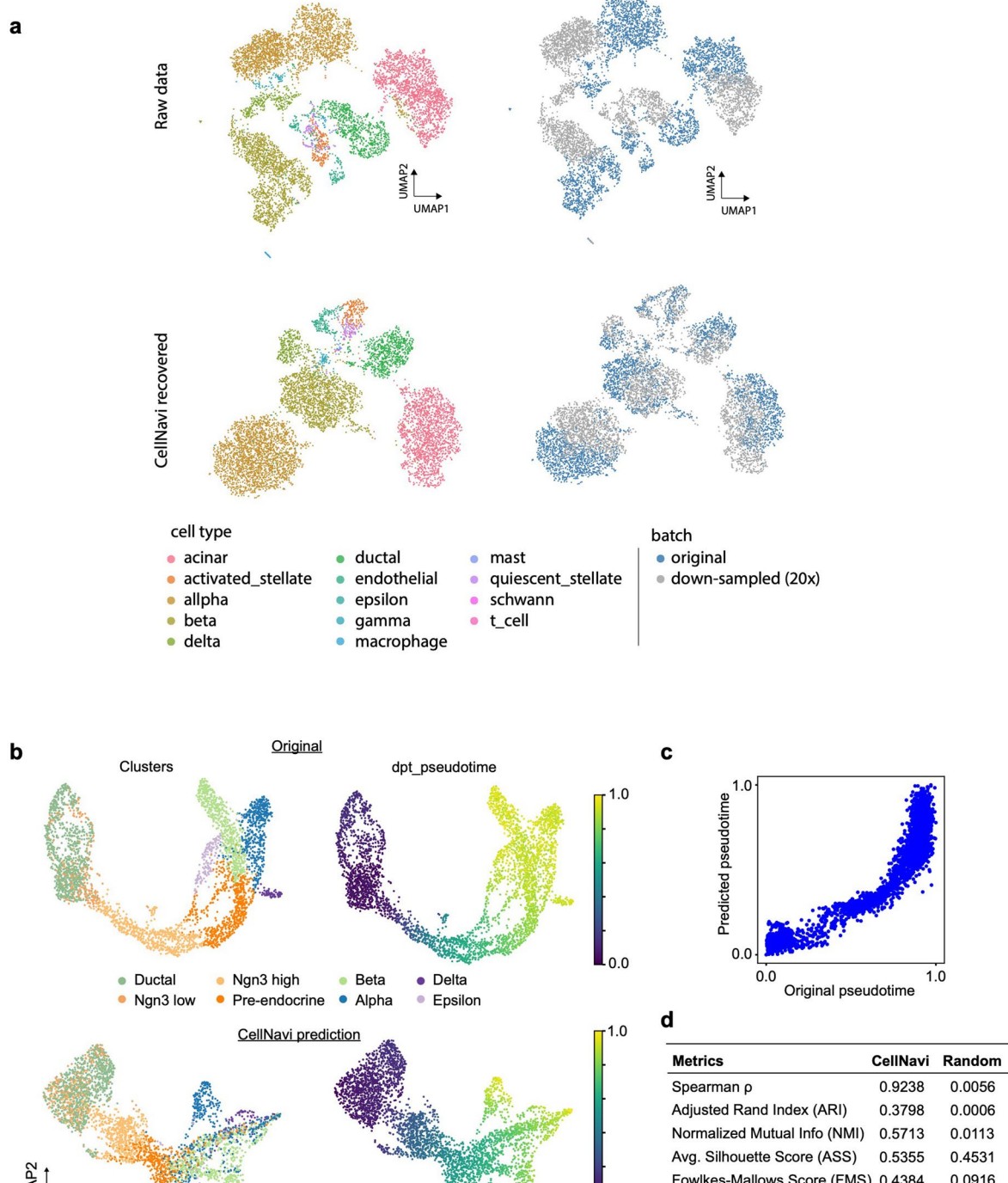

**Extended Data Fig. 2 | CellNavi can align the same cell types across varying sequencing depths and reconstruct developmental trajectories. (a)** Upper panel: After 20-fold down-sampling on the expression profile[83], major cell types remained distinguishable, but a severe batch effect was observed even after using standard integration methods. Lower panel: the Cell Manifold Model (CMM) in CellNavi can integrate embeddings from the down-sampled and original profiles, while preserving the biological structure. Integration quality was quantitatively assessed using iLISI (1: worst, 2: best) and cLISI (1: best, 2: worst), which evaluate batch mixing and cell type separation, respectively. See details of experimental rationales in Supplementary Note 2. **(b)** UMAPs colored by cell type (left) and diffusion pseudotime (DPT) (right, indicated by the colorbars), inferred from the original data (upper panel) and CellNavi-predicted data (lower panel). Cell type labels were assigned based on known markers, and pseudotime was computed using Scanpy's DPT implementation. **(c)** Spearman correlation between pseudotime inferred from the original data (x-axis) and from the CellNavi-predicted data (y-axis) with a quantification in d. **(d)** Summary of quantitative evaluation using Spearman correlation, along with clustering evaluation metrics including Adjusted Rand Index (ARI), Normalized Mutual Information (NMI), Average Silhouette Score (ASS), and Fowlkes-Mallows Score (FMS). Scores computed from random embeddings are included for comparison. We used a mouse pancreatic endocrinogenesis dataset from Bastidas-Ponce et al.[84]. Therefore, we enhanced our model with mouse scRNA-seq for this task.

**a**

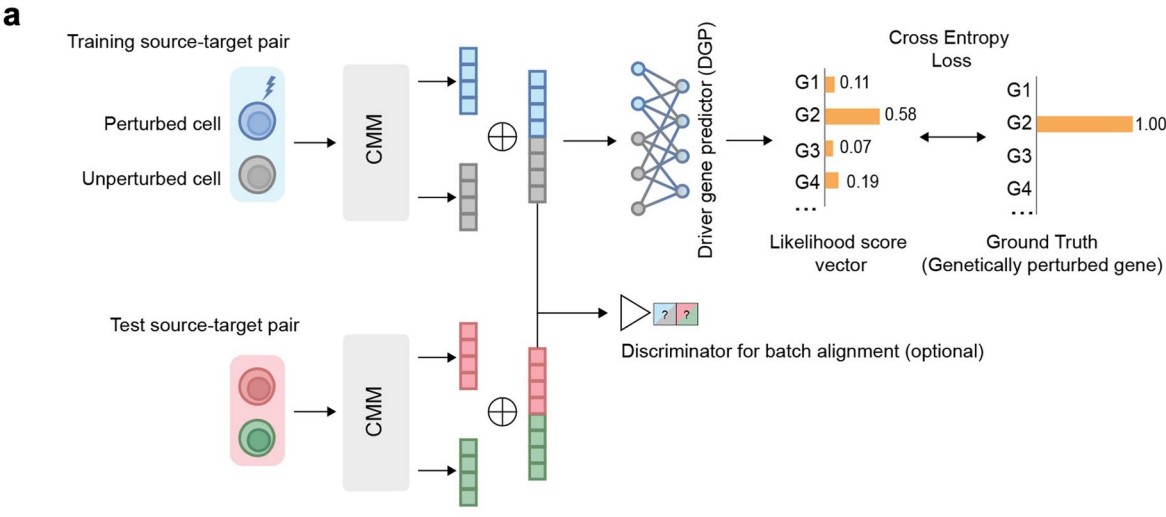

**b**

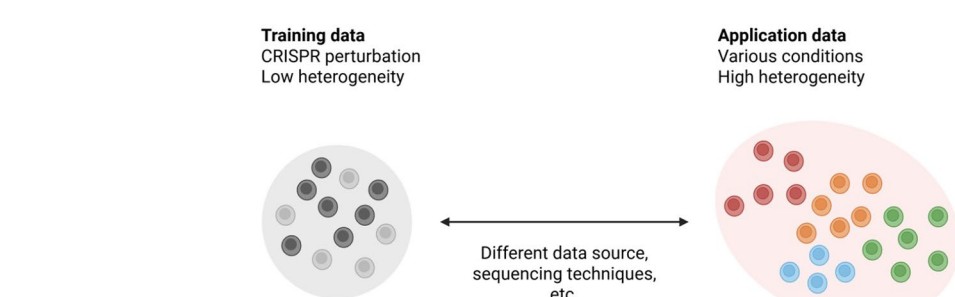

**Extended Data Fig. 3 | Details of the Driver Gene Predictor (DGP) implementation.** (**a**) The DGP processes concatenated cell coordinates pairs output by the CMM to predict driver genes by generating a likelihood score vector. An optional discriminator is included to align training and test data, ensuring consistency and accuracy in predictions (Supplementary Note 4). (**b**) The fine-tuning data (left) consists of CRISPR perturbation datasets with low heterogeneity, typically derived from controlled experiments with limited variability. The application data (right) encompasses diverse biological conditions with high heterogeneity, including different cell types, perturbation types, and sequencing platforms. The bidirectional arrow indicates the challenges posed by differences in data sources, sequencing techniques, and biological variability when generalizing the model from fine-tuning to real-world applications. Schematic elements created with BioRender.com.

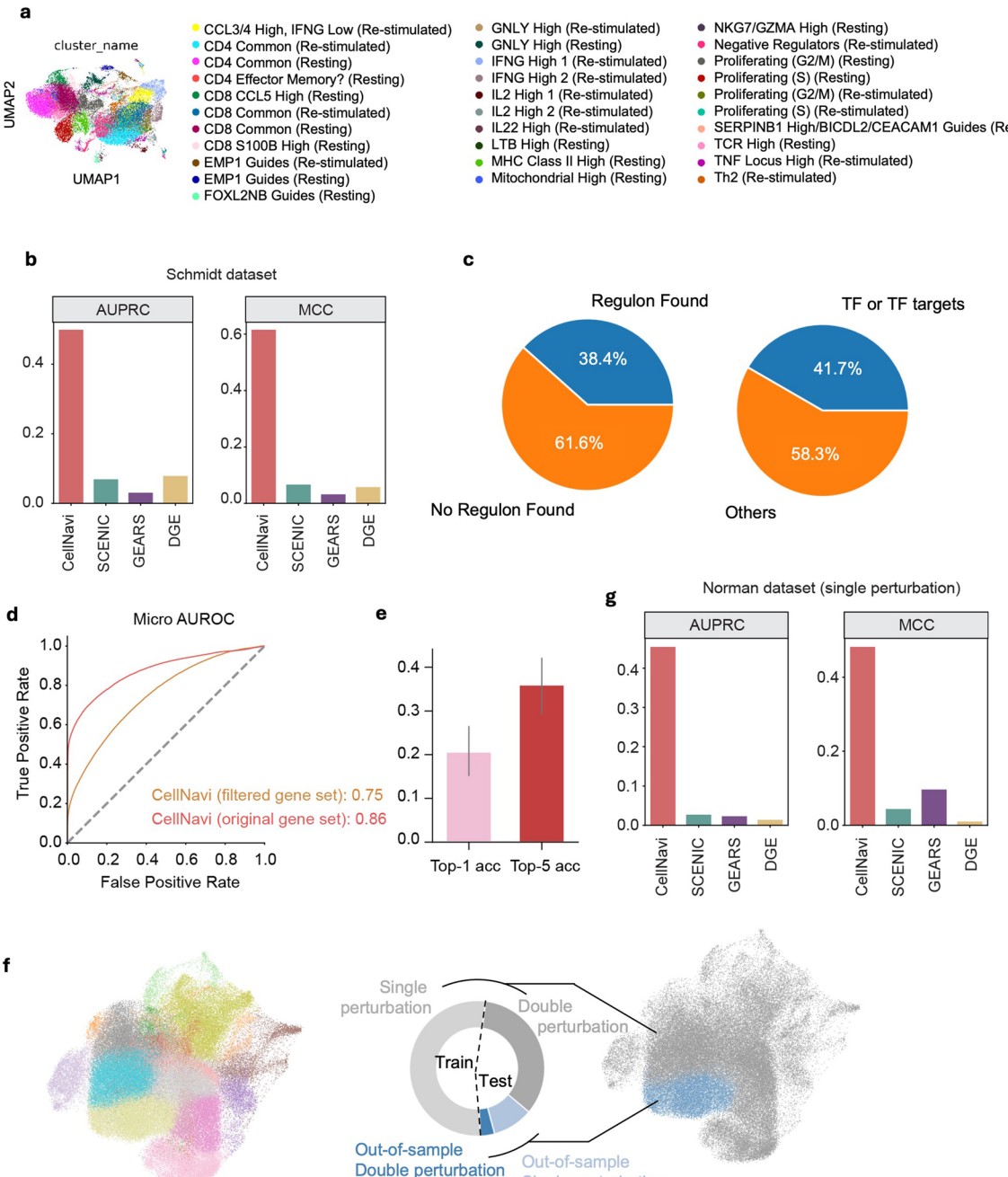

**Extended Data Fig. 4 | Quantitative assessment of CellNavi (extended).**
(**a**) UMAP visualization of perturbed resting T cells and restimulated T cells sequenced by Schmidt et al, colored by cell clusters identified in the original study. (**b**) AUPRC (Area Under Precision-Recall Curve) and MCC (Matthews Correlation Coefficient) for driver gene prediction in the Schmidt dataset, comparing CellNavi with alternative methods. Source data are available in Supplementary Table 1. (**c**) Compatibility of the test dataset to SCENIC/SCENIC+. Upper panel: 61.6% of samples cannot be predicted by SCENIC/SCENIC+ due to missing regulons. Bottom panel: 58.3% of candidate driver genes are excluded because they are neither transcription factor (TF) nor TF-target genes. (**d**) Micro-AUROC of CellNavi applied to the original single-cell transcriptomic profile and the filtered transcriptomic profile in which perturbed genes are excluded, showing that CellNavi identifies driver genes excluded from gene expression profiles. (**e**) Top $K$ ($K$ = 1 or 5) accuracy of CellNavi applied to transcriptomic data with perturbed gene excluded. Error bar, standard error. $n$ = 69. (**f**) Cells were stratified into distinct states using the unsupervised Leiden algorithm,

with each state represented by a different color. A specific cell cluster was selected as the test set, while the remaining clusters were used for training. To ensure rigorous evaluation, all multi-gene perturbations were excluded from training. Consequently, the training set (light gray) consisted only of single-gene perturbations within certain clusters, while the test set was divided into: 1) single-gene perturbations from the held-out cluster (light blue), 2) double-gene perturbations from the held-out cluster (dark blue), and 3) double-gene perturbations from the training clusters (dark gray). Results reported in the main text are derived from the held-out cluster (blue cluster in the UMAP). To further ensure fairness, the test cluster was shuffled (similar to cross-validation) to obtain robust and unbiased results (Supplementary Table 3). (**g**) AUPRC (Area Under Precision-Recall Curve) and MCC (Matthews Correlation Coefficient) for driver gene prediction in the Norman dataset (single perturbation), comparing CellNavi with alternative methods. Source data are available in Supplementary Table 1.

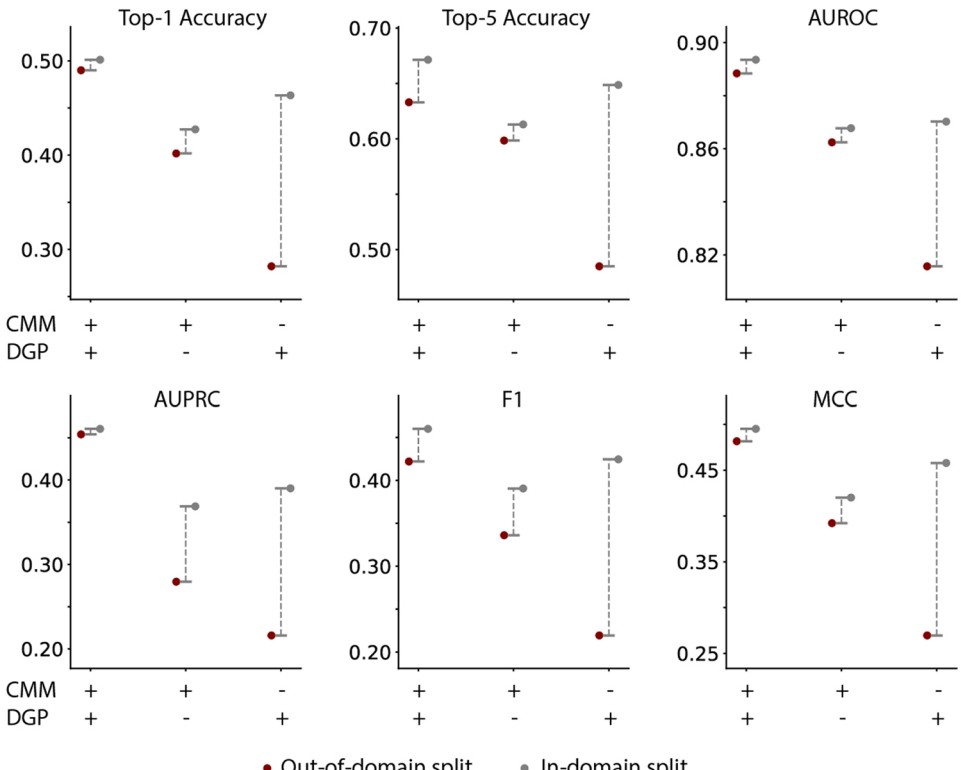

● Out-of-domain split ● In-domain split

**Extended Data Fig. 5 | Ablation study on CellNavi components.** The figure compares the performance of CellNavi under three configurations: (1) CellNavi with both CMM pretraining and DGP fine-tuning, (2) coupling the DGP with raw gene expression vectors instead of gene embeddings produced by the CMM, and (3) replacing the DGP with a simpler multinomial logistic regression model on top of the CMM. Performance is assessed under two evaluation settings: out-of-domain (the previously used Norman single perturbation split that holds one cluster out from training, red) and in-domain (a random split on the Norman dataset with the same train/test size, gray). Metrics include Top-1 accuracy, Top-5 accuracy, AUROC, AUPRC, F1 score, and MCC (Matthews Correlation Coefficient). Dashed lines indicate performance change between evaluation settings.

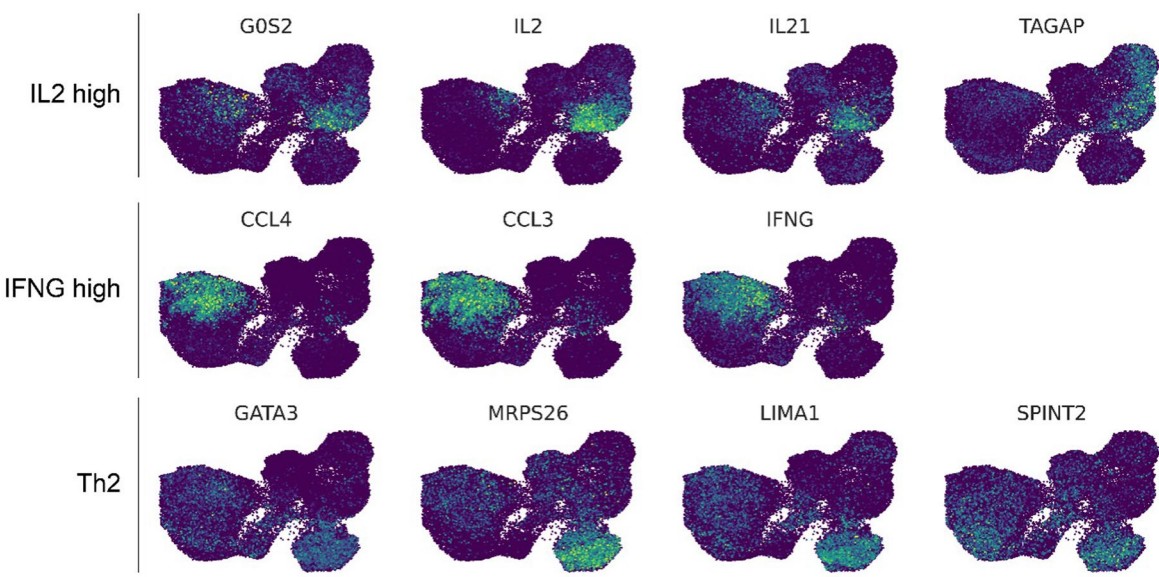

**Extended Data Fig. 6 | Transcriptional changes of canonical marker genes in IL2-high, IFNG-high, and Th2 cell types.** IL2-high marker genes: *G0S2, IL2, IL21, and TAGAP*. IFNG-high marker genes: *IFNG, CCL3, CCL4, and CCL3L3*. Th2 marker genes: *GATA3, MRPS26, LIMA1 and SPINT2*.

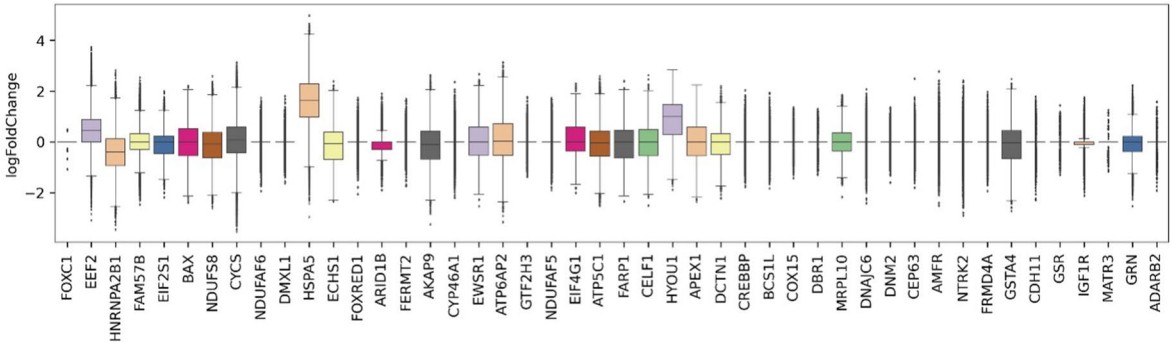

**Extended Data Fig. 7 | Expression changes for the top-20 predicted genes across cell pairs.** Center line, median; box limits, upper and lower quartiles; whiskers, 1.5x interquartile range; points, outliers. $n$ = 47,437. Numerical data are available in the SI source data file.

a

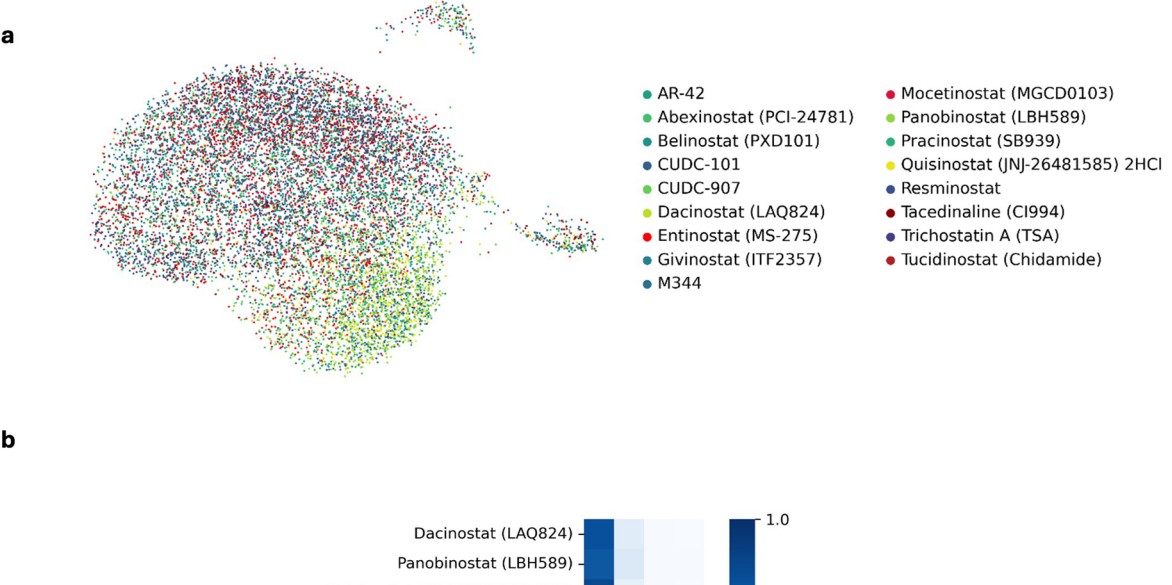

b

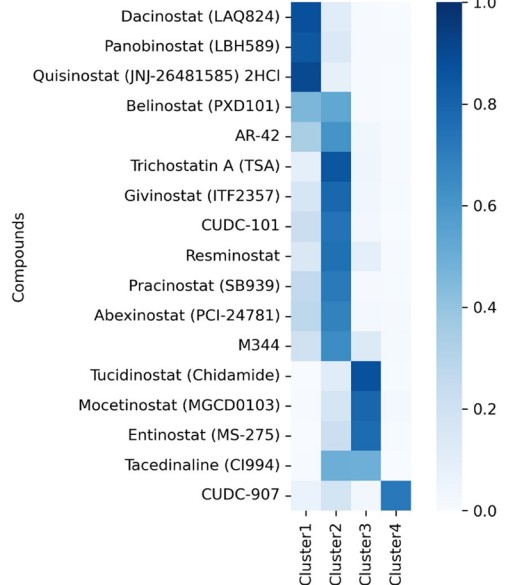

**Extended Data Fig. 8 | Stratification of HDAC inhibitor-treated K62 cells. (a)** UMAP visualization of single-cell transcriptomic profiles from K562 cells treated with 17 distinct HDAC inhibitors. (**b**) The percentage of cells treated with different drug compounds within each cluster identified in Fig. 5a.

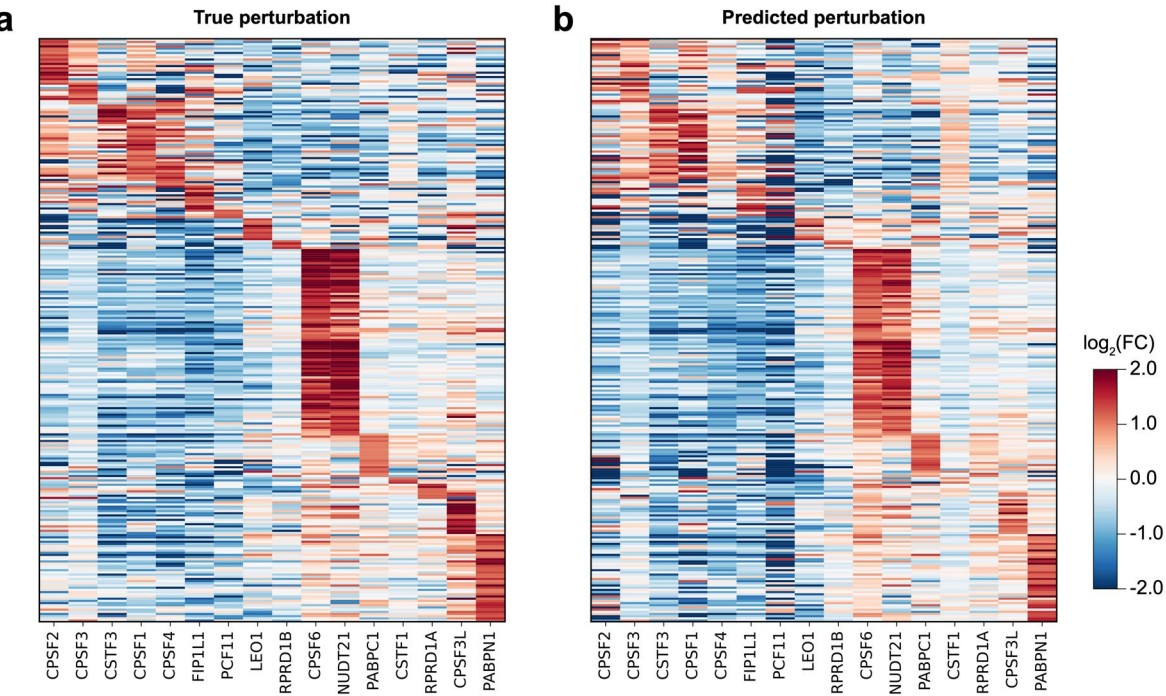

**Extended Data Fig. 9 | Heatmap showing gene expression across different perturbation groups for K562 cells.** Rows represent genes, and columns represent true perturbations (**a**) or predicted perturbations by CellNavi (**b**).

**Extended Data Table 1 | Comparison of the predictive performance of CellNavi using NicheNet and alternative graph configurations**

| | Top-1 accuracy | Top-5 accuracy | AUROC | AUPRC | F1 score | MCC |
|---|---|---|---|---|---|---|
| NichetNet | 0.6206 | 0.7326 | 0.8745 | 0.4991 | 0.6154 | 0.6155 |
| GENIE3 | 0.4916 | 0.6149 | 0.8216 | 0.3642 | 0.5105 | 0.4917 |
| GRNBoost2 | 0.4091 | 0.5201 | 0.8063 | 0.2850 | 0.3982 | 0.4111 |
| RENGE | 0.5600 | 0.6835 | 0.8443 | 0.4487 | 0.5800 | 0.5551 |
| Fully connected graph | 0.3467 | 0.4555 | 0.7484 | 0.2103 | 0.3034 | 0.3468 |
| | 0.3184 | 0.4435 | 0.7525 | 0.1827 | 0.3049 | 0.3147 |
| 1/20 sparsity | 0.3806 | 0.5101 | 0.7794 | 0.2696 | 0.3667 | 0.3853 |
| | 0.3291 | 0.4664 | 0.7746 | 0.1736 | 0.3366 | 0.3358 |
| | 0.3338 | 0.4542 | 0.7721 | 0.2153 | 0.3193 | 0.3347 |
| 1/10 sparsity | 0.3857 | 0.5045 | 0.7821 | 0.2394 | 0.3776 | 0.3857 |
| | 0.3248 | 0.4526 | 0.7690 | 0.2069 | 0.3010 | 0.3236 |
| | 0.3916 | 0.5010 | 0.7835 | 0.2483 | 0.3716 | 0.3893 |
| Random graph | 0.3546 | 0.4839 | 0.7633 | 0.2465 | 0.3511 | 0.3591 |
| | 0.3911 | 0.5287 | 0.7965 | 0.2619 | 0.3711 | 0.3893 |

The table summarizes the predictive performance of CellNavi across different gene graph configurations, including NicheNet and alternative designs, evaluated on the Schmidt dataset. Performance metrics include Top-1 accuracy, Top-5 accuracy, area under the receiver operating characteristic curve (AUROC), area under the precision-recall curve (AUPRC), F1 score, and Matthews Correlation Coefficient (MCC). For sparsified and random graph configurations, three independent graphs were generated and evaluated for each setup.

# Reporting Summary

## Statistics

For all statistical analyses, confirm that the following items are present in the figure legend, table legend, main text, or Methods section.

| n/a | Confirmed | |
|---|---|---|
| ☐ | ☒ | The exact sample size (*n*) for each experimental group/condition, given as a discrete number and unit of measurement |
| ☒ | ☐ | A statement on whether measurements were taken from distinct samples or whether the same sample was measured repeatedly |
| ☐ | ☒ | The statistical test(s) used AND whether they are one- or two-sided<br>*Only common tests should be described solely by name; describe more complex techniques in the Methods section.* |
| ☒ | ☐ | A description of all covariates tested |
| ☐ | ☒ | A description of any assumptions or corrections, such as tests of normality and adjustment for multiple comparisons |
| ☐ | ☒ | A full description of the statistical parameters including central tendency (e.g. means) or other basic estimates (e.g. regression coefficient) AND variation (e.g. standard deviation) or associated estimates of uncertainty (e.g. confidence intervals) |
| ☐ | ☒ | For null hypothesis testing, the test statistic (e.g. *F*, *t*, *r*) with confidence intervals, effect sizes, degrees of freedom and *P* value noted<br>*Give P values as exact values whenever suitable.* |
| ☒ | ☐ | For Bayesian analysis, information on the choice of priors and Markov chain Monte Carlo settings |
| ☒ | ☐ | For hierarchical and complex designs, identification of the appropriate level for tests and full reporting of outcomes |
| ☒ | ☐ | Estimates of effect sizes (e.g. Cohen's *d*, Pearson's *r*), indicating how they were calculated |

*Our web collection on statistics for biologists contains articles on many of the points above.*

## Software and code

Policy information about availability of computer code

| Data collection | All data analyzed within this manuscript are publicly available. No additional software was used for the data collection process. |
|---|---|
| Data analysis | python==3.8.19, torch==2.4.0, pandas=2.2.2, numpy==1.23.5, scikit-learn==1.5.1, scipy==1.14.1, networkx==3.3, scanpy==1.10.3. cell-gears==0.1.2, pyscenic==0.12.1, renge==0.0.3, cell-gears=0.1.1, goatools==1.4.12, pertpy (2024.04), AutoDock Vina, PyMol. Custom code developed in this study: https://github.com/DLS5-Omics/CellNavi |

For manuscripts utilizing custom algorithms or software that are central to the research but not yet described in published literature, software must be made available to editors and reviewers. We strongly encourage code deposition in a community repository (e.g. GitHub). See the Nature Portfolio guidelines for submitting code & software for further information.

## Data

Policy information about availability of data

All manuscripts must include a data availability statement. This statement should provide the following information, where applicable:
- Accession codes, unique identifiers, or web links for publicly available datasets
- A description of any restrictions on data availability
- For clinical datasets or third party data, please ensure that the statement adheres to our policy

HCA data for Cell Manifold Model training is downloaded from the HCA data explorer (https://explore.data.humancellatlas.org/projects). The Norman et al., Tian et al. and Srivatsan et al. datasets were downloaded from the scPerturb project on Zenodo (https://zenodo.org/records/10044268). The raw count data of Schmidt et

# Research involving human participants, their data, or biological material

Policy information about studies with human participants or human data. See also policy information about sex, gender (identity/presentation), and sexual orientation and race, ethnicity and racism.

| | |
|---|---|
| Reporting on sex and gender | N/A |
| Reporting on race, ethnicity, or other socially relevant groupings | N/A |
| Population characteristics | N/A |
| Recruitment | N/A |
| Ethics oversight | N/A |

Note that full information on the approval of the study protocol must also be provided in the manuscript.

# Field-specific reporting

Please select the one below that is the best fit for your research. If you are not sure, read the appropriate sections before making your selection.

☒ Life sciences  ☐ Behavioural & social sciences  ☐ Ecological, evolutionary & environmental sciences

For a reference copy of the document with all sections, see nature.com/documents/nr-reporting-summary-flat.pdf

# Life sciences study design

All studies must disclose on these points even when the disclosure is negative.

| | |
|---|---|
| Sample size | All data used in this study were obtained from publicly available sources, and sample sizes are consistent with those reported in the original publications. No additional selection was performed, except for the data exclusion criteria described below.<br><br>References for test datasets:<br>1. Schmidt, R. et al. CRISPR activation and interference screens decode stimulation responses in primary human T cells. Science 375, eabj4008 (2022).<br>2. Norman, T. M. et al. Exploring genetic interaction manifolds constructed from rich single-cell phenotypes. Science 365, 786–793 (2019).<br>3. Cano-Gamez, E. et al. Single-cell transcriptomics identifies an effectorness gradient shaping the response of CD4+ T cells to cytokines. Nat Commun 11, 1801 (2020).<br>4. Fernandes, H. J. R. et al. Single-Cell Transcriptomics of Parkinson's Disease Human In Vitro Models Reveals Dopamine Neuron-Specific Stress Responses. Cell Reports 33, 108263 (2020).<br>5. Srivatsan, S. R. et al. Massively multiplex chemical transcriptomics at single-cell resolution. Science 367, 45–51 (2020).<br>6. Kowalski, M. H. et al. Multiplexed single-cell characterization of alternative polyadenylation regulators. Cell 0, (2024).<br>7. Baron, M. et al. A Single-Cell Transcriptomic Map of the Human and Mouse Pancreas Reveals Inter- and Intra-cell Population Structure. cels 3, 346-360.e4 (2016).<br>8. Bastidas-Ponce, A. et al. Comprehensive single cell mRNA profiling reveals a detailed roadmap for pancreatic endocrinogenesis. Development 146, dev173849 (2019). |
| Data exclusions | For Schmidt dataset, we excluded cells not mentioned in metadata and removed genes appeared in less than 50 cells. For Cano-Gamez et al., clusters 14-17 were excluded as their source cell could hardly be decided. For Kowalski et al., we excluded perturbations which consist of less than 200 cells. |
| Replication | This is not relevant for our study since we did not perform any wet-lab experiment. The replication of computational experiments were done with cross-validation, by partitioning data into training and testing subsets across multiple folds and evaluate the overall performance of models, when possible (Supplementary Table 4, Supplementary Table 5, Figure 3h), to confirm that model predictions are robust across different data subsets. |
| Randomization | This is not relevant for our study since we did not perform any wet-lab experiment |
| Blinding | This is not relevant for our study since we did not perform any wet-lab experiment |

# Reporting for specific materials, systems and methods

We require information from authors about some types of materials, experimental systems and methods used in many studies. Here, indicate whether each material, system or method listed is relevant to your study. If you are not sure if a list item applies to your research, read the appropriate section before selecting a response.

## Materials & experimental systems

| n/a | Involved in the study |
|-----|-----------------------|
| ☒ | ☐ Antibodies |
| ☒ | ☐ Eukaryotic cell lines |
| ☒ | ☐ Palaeontology and archaeology |
| ☒ | ☐ Animals and other organisms |
| ☒ | ☐ Clinical data |
| ☒ | ☐ Dual use research of concern |
| ☒ | ☐ Plants |

## Methods

| n/a | Involved in the study |
|-----|-----------------------|
| ☒ | ☐ ChIP-seq |
| ☒ | ☐ Flow cytometry |
| ☒ | ☐ MRI-based neuroimaging |

## Plants

| | |
|---|---|
| Seed stocks | *Report on the source of all seed stocks or other plant material used. If applicable, state the seed stock centre and catalogue number. If plant specimens were collected from the field, describe the collection location, date and sampling procedures.* |
| Novel plant genotypes | *Describe the methods by which all novel plant genotypes were produced. This includes those generated by transgenic approaches, gene editing, chemical/radiation-based mutagenesis and hybridization. For transgenic lines, describe the transformation method, the number of independent lines analyzed and the generation upon which experiments were performed. For gene-edited lines, describe the editor used, the endogenous sequence targeted for editing, the targeting guide RNA sequence (if applicable) and how the editor was applied.* |
| Authentication | *Describe any authentication procedures for each seed stock used or novel genotype generated. Describe any experiments used to assess the effect of a mutation and, where applicable, how potential secondary effects (e.g. second site T-DNA insertions, mosiacism, off-target gene editing) were examined.* |

