## [Peer Review File · Nature Cell Biology]

CellNavi predicts genes directing cellular transitions by learning a gene graph-enhanced cell state manifold

Corresponding Author: Dr Pan Deng

Version 0:

Decision Letter:

Dear Dr Deng,

Thank you for your interest in submitting your work to Nature Cell Biology.

I have discussed the information you provided with my colleagues, and we think that the study sounds interesting and could be appropriate for this journal. However, given the limited information provided, we would need to evaluate the complete manuscript before deciding whether to formally review it.

Please use this link to submit the complete manuscript:

Link Redacted

Please feel free to contact me if you have any questions.

Kind regards,

Melina

Melina Casadio, PhD
Senior Editor, Nature Cell Biology
ORCID ID: <https://orcid.org/0000-0003-2389-2243>

Version 1:

Decision Letter:

*Please delete the link to your author homepage if you wish to forward this email to co-authors.

Dear Dr Deng,

Thank you again for submitting your manuscript, "Directing cellular transitions on gene graph-enhanced cell state manifold", to Nature Cell Biology and I sincerely apologize for the delay in obtaining the reviews and sharing our decision with you.

Your manuscript has now been seen by 3 referees, who are experts in computational genomics (Referee #1); computational genomics, gene regulatory networks (Referee #2); and computational genomics, gene regulatory networks (Referee #3). As you will see from their comments (attached below), they found the work of potential interest but have raised substantial concerns, which in our view would need to be addressed with considerable revisions before we can consider publication in Nature Cell Biology.

Nature Cell Biology editors discuss the referee reports in detail within the editorial team, including the Chief Editor, to identify key referee points that should be addressed with priority, and requests that are overruled as being beyond the scope of the current study. To guide the scope of the revisions, I have listed these points below. Our standard revision period is six months and we are committed to providing a fair and constructive peer-review process, so please feel free to contact me if you would like to discuss any of the referee comments further or anticipate any issues or delays addressing the reviews.

In particular, it would be essential to dedicate efforts in revision to address the following points:

-- The reviewers asked for further benchmarking studies, which we agree are necessary for the tool to stand within its ecosystem:

Rev#1 point #8

We also think Rev#1's questions about selecting NicheNet should be addressed (#2)

Rev#3 point #2

-- Please address all reviewer points about the construction of the cell state manifold, the performance of CellNavi, potential biases, and generalizability:

Rev#1 points #3, 4, 5, 7

Please address the request from Rev#1 point #1 and Rev#2 point #1 for ablation studies

Rev#2 points #2, 3, 4, 5, 6, 7, 8, 9, 10

Rev#3 points #1, 3, 4 and we encourage you to address their point #5 if possible

-- All other referee concerns pertaining to strengthening existing data, applications of the method (e.g., Rev#1 point #6), providing controls, methodological details, clarifications and textual changes should also be addressed.

-- Finally, please pay close attention to our guidelines on statistical and methodological reporting (listed below) as failure to do so may delay the reconsideration of the revised manuscript. In particular, please provide:

We would be happy to consider a revised manuscript that would satisfactorily address these points, unless a similar paper is published elsewhere or is accepted for publication in Nature Cell Biology in the meantime.

- ensure that it conforms to our format instructions and publication policies (see below and <https://www.nature.com/nature/for-authors>).

- provide a point-by-point rebuttal to the full referee reports verbatim, as provided at the end of this letter.

- provide the completed Reporting Summary (found here <https://www.nature.com/documents/nr-reporting-summary.pdf>) <https://www.nature.com/documents/nr-reporting-summary.pdf>). This is essential for reconsideration of the manuscript will be available to editors and referees in the event of peer review. For more information see <http://www.nature.com/authors/policies/availability.html> or contact me.

When submitting the revised version of your manuscript, please pay close attention to our <https://www.nature.com/nature-portfolio/editorial-policies/image-integrity> Digital Image Integrity Guidelines and to the following points below:

-- that unprocessed scans are clearly labelled and match the gels and western blots presented in figures.

-- that control panels for gels and western blots are appropriately described as loading on sample processing controls

-- all images in the paper are checked for duplication of panels and for splicing of gel lanes.

Nature Cell Biology is committed to improving transparency in authorship. As part of our efforts in this direction, we are now requesting that all authors identified as 'corresponding author' on published papers create and link their Open Researcher and Contributor Identifier (ORCID) with their account on the Manuscript Tracking System (MTS), prior to acceptance. ORCID helps the scientific community achieve unambiguous attribution of all scholarly contributions. You can create and link your ORCID from the home page of the MTS by clicking on 'Modify my Springer Nature account'. For more information please visit <http://www.springernature.com/orcid>.

This journal strongly supports public availability of data. Please place the data used in your paper into a public data repository, or alternatively, present the data as Supplementary Information. If data can only be shared on request, please explain why in your Data Availability Statement, and also in the correspondence with your editor. Please note that for some

data types, deposition in a public repository is mandatory - more information on our data deposition policies and available repositories appears below.

Link Redacted

We hope that you will find our referees' comments and editorial guidance helpful. Please do not hesitate to contact me if there is anything you would like to discuss. Thank you again for considering the journal for your work.

Best wishes,

Melina

Melina Casadio, PhD
Senior Editor, Nature Cell Biology
Consulting Editor, Nature Structural & Molecular Biology
ORCID ID: <https://orcid.org/0000-0003-2389-2243>

Reviewers' Comments:

Reviewer #1 (Remarks to the Author):

In this manuscript, authors propose a Graph-based transformer model leveraging pathway information to create: (1) cellular embeddings; and (2) prediction of driver genes/drug target from perturbation and drug treatment experiments. A novel aspect is the use of pathway information, which are used to create single cell level graphs. Benchmarking results showed very positive results in particular the evaluation of perturb-seq data (Fig. 2). An unclear aspect here is the impact of the network in the predictions, i.e. by use of ablation experiments, or the choice of causal network (NicheNet). One missing opportunity of CellNavi is the lack of explainability of its prediction, and how they relate to these causal networks. Another unclear aspect about CellNavi is how it can be used in real world problems. The method introduction suggests that this is a foundation model (trained on 40 millions of HCA cells); however individual case studies indicate that novel models are trained in every case. Moreover, trained models are mostly related to the predicted tasks. The need for related train data makes the method applicability very restrictive. We also had issues in running the software due to incomplete documentation.

Major points

1. Authors should include ablation experiments (in experiments shown in Fig. 2) to highlight the importance of the pathway networks in the predictions. This could include the use of random networks, fully connected networks, or sparsified networks.
2. On the same topic, why is the Nichenet pathway adopted? How much would NicheNet be affected if other pathways are used instead?
3. The procedure to obtain networks is quite simple, i.e. removing edges of genes with zero expression. While simplicity is not bad per se, authors could justify this choice.
4. In Figure S1.1a, the scheme shows that down-sampling is performed on expression profiles for self-supervision. The same figure suggests that down sampling does not change how graphs are computed. What is the rationale behind this? Would this generate some information leakage as the graph will have information about zero/non-zero genes?
5. CellNavi users would profit from an explanation of how the methods achieved the predictions. It would be interesting to relate how attention mechanisms support gene predictions in the context of the causal networks.
6. Authors need to clarify how the model can be used in real word applications; and this should be clear from the method introduction. The current figure 1 suggests a foundation model approach, but case studies indicates the necessity of additional training for specific tasks.
7. Expanding point 6, could it be that training the model is HCA data (mostly healthy samples) makes CellNavi not generalizable to other tasks as T cell activation (Fig. 3) or neurodegenerative pathogenesis (Fig. 4).
8. A recent benchmarking indicates that linear models outperforms models as GEARS: <https://www.biorxiv.org/content/10.1101/2024.09.16.613342v4> . Authors should improve the benchmarking of CellNavi (Fig. 2) to consider linear models; and additional data sets as done by Ahlmann-Eltze et al.
9. Case studies shown in figures 3, 4 and 5 do not include competing methods; and are based on anecdotal evaluation.

Authors could shorten this part, and focus on more benchmarking evaluation (see points above).

10. The cell type annotation evaluation, which is reported in the discussion, is out of place. Authors should remove; move somewhere else, or contextualize the need for this.

Minor points

Line 171 indicates “we stratified the cells into distinct states, selecting a specific cell cluster for testing while training on the remaining clusters (Fig. 2a).” This does not fit the figure legend or the figure itself. Please consider improving this.

Abstract mentions casual connections but this is not explained and further supported in the main manuscript.

Reviewer #1 (Remarks on code availability):

The software description was sparse, which makes its execution difficult. I ran into a system-specific error (Error: ValueError: ProcessGroupNCCL is only supported with GPUs, no GPUs found). Authors need to clarify hardware requirements for CellNavi; and include more details for uses. See suggestions below.

Publish on PyPI/Conda: Simplifying the installation process by publishing the software on PyPI or Conda will make it more user-friendly and accessible. A single-command installation (pip install or conda install) would significantly improve usability and adoption.

Create Documentation and using Google Colab : Providing detailed and interactive documentation would help users better understand and use the software. To make the tutorial more engaging and to ensure results can be reproduced effectively, I suggest creating an interactive notebook using Google Colab. This approach allows users to experiment with the software directly in their browser without needing to set up the environment locally, making it more accessible and user-friendly. See <https://github.com/snap-stanford/GEARS?tab=readme-ov-file>.

The current script for launching training (launch_train.sh) is highly system-specific and introduces unnecessary complexity for general users. It assumes certain configurations (e.g., NVIDIA GPUs, distributed training) and lacks flexibility, which can hinder usability.

Key variables like RANK, WORLD_SIZE, and distributed settings are not explained. To improve this add clear documentation or comments to explain the purpose and usage of the script.

GPU detection relies on nvidia-smi, which is specific to NVIDIA systems. It would be better to use a general detection method like torch.cuda.device_count() for broader compatibility

Reviewer #2 (Remarks to the Author):

This manuscript, titled "Directing Cellular Transitions on Gene Graph-Enhanced Cell State Manifold" by Tianze Wang et al., presents CellNavi, an innovative deep learning framework designed to predict key driver genes governing cellular state transitions. Recognizing the challenges in identifying these genes due to the complexity and scale of transcriptomic data, the authors designed CellNavi to integrate single-cell transcriptomics with causal gene graphs, creating a biologically meaningful cell state manifold. This manifold reduces high-dimensional data into a lower-dimensional representation that preserves intrinsic biological features and relationships. CellNavi comprises two core components: the Cell Manifold Model (CMM) and the Driver Gene Predictor (DGP). The CMM captures and encodes cell states using a graph-based Transformer architecture that incorporates gene interaction networks, enhancing the biological relevance of the manifold. The DGP leverages this manifold to identify key driver genes by analyzing transcriptomic changes associated with transitions between source and target cell states. Then, the authors rigorously evaluated CellNavi across a diverse range of tasks and datasets, including cellular differentiation, disease progression, and drug response. Through testing on CRISPR perturbation datasets and benchmarking against established methods such as SCENIC+ and GEARS, the manuscript demonstrates CellNavi's superior predictive performance and generalization capabilities. Notably, the model excels in identifying critical genetic regulators in contexts as varied as T cell differentiation and neurodegenerative disease progression, underscoring its broad applicability. Furthermore, the framework's ability to infer mechanisms of action for drug compounds highlights its potential utility in therapeutic innovation and drug discovery. While the study presents notable advancements, some aspects of this work should be clarified/improved, as detailed below.

Major:

1. The integration of causal gene graphs is a key innovation in this study. To highlight their importance, the authors could conduct ablation studies to assess the impact of gene graphs on model performance. This would provide clearer insights into their critical role and validate their contribution.
2. The quality and completeness of the causal gene graphs likely impact performance. How do the authors address potential biases in model predictions, especially considering that gene graphs might encode biased relationships?

3. The construction of the cell state manifold is central to the model. The exact method for determining the dimensionality of the cell manifold is unclear. What criteria are used to determine the optimal dimensionality of the manifold?
4. The manuscript mentions using down-sampling as part of the pretraining process. How does this affect the fidelity of the learned manifold, and does it introduce biases in the representation?
5. The authors highlight that the model constructs a lower-dimensional manifold for cell states, capturing their intrinsic features while preserving relative similarities between cells. Could the authors provide both quantitative and qualitative evidence to support this claim?
6. The CMM is pretrained with reconstruction tasks before joint training with the DGP. How do pretraining and fine-tuning contribute to final model performance? Is there a trade-off between these stages?
7. Biological data are inherently noisy, which can significantly impact model performance. Evaluating how CellNavi performs under varying degrees of noise in single-cell gene expression data would provide valuable insights into the model's reliability and its capacity to handle noisy, real-world datasets effectively.
8. As depicted in Figure 2a, the manuscript describes using a specific cell cluster as the test set while others are used for training. While this approach assesses generalization across clusters, relying on a single train-test split may introduce bias depending on the selected cluster. Incorporating cross-validation would offer a more comprehensive and reliable evaluation of the model's performance.
9. The manuscript evaluates model performance using AUROC and F1 score (e.g., Figure 2b-c), which offer valuable insights into overall performance. However, including additional metrics, such as precision-recall curves or Matthews Correlation Coefficient (MCC), could provide a more nuanced and comprehensive evaluation of the model's effectiveness.
10. The manuscript provides insights into the performance of CellNavi but does not appear to include details about computational efficiency. What are the runtimes for training and inference, along with the hardware specifications (e.g., GPU, memory) used?

Minor:

1. In line 64, the word "that" following the colon in "that Cell Manifold Model" should be revised to "the" for grammatical accuracy.
2. In line 298, there is a minor spelling error in the figure caption: "cekers" should be corrected to "whiskers," the appropriate term for describing elements of a box plot.

Reviewer #3 (Remarks to the Author):

This manuscript presents CellNavi, a novel deep learning framework for predicting driver genes in cell state transitions. The authors address a crucial and challenging problem, employing a sophisticated approach that integrates manifold learning with prior knowledge from gene interaction graphs. Incorporating these graphs into the Cell Manifold Model is a notable innovation, likely improving the model's ability to capture complex gene-state relationships. The self-supervised pretraining task for the CMM is also a strength, leveraging the abundance of unlabeled single-cell data. The demonstration of CellNavi's performance across diverse biological contexts, from T cell differentiation to drug responses, showcases its potential utility. The framework's ability to generalize across cell types and predict driver genes even in scenarios not explicitly present in the training data, such as drug treatments, is particularly promising.

I have a few questions and suggestions that could further strengthen the manuscript:

1. Sensitivity to Downsampling: The CMM's pretraining incorporates a downsampling step. Could the authors provide a more detailed analysis of the model's sensitivity to the choice of downsampling rate? How do varying rates influence the learned manifold and the downstream driver gene predictions?

2. Comparison with other GRN inference methods: While the comparison to SCENIC+ and GEARS is relevant, benchmarking CellNavi against a wider range of established gene regulatory network inference methods would be valuable. Including methods like GENIE3, GRNBoost2, and single-cell specific methods like RENG (https://www.nature.com/articles/s42003-023-05594-4?fromPaywallRec=false) would provide a more comprehensive performance assessment relative to the state-of-the-art.

3. Limitations of the CRISPR approach: While the CRISPR screens provide a more controlled setting for evaluation than purely observational data, certain limitations should be explicitly addressed. Specifically, the incompleteness of typical CRISPR screens (targeting only a subset of genes) and the potential for off-target effects can influence the apparent accuracy of the method. The authors should discuss these limitations and how they might affect their interpretation of CellNavi's performance.

4. Addressing Temporal Dynamics and Stochasticity: CellNavi primarily focuses on predicting driver genes based on static snapshots of cell states. However, cell state transitions are dynamic processes often influenced by intrinsic noise and stochasticity. How might these temporal aspects and inherent randomness affect the model's performance? The authors should discuss these limitations and potential future directions for incorporating temporal information and stochasticity into the CellNavi framework.

5. Evaluation with Synthetic Data: While the CRISPR data provides some level of ground truth, a more rigorous evaluation using synthetic data generated from a known mathematical model would greatly benefit the manuscript. Simulations based on stochastic dynamical systems models, where the true driver genes are predefined, would allow for a precise quantification of CellNavi's accuracy. This approach would also enable systematic exploration of the model's performance

under different conditions (such as varying the level of noise) and parameter settings, offering valuable insights into its strengths and limitations. This would address the limitations of real CRISPR data, such as incomplete perturbation coverage and off-target effects.

This manuscript presents a promising new tool with clear potential for advancing the study of cell state transitions. Addressing the points above would significantly strengthen the work and increase confidence in the capabilities of the CellNavi framework.

Reviewer #3 (Remarks on code availability):

I will review the code on the next iteration of the manuscript

METHODS – Nature Cell Biology publishes methods online. The methods section should be provided as a separate Word

document, which will be copyedited and appended to the manuscript PDF, and incorporated within the HTML format of the paper.

Methods should be written concisely, but should contain all elements necessary to allow interpretation and replication of the results. As a guideline, Methods sections typically do not exceed 3,000 words. The Methods should be divided into subsections listing reagents and techniques. When citing previous methods, accurate references should be provided and any alterations should be noted. Information must be provided about: antibody dilutions, company names, catalogue numbers and clone numbers for monoclonal antibodies; sequences of RNAi and cDNA probes/primers or company names and catalogue numbers if reagents are commercial; cell line names, sources and information on cell line identity and authentication. Animal studies and experiments involving human subjects must be reported in detail, identifying the committees approving the protocols. For studies involving human subjects/samples, a statement must be included confirming that informed consent was obtained. Statistical analyses and information on the reproducibility of experimental results should be provided in a section titled "Statistics and Reproducibility".

All Nature Cell Biology manuscripts submitted on or after March 21 2016 must include a Data availability statement as a separate section after Methods but before references, under the heading "Data Availability". For Springer Nature policies on data availability see <http://www.nature.com/authors/policies/availability.html>; for more information on this particular policy see <http://www.nature.com/authors/policies/data/data-availability-statements-data-citations.pdf>. The Data availability statement should include:

- Accession codes for primary datasets (generated during the study under consideration and designated as "primary accessions") and secondary datasets (published datasets reanalysed during the study under consideration, designated as "referenced accessions"). For primary accessions data should be made public to coincide with publication of the manuscript. A list of data types for which submission to community-endorsed public repositories is mandated (including sequence, structure, microarray, deep sequencing data) can be found here <http://www.nature.com/authors/policies/availability.html#data>.
- Unique identifiers (accession codes, DOIs or other unique persistent identifier) and hyperlinks for datasets deposited in an approved repository, but for which data deposition is not mandated (see here for details <http://www.nature.com/sdata/data-policies/repositories>).
- At a minimum, please include a statement confirming that all relevant data are available from the authors, and/or are included with the manuscript (e.g. as source data or supplementary information), listing which data are included (e.g. by figure panels and data types) and mentioning any restrictions on availability.
- If a dataset has a Digital Object Identifier (DOI) as its unique identifier, we strongly encourage including this in the Reference list and citing the dataset in the Methods.

We recommend that you upload the step-by-step protocols used in this manuscript to protocols.io. More details can be found at <https://www.protocols.io/help/publish-articles>.

All imaging data should be accompanied by scale bars, which should be defined in the legend. Cropped images of gels/blots are acceptable, but need to be accompanied by size markers, and to retain visible background signal within the linear range (i.e. should not be saturated). The boundaries of panels with low background have to be demarked with black lines. Splicing of panels should only be considered if unavoidable, and must be clearly marked on the figure, and noted in the legend with a statement on whether the samples were obtained and processed simultaneously. Quantitative comparisons between samples on different gels/blots are discouraged; if this is unavoidable, it should only be performed for samples derived from the same experiment with gels/blots were processed in parallel, which needs to be stated in the legend.

- For line art, graphs, charts and schematics we prefer Adobe Illustrator (.AI), Encapsulated PostScript (.EPS) or Portable

Document Format (.PDF). Files should be saved or exported as such directly from the application in which they were made, to allow us to restyle them according to our journal house style.

The total number of Supplementary Figures (not including the "unprocessed scans" Supplementary Figure) should not exceed the number of main display items (figures and/or tables (see our Guide to Authors and March 2012 editorial <http://www.nature.com/ncb/authors/submit/index.html#suppinfo>; <http://www.nature.com/ncb/journal/v14/n3/index.html#ed>). No restrictions apply to Supplementary Tables or Videos, but we advise authors to be selective in including supplemental data.

GUIDELINES FOR EXPERIMENTAL AND STATISTICAL REPORTING

REPORTING REQUIREMENTS – We are trying to improve the quality of methods and statistics reporting in our papers. To that end, we are now asking authors to complete a reporting summary that collects information on experimental design and

reagents. The Reporting Summary can be found here <https://www.nature.com/documents/nr-reporting-summary.pdf>. If you would like to reference the guidance text as you complete the template, please access these flattened versions at <http://www.nature.com/authors/policies/availability.html>.

----- Please don't hesitate to contact NCB@nature.com should you have queries about any of the above requirements ----

Version 2:

Decision Letter:

Our ref: NCB-A55223B

22nd May 2025

Dear Dr. Deng,

Thank you for submitting your revised manuscript "Directing cellular transitions on gene graph-enhanced cell state manifold" (NCB-A55223B). It has now been seen by the original referees and their comments are below. The reviewers find that the paper has improved in revision, and therefore we'll be happy in principle to publish it in Nature Cell Biology, pending minor revisions to satisfy the referees' final requests and to comply with our editorial and formatting guidelines.

Thank you again for your interest in Nature Cell Biology Please do not hesitate to contact me if you have any questions.

Sincerely,

Angela R Parrish, PhD
Locum Senior Editor
Nature Cell Biology

Reviewer #1 (Remarks to the Author):

I am mostly happy with the current version of the manuscript and with the efforts of the authors to address my requests. I still have some requests for clarifications and suggestions to better balance some of the author's statements.

1. In line 532, authors state. "However, it is worth noting that our approach to constructing cell-type-specific gene graphs involves a simplifying assumption: edges are removed for genes with zero expression. This approach is based on the premise that non-expressed genes are unlikely to participate in meaningful regulatory or signaling interactions. Filtering out these genes reduces noise and allows the model to focus on biologically active relationships." Authors need to consider that zero entries might also be related to expression dropouts / lower sequencing depth of a particular cell. This aspect should be

also discussed in the text.

2. While results regarding interpretability are interesting, i.e. TFs have more attention, the authors do not really address the interpretability request in full. Authors should stress the explainability limitations of their work in the discussions, by for example adopting a more critical view in their discussion as presented in the reply letter (see below).

Current text in the discussion line 562 "This explicit focus on regulatory elements provides CellNavi with a distinct advantage, enabling it to better model complex biological processes, and highlights the value of graph-based learning in improving model interpretability and biological relevance".

Reply to my request: "We acknowledge that explainability remains a critical challenge in deep learning models, including CellNavi, while our current work focuses on benchmarking performance and demonstrating the utility of pathway-based graphs. This is an area for future exploration, where we aim to develop methods to better visualize and interpret the graph attention mechanisms and how they relate to pathway connectivity and driver gene predictions."

Reviewer #2 (Remarks to the Author):

I have carefully reviewed the revised manuscript "Directing Cellular Transitions on Gene Graph-Enhanced Cell State Manifold" by Wang et al., together with the authors' detailed point-by-point responses. I am pleased to see that the authors have comprehensively addressed all of my original concerns and substantially strengthened the work. In particular: comprehensive ablation studies quantifying the contributions of CMM, DGP, and various gene-graph configurations; a candid discussion of potential biases in graph construction with suggestions for confidence scoring and experimental validation; a clear rationale for the 2,048-dimensional manifold grounded in Nash embedding theory, precedent models, and GPU constraints; robust down-sampling experiments demonstrating alignment across 20× sequencing depth variation; trajectory-reconstruction analyses combining UMAP, Spearman pseudotime correlation, and clustering metrics; ablations showing that both pretraining and fine-tuning are essential—especially for out-of-domain generalization; rigorous cross-validation via five-fold cluster holdouts and reverse splits with expanded metrics; and full details on runtimes (pretraining: 16 days on 32× A100; fine-tuning: 14–40 h), inference speed (0.38 s/cell), and hardware specifications. These additions substantially enhance the manuscript's rigor, clarity, and reproducibility. I have no further comments and am happy to recommend publication.

Reviewer #3 (Remarks to the Author):

I believe the authors have addressed my main concerns comprehensively and adequately. I think the comparison with synthetic datasets produced by mathematical models may be a little tangential to their study and perhaps they can follow this up in future work.

Reviewer #3 (Remarks on code availability):

One small suggestion here is in the "About" section, I'd suggest the authors add some keywords to help the scientific community find their code.

Version 3:

Decision Letter:

Dear Dr Deng,

I am pleased to inform you that your manuscript, "CellNavi predicts genes directing cellular transitions by learning a gene graph-enhanced cell state manifold", has now been accepted for publication in Nature Cell Biology.

Please note that *Nature Cell Biology* is a Transformative Journal (TJ). Authors may publish their research with us through the traditional subscription access route or make their paper immediately open access through payment of an article-processing charge (APC). Authors will not be required to make a final decision about access to their article until it has been accepted. [Find out more about Transformative Journals](https://www.springernature.com/gp/open-research/transformative-journals)

Authors may need to take specific actions to achieve compliance with funder and institutional open access mandates. If your research is supported by a funder that requires immediate open access (e.g. according to [Plan S principles](https://www.springernature.com/gp/open-science/plan-s-compliance) or the [NIH public access policy](https://www.springernature.com/gp/open-science/us-federal-agency-compliance)) then you should select the gold OA route, and we will direct you to the compliant route where possible. Because authors warrant under our subscription licensing terms that they haven't committed to licensing any version of their article under a licence inconsistent with the terms of our agreement – including the applicable embargo period – publication under the subscription model isn't suitable for authors whose funders require no embargo.

If you have not already done so, we strongly recommend that you upload the step-by-step protocols used in this manuscript to protocols.io (<https://protocols.io>), an open online resource that allows researchers to share their detailed experimental know-how. All uploaded protocols are made freely available and are assigned DOIs for ease of citation. Protocols and Nature Portfolio journal papers in which they are used can be linked to one another, and this link is clearly and prominently visible in the online versions of both. Authors who performed the specific experiments can act as primary authors for the Protocol as they will be best placed to share the methodology details, but the Corresponding Author of the present research paper should be included as one of the authors. By uploading your Protocols onto protocols.io, you are enabling researchers to more readily reproduce or adapt the methodology you use, as well as increasing the visibility of your protocols and papers. You can also establish a dedicated workspace to collect your lab Protocols. Further information can be found at <https://www.protocols.io/help/publish-articles>.

Nature Cell Biology encourages authors presenting evidence for cell, biological, molecular, and genetic interactions to consider communicating these findings using Biofactoid (<https://biofactoid.org/>). This tool helps users share a searchable representation of interactions (e.g. binding, gene expression, post-translational modification) between genes, gene products, or chemicals. Information added to Biofactoid, with author attribution, is shared on social media and public databases, such as Pathway Commons, where it can be discovered and analyzed in the context of a large and growing corpus of knowledge.

With kind regards,

Angela R Parrish, PhD
Locum Senior Editor
Nature Cell Biology

** Visit the Springer Nature Editorial and Publishing website at http://editorial-jobs.springernature.com?utm_source=ejp_NCB_email&utm_medium=ejp_NCB_email&utm_campaign=ejp_NCB for more information about our career opportunities. If you have any questions please click [here](mailto:editorial.publishing.jobs@springernature.com).

Reviewer #1 (Remarks to the Author):

In this manuscript, authors propose a Graph-based transformer model leveraging pathway information to create: (1) cellular embeddings; and (2) prediction of driver genes/drug target from perturbation and drug treatment experiments. A novel aspect is the use of pathway information, which are used to create single cell level graphs. Benchmarking results showed very positive results in particular the evaluation of perturb-seq data (Fig, 2). An unclear aspect here is the impact of the network in the predictions, i.e. by use of ablation experiments, or the choice of casual network (NicheNet). One missing opportunity of CellNavi is the lack of explainability of its prediction, and how they relate to these causal networks. Another unclear aspect about CellNavi is how it can be used in real world problems. The method introduction suggests that this is a foundation model (trained on 40 millions of HCA cells); however individual case studies indicate that novel models are trained in every case. Moreover, trained models are mostly related to the predicted tasks. The need for related train data makes the method applicability very restrictive. We also had issues in running the software due to incomplete documentation.

Thank you for your detailed comments and suggestions. We appreciate the opportunity to address them and clarify our work.

In the revised manuscript, we have conducted **comprehensive comparisons using different graph configurations**, including graphs with varying connectivity, derived from different studies, and graphs featuring alternative link structures while maintaining similar overall topologies. Additionally, we discussed the operations performed on the graph (e.g., removing gene nodes with zero expression) and analyzed the role of the graph in model performance. These analyses are included in the revised manuscript, and we expect they provide robust support for our choice of graph. We acknowledge that explainability remains a critical challenge in deep learning models, with CellNavi as a non-exception, while our current work focuses on benchmarking and demonstrating the utility of pathway-based graphs. The explainability of our model is an area for future exploration, where we aim to develop methods to better visualize and interpret the graph attention mechanisms and how they relate to pathway connectivity and driver gene predictions.

In the discussion section, we demonstrate how the model can generalize to completely different contexts, while we recommend cautious use when applying it to domains with significant differences in biology or experimental conditions. Training on task-related data is suggested to ensure optimal performance, but we do not view this as a restrictive limitation. **The CRISPR data used for training in this study is relatively homogeneous, typically derived from controlled experiments with limited variability.** In contrast, the application data encompasses diverse biological conditions

with high heterogeneity, including various cell types, perturbation types, and sequencing platforms. However, we acknowledge that further data collection and model improvement are both needed to build a truly generalized model across broader biological contexts.

Driver gene prediction remains in its early stages, and we present one of the first studies to explore what deep learning models can achieve in this space. CellNavi demonstrates strong performance and is expected to further improve as more data becomes available (a feature of deep learning methods). As another important contribution, we compiled driver gene prediction benchmarks, which are essential to promote the development of deep learning methods in this area. We aim to inspire the community—both among computational and experimental experts—by defining a clear task and evaluation framework, as well as showcasing interesting applications with a robust tool. We hope our work serves as a solid starting point for further advancements in this field.

Finally, we have made substantial improvements to the documentation of the code and the application, including updated READMEs, detailed tutorials, and scripts for training and evaluation. We added detailed specifications on the software requirements to ensure smooth user experience.

We hope these clarifications and improvements address your concerns.

Major points

1. Authors should include ablation experiments (in experiments shown in Fig. 2) to highlight the importance of the pathway networks in the predictions. This could include the use of random networks, fully connected networks, or sparsified networks.

Reply: Thank you for your insightful comment. To evaluate the role of pathway networks in CellNavi's predictions, we introduced three new types of networks, as suggested, to assess the impact of connection distribution and graph connectivity on predictive performance: 1) **Fully Connected Graph:** Every two genes are connected by edges of equal weight, representing a maximally connected network. 2) **Sparsified Graphs:** We down-sampled the total number of edges to 1/10 and 1/20 of the original graph to evaluate the influence of reduced connectivity. 3) **Random Graphs:** These graphs preserved the number of nodes and certain structural characteristics of the original pathway network, including self-loops, with edges introduced probabilistically to maintain consistency.

Using the Schmidt et al. dataset, we evaluated these alternative networks, alongside the original NicheNet graph, on predictive accuracy, F1-score, AUROC, AUPRC, and Matthew Correlation Coefficient (MCC). As shown in Figure R1 and Table R1, which are also included in the revised manuscript (Table S6), the structured pathway network in NicheNet provided a significant performance advantage over random and sparsified

graphs. Interestingly, the fully connected graph performed worse compared to random and sparsified networks, suggesting that excessive connectivity may introduce noise, diluting biologically relevant interactions and reducing predictive power.

Figure R1. Top-1 accuracy, Top-5 accuracy, F1-score, AUROC, AUPRC, and Matthews Correlation Coefficient (MCC) for various graph configurations utilized in combination with the CellNavi framework, evaluated on the Schmidt dataset. For sparsified and random graphs, three independent graphs were generated for each configuration. Error bar: standard error.

	Top-1 accuracy	Top-5 accuracy	AUROC	AUPRC	F1 score	MCC
NicheNet	0.6206	0.7326	0.8745	0.4991	0.6154	0.6155
Fully connected graph	0.3467	0.4555	0.7484	0.2103	0.3034	0.3468
Random graph	0.3184	0.4435	0.7525	0.1827	0.3049	0.3147
1/20 sparsity	0.3806	0.5101	0.7794	0.2696	0.3667	0.3853
1/10 sparsity	0.3291	0.4664	0.7746	0.1736	0.3366	0.3358
1/10 sparsity	0.3338	0.4542	0.7721	0.2153	0.3193	0.3347
1/10 sparsity	0.3857	0.5045	0.7821	0.2394	0.3776	0.3857

	0.3248	0.4526	0.7690	0.2069	0.3010	0.3236
	0.3916	0.5010	0.7835	0.2483	0.3716	0.3893
Random graph	0.3546	0.4839	0.7633	0.2465	0.3511	0.3591
	0.3911	0.5287	0.7965	0.2619	0.3711	0.3893

Table R1. Predictive performance comparison of CellNavi using NicheNet and alternative graph configurations. The table presents a comparison of predictive performance for CellNavi using different gene graphs, including NicheNet and alternative configurations, on the Schmidt dataset. Performance metrics include Top-1 accuracy, Top-5 accuracy, area under the receiver operating characteristic curve (AUROC), area under the precision-recall curve (AUPRC), F1 score, and Matthews Correlation Coefficient (MCC).

These results show that the NicheNet graph enables superior accuracy and robustness compared to alternative graph structures, which further emphasizes the role of biologically meaningful connectivity in driver gene prediction.

With the outstanding performance by incorporating biological network information, we acknowledge that the detailed reasons behind the observed performance differences remain challenging to fully dissect. The completeness or accuracy of the pathway information encoded in the NicheNet graph is also worth further investigation. We will continue to explore these aspects in future studies, as they represent an important direction for improving the interpretability and generalizability of graph-based models.

We have now included a new section in the Results (**Evaluating model components and graph configurations**), which presents ablation experiments using different networks. In addition, we have added a corresponding paragraph in the Discussion to contextualize these findings. We believe these additions will better demonstrate the critical role of gene networks in improving the predictive accuracy of the model.

2. On the same topic, why is the NicheNet pathway adopted? How much would NicheNet be affected if other pathways are used instead?

Reply: We chose to adopt NicheNet because it integrates both gene regulatory relationships and cellular signaling pathways, providing a comprehensive biological context compared to methods that rely solely on transcription factor–target gene interactions.

To validate this point of view, we evaluated the impact of alternative gene networks constructed with GENIE3 [1], GRNBoost2 [2], and the recently developed single-cell-specific method RENG [3]. These methods are prevalent for constructing gene regulatory networks (GRNs) in various contexts.

We constructed GRNs using cells from the Schmidt dataset and the Norman dataset, respectively, maintaining the default parameters reported whenever applicable. Due to computational memory constraints, for GENIE3 and RENGE, we limited the analysis to the top 5,000 highly variable genes, which may have influenced the completeness of the inferred networks. Moreover, for RENGE—which is explicitly designed to infer GRNs using time-series scRNA-seq data—we adapted the method to work with static single-cell RNA-seq data. This adaptation was necessary since the Norman and Schmidt datasets lack time-series information, but it inherently removed key features of RENGE, which may have impacted RENGE’s performance in this context.

Our results demonstrate that NicheNet consistently outperforms these alternatives (**Table R2**). We attribute this to the fact that, while alternative methods focus exclusively on transcription factor–target gene relationships, **NicheNet integrates both gene regulatory and signaling pathway information**, providing a broader and more nuanced representation of cellular states. This comprehensive description is particularly advantageous for our task, which involves capturing **subtle cellular state changes** induced by genetic or chemical perturbations that target non-transcription factor genes.

	Top-1 acc	Top-5 acc	AUROC	AUPRC	F1 score	MCC
Schmidt Dataset						
NicheNet	0.6206	0.7326	0.8745	0.4991	0.6154	0.6155
GENIE3	0.4916	0.6149	0.8216	0.3642	0.5105	0.4917
GRNBoost2	0.4091	0.5201	0.8063	0.2850	0.3982	0.4111
RENGE	0.5600	0.6835	0.8443	0.4487	0.5800	0.5551
Norman Dataset						
NicheNet	0.4900	0.6330	0.8884	0.4540	0.4219	0.4816
GENIE3	0.1390	0.3819	0.7935	0.0941	0.1107	0.1271
GRNBoost2	0.2533	0.4495	0.8210	0.2012	0.2295	0.2399
RENGE	0.4184	0.5974	0.8532	0.3518	0.3733	0.4074

Table R2. Top-1 accuracy, Top-5 accuracy, F1-score, AUROC, AUPRC, and Matthews Correlation Coefficient (MCC) for various prior graphs utilized in combination with the CellNavi framework, evaluated on the Schmidt and Norman datasets.

In responding to this comment, we have added a new section in the Results (**Evaluating model components and graph configurations**) presenting experiments using diverse graphs. Additionally, we have included a corresponding discussion in the **Discussion** section to contextualize these findings. We believe these additions effectively highlight the crucial role of comprehensive gene networks in enhancing the predictive accuracy of the model.

References

1. Huynh-Thu, V. A., Irrthum, A., Wehenkel, L., & Geurts, P. Inferring regulatory networks from expression data using tree-based methods. *PLoS one*, 5(9), e12776 (2010).
2. Moerman, T., Aibar Santos, S., Bravo González-Blas, C. *et al.* GRNBoost2 and Arboreto: efficient and scalable inference of gene regulatory networks. *Bioinformatics*, 35(12), 2159-2161 (2019).
3. Ishikawa, M., Sugino, S., Masuda, Y. *et al.* RENGE infers gene regulatory networks using time-series single-cell RNA-seq data with CRISPR perturbations. *Commun Biol* **6**, 1290 (2023)

3. The procedure to obtain networks is quite simple, i.e. removing edges of genes with zero expression. While simplicity is not bad per se, authors could justify this choice.

Reply: Thank you for raising this important point. Our approach to obtaining networks by removing edges of genes with zero expression is grounded in solid biological rationale. Non-expressed genes are unlikely to participate in meaningful regulatory or signaling interactions, and filtering them out allows us to reduce noise and focus on biologically active relationships. This principle is commonly applied in single-cell pretrained models, where zero-expression genes are often excluded to enhance model interpretability and computational efficiency [1,2,3]. Similarly, recent work on cell-type-specific protein representation learning has adopted this strategy [4], further supporting its utility and relevance.

We acknowledge, however, that this is not the only method for constructing gene networks. For instance, single-cell gene regulatory network inference approaches [5] provide a more dynamic and cell-specific representation of regulatory relationships. These methods could potentially capture additional layers of biological complexity. In future work, we plan to compare our approach with such methods to examine their impact on predictive performance and biological insights.

To make sure that this point is explicitly discussed, we added the following discussion to our manuscript:

However, it is worth noting that our approach to constructing cell-type-specific gene graphs involves a simplifying assumption: edges are removed for genes with zero expression. This approach is based on the premise that non-expressed genes are unlikely to participate in meaningful regulatory or signaling interactions. Filtering out these genes reduces noise and allows the model to focus on biologically active relationships. This principle has been applied in prior single-cell foundation models and cell-type-specific protein representation studies. Still, rigorous evaluation should be conducted to determine whether the presented construction is the most optimal strategy. Comparing it to alternative approaches, such as constructing

gene regulatory networks at the single-cell level, will provide additional insights. Future work could explore how different strategies for cell-type-specific network construction impact predictive performance and biological insights.

Reference:

1. Theodoris, C.V., Xiao, L., Chopra, A. et al. Transfer learning enables predictions in network biology. *Nature* 618, 616–624 (2023). <https://doi.org/10.1038/s41586-023-06139-9>
2. Cui, H., Wang, C., Maan, H. et al. scGPT: toward building a foundation model for single-cell multi-omics using generative AI. *Nat Methods* 21, 1470–1480 (2024). <https://doi.org/10.1038/s41592-024-02201-0>
3. Hao, M., Gong, J., Zeng, X. et al. Large-scale foundation model on single-cell transcriptomics. *Nat Methods* 21, 1481–1491 (2024). <https://doi.org/10.1038/s41592-024-02305-7>
4. Li, M.M., Huang, Y., Sumathipala, M. et al. Contextual AI models for single-cell protein biology. *Nat Methods* 21, 1546–1557 (2024). <https://doi.org/10.1038/s41592-024-02341-3>.
5. Feng G., Qin, X., Zhang J. et al. CellPolaris: Decoding Cell Fate through Generalization Transfer Learning of Gene Regulatory Networks. *BioRxiv* (2023). <https://doi.org/10.1101/2023.09.25.559244>.

4. In Figure S1.1a, the scheme shows that down-sampling is performed on expression profiles for self-supervision. The same figure suggests that down sampling does not change how graphs are computed. What is the rationale behind this? Would this generate some information leakage as the graph will have information about zero/non-zero genes?

Reply: We apologize for the unclear presentation. We define three types, or levels, of graphs used in our model:

1. **Original gene graph:** This graph is curated from general gene-gene relationships without cell type specificity. It serves as the foundational prior. NicheNet is the original gene graph in our context.
2. **Cell-type-specific graph:** This graph is curated from the transcriptome of individual single cells on top of the original gene graph, incorporating cell type-specific information to better reflect the biological context.
3. **Sample-specific graphs:** These are subgraphs derived from the cell-type-specific graphs and are used exclusively during the pre-training stage when down-sampling is performed. The sample-specific graphs only include genes selected as input for each training sample. Only non-zero genes after down-sampling are retained as input. Therefore, sample-specific graphs are constructed with a reduced non-zero gene set, they do not introduce data leakage during training.

On the other hand, our primary goal is not to train a model capable of recovering all genes but rather to develop robust cell embeddings. During inference and the fine-tuning stage

for tasks such as driver gene prediction, we do not perform down-sampling. Instead, we input all non-zero gene expression values into the model and utilize the full cell-type-specific graphs. This ensures that the model fully leverages the complete biological context, enhancing its predictive accuracy and biological relevance.

The key point is that the model utilizes only a subgraph of the gene graph, containing nodes corresponding to the input genes, to guide its reconstruction task. To clarify this strategy, we have added the following description to the caption of Figure S1.1:

To be noted, only sub gene graphs with nodes in the input sample (all non-zero genes) are used for attention calculation.

The description in Methods is also updated accordingly:

During pre-training, when down-sampling is performed on single-cell transcriptomes, only the non-zero genes included as model input are retained to generate sample-specific graphs that guide the model's task.

5. CellNavi users would profit from an explanation of how the methods achieved the predictions. It would be interesting to relate how attention mechanisms support gene predictions in the context of the causal networks.

Reply: We appreciate your interest in understanding how CellNavi achieves its predictions, particularly through its attention mechanisms in the context of causal networks. To address this, we analyzed the role of the GeneGraph Attention Layer in CellNavi and its ability to leverage the prior gene-gene interaction graph to enhance prediction accuracy.

The GeneGraph Attention Layer, a key component of CellNavi's Cell Manifold Model (CMM), integrates prior knowledge of gene interactions into the Transformer architecture. To evaluate how it supports driver gene predictions, we conducted an analysis of the attention weights learned by the model. Specifically, we compared CellNavi, which incorporates the prior gene graph, to a model with a similar architecture excluding the graph. Our analysis focused on transcription factors (TFs), which are well-established regulators of cell state transitions. Across all combinations of attention heads and layers, CellNavi consistently assigned significantly higher attention weights to TFs compared to non-TFs (**Figure R2**, included in the revision as Figure S6.2). This is consistent with their central role in modulating cell identity and transitions. In contrast, the model without a graph rarely showed such preferences on TFs, highlighting the critical role of the causal network in guiding the attention mechanism toward biologically meaningful interactions.

Furthermore, we observed that the attention bias learned from the gene graph exhibited a substantially higher magnitude for TFs compared to non-TFs (**Figure R3**, see Figure S6.3). This behavior underscores the importance of incorporating graph-based information, as it allows CellNavi to focus on genes that are likely to drive cellular transitions rather than being influenced by noise or indirect relationships. These findings suggest that the GeneGraph Attention Layer not only enhances the interpretability of the predictions but also ensures that biologically relevant regulators, such as TFs, are appropriately prioritized.

To make these insights accessible to users, we have included additional visualizations and analyses in Supplementary Figures S6.2 and S6.3. The additional figures help illustrate that the attention mechanisms leverage the gene networks to improve the prediction of driver genes by systematically focusing on transcriptional regulators and other key genes.

We acknowledge that explainability remains a critical challenge in deep learning models, including CellNavi, while our current work focuses on benchmarking performance and demonstrating the utility of pathway-based graphs. This is an area for future exploration, where we aim to develop methods to better visualize and interpret the graph attention mechanisms and how they relate to pathway connectivity and driver gene predictions.

Figure R2 (Figure S6.2). Differential attention weights between TFs and non-TFs. The heatmap displays the contrast in attention weights directed towards transcription factors (TFs) versus non-transcription factors (non-TFs) when the prior gene-gene interaction graph is included (left) or excluded (right). The color gradient represents the magnitude of the difference in attention weights. In the presence of the gene-gene graph, 88 attention heads demonstrate a statistically significant bias towards TFs, while without the graph, 15 attention heads show this preference ($P < 0.05$, Wilcoxon rank sum test, with false discovery rate (FDR) adjustment using the Benjamini–Hochberg procedure).

Figure R3 (Figure S6.3). The attention bias learned from the gene graph exhibited a notably higher magnitude for TFs compared to non-TFs in distribution ($P < 0.05$, Wilcoxon rank sum test). To aid visualization, the sample sizes for both TFs and non-TFs have been equalized through random sampling.

6. Authors need to clarify how the model can be used in real word applications; and this should be clear from the method introduction. The current figure 1 suggests a foundation model approach, but case studies indicate the necessity of additional training for specific tasks.

Reply:

Thank you for your suggestion. To further clarify the application of the CellNavi model, we have made the following revision in our manuscript:

We demonstrate that CellNavi, fine-tuned with CRISPR screening datasets—typically conducted on cultured cells or homogeneous populations and focused on immediate genetic perturbations—can be extended to predict more complex transitions in heterogeneous tissues and primary cells (Fig. 1g and Fig. S1.5). By leveraging a biologically meaningful manifold, CellNavi generalizes knowledge gained from CRISPR screenings beyond their original scope, to cellular transitions that are otherwise challenging to investigate using traditional CRISPR methodologies. While CellNavi has shown promising performance in highly generalized scenarios (Discussion), its performance in specific applications may benefit from additional fine-tuning on relevant CRISPR datasets. Further iterations

incorporating expanded experimental data may enhance its generalizability and applicability across diverse biological settings, requiring little to no fine-tuning.

Figure S1.5. Principle of CellNavi application. The fine-tuning data (left) consists of CRISPR perturbation datasets with low heterogeneity, typically derived from controlled experiments with limited variability. The application data (right) encompasses diverse biological conditions with high heterogeneity, including different cell types, perturbation types, and sequencing platforms. The bidirectional arrow indicates the challenges posed by differences in data sources, sequencing techniques, and biological variability when generalizing the model from fine-tuning to real-world applications.

Also, we discussed the boundary of generalization and its limitation:

The current pipeline requires fine-tuning on single-cell CRISPR screen data relevant to the system of interest. While our proof-of-concept test involving HEK293FT and K562 cells demonstrated promising results (Figure 6), the extent to which CellNavi can generalize to entirely new cell types or experimental systems remains unclear. Addressing this limitation will require exploring the boundaries of generalization across more diverse contexts and quantifying the “distance” between cell types or experiments to determine when fine-tuning is necessary. A long-term goal is to develop a method that generalizes across cellular contexts with minimal experimental effort, reducing the reliance on fine-tuning datasets.

Below, we elaborate on our rationale:

1. Pretraining and Fine-Tuning Strategy

We employ a pretraining–fine-tuning strategy similar to other single-cell foundation models, such as Geneformer and scGPT. The pretrained model, through self-supervised learning, captures cell representations and cell-type-specific gene expression patterns, enabling direct application to tasks like cell type annotation without additional fine-tuning.

However, for more complex tasks, such as *in silico* perturbation and driver gene prediction, the model has not encountered such tasks during pretraining and therefore cannot be directly applied. In these cases, at least a Multi-Layer Perceptron (MLP) and its fine-tuning on small-scale task-specific data is necessary to achieve meaningful predictions.

In CellNavi, we distinguish between:

- The Cell Manifold Model, which serves as a foundation model pretrained on large-scale single-cell RNA-seq data, capturing manifold representations of transcriptomic landscapes.
- The Driver Gene Predictor, a task-specific module designed for driver gene prediction, which is trained only during the fine-tuning stage using labeled data that includes source–target cell pairs and perturbation information for supervised learning.

By first pretraining on large-scale single-cell RNA-seq datasets, the model learns a broad transcriptional landscape. We then fine-tune it on genetic perturbation datasets (e.g., Perturb-seq) to align it with specific downstream tasks, such as identifying key driver genes in cell state transitions. This approach balances generalization from foundation modeling with task-specific adaptability via fine-tuning.

2. Application to Real-World Scenarios

Ideally, if large-scale perturbation datasets existed with comparable scale and coverage to those used in pretraining, we could fine-tune a generalized version of CellNavi, eliminating the need for additional fine-tuning by users. However, due to the scarcity and variable quality of perturbation datasets, training a universally generalizable model remains challenging.

Neural networks learn from the data distribution they are trained on and have limited extrapolation ability beyond this scope. As discussed in the Discussion section, assessing the generalization capability of the model remains difficult given current data limitations. At this stage, we believe it is premature to claim that CellNavi generalizes across all possible scenarios.

However, as demonstrated in our quantitative assessments and case studies, our method successfully transfers between different cell states, meaning that a model trained on one condition can generalize to another across cell types, sequencing techniques, and perturbation types. This suggests that CellNavi captures fundamental transcriptomic relationships, allowing it to be reused across different biological contexts with minimal additional training.

We believe this represents an important first step toward a fully generalizable model for single-cell perturbation analysis. As more perturbation datasets become available and

model architectures continue to evolve, it will become increasingly feasible to develop a version of CellNavi that is directly applicable across diverse biological scenarios without fine-tuning. We anticipate that future iterations will require less manual adaptation, ultimately improving usability in real-world applications.

7. Expanding point 6, could it be that training the model is HCA data (mostly healthy samples) makes CellNavi not generalizable to other tasks as T cell activation (Fig. 3) or neurodegenerative pathogenesis (Fig. 4).

Reply: Some level of generalization ability is indeed inherently limited due to the nature of HCA data, which consists mostly of healthy samples and does not fully cover all biological contexts. However, the principle of a foundation model is to learn a latent space that captures intrinsic transcriptomic structures across diverse cell types, enabling interpolation between different cell states. Since HCA spans a broad range of cell types, it provides a comprehensive training corpus, supporting the learning of a latent space that helps generalize to diverse cell states and understand cell identity and state relationships.

The key missing generalization component is the ability to connect specific genes to cell state changes. While the foundation model captures global cell state manifolds, it does not inherently learn how genetic perturbations drive these transitions.

To bridge this gap, fine-tuning on CRISPR perturbation datasets is required. These datasets explicitly link genetic perturbations to resulting cell state changes, enabling the model to learn causal relationships between genes and cellular transitions. This step is essential for accurate driver gene predictions, especially in disease-related scenarios, where cell states are more dynamic and context-dependent.

As mentioned in our response to the previous question (#7), the limited availability of CRISPR perturbation data constrains generalization ability. To further improve CellNavi's performance in real-world applications, integrating more diverse perturbation datasets will be critical. Expanding these datasets will reduce the need for users' fine-tuning by allowing the model to better learn gene-cell transition patterns across a broader range of biological conditions and the associated driver genes.

8. A recent benchmarking indicates that linear models outperforms models as GEARS: <https://www.biorxiv.org/content/10.1101/2024.09.16.613342v4> . Authors should improve the benchmarking of CellNavi (Fig. 2) to consider linear models; and additional data sets as done by Ahlmann-Eltze et al.

Reply: Thank you for bringing this paper to our attention. Overcomplicating models is indeed a common issue in omics and cell modeling. We appreciate the opportunity to clarify the differences in our benchmarking approach.

We noticed two differences in the setting between our task and the task presented in the paper by Ahlmann-Eltze et al.. First, we are **predicting genes that drive changes** in transcriptomes, instead of the outcome upon perturbation, so a no-change baseline is not applicable. Second, our goal is to develop a method that applies to **unseen cell states (cross-cell-state prediction)**, rather than unseen genes. These differences demand adaptations to baseline methods, and indicate that the outcome is likely to be different from the proposed paper.

1. Additive Baseline (for double perturbation)

For the **additive baseline**, we conducted the following experiment:

- For each perturbation pair, we simulated the double perturbation outcome as $y_a + y_b - 2c$, by randomly sampling 100 combinations of y_a , y_b , and c independently, where y_a and y_b represent the transcriptomes of gene g_a and g_b perturbations, and c is the control, and calculating the average.
- This generated a matrix $G \times P_{pair}$, where each row represents a gene g , each column represents a perturbation pair p_{pair} , and each cell in the matrix represents the expression value of gene g upon perturbation of p_{pair} .
- For each test sample (a pair of control and perturbed cells), we calculated the difference $y_{test} - c_{test}$, computed the cosine similarity between this difference and each column in $G \times P_{pair}$ and ranked the perturbation pairs.

Using the **Norman dataset**, we applied a train-test split where single-perturbations from all but one cluster, stratified using Leiden algorithm, were used for training (the same as in CellNavi manuscript) and double-perturbations were used for testing. We found that: 1) For unseen cell states (not included in the training data), the additive model has no prediction power, while CellNavi provides good predictions; and 2) on seen cell states, the additive model also performed poorly compared to CellNavi (**Table R3**). These results indicate that the additive model is not a viable alternative for the driver gene prediction tasks.

	Top-2 accuracy	Top-5 accuracy	Top-10 accuracy	F1 score	AUROC	AUPRC
In domain (cell state seen)						
Additive model	0.0288	0.1550	0.1969	0.0893	0.5063	0.0090
CellNavi	0.2094	0.3627	0.4881	0.4100	0.8100	0.3938
Out of domain (cell state unseen)						
Additive model	0.0000	0.0039	0.0039	0.0129	0.5412	0.0098
CellNavi	0.1358	0.2825	0.3760	0.3520	0.7916	0.3078

Table R3. Comparison of predictive performance between the additive model and CellNavi for double perturbation prediction. The table summarizes the predictive performance of the additive model and CellNavi in two scenarios: (1) **in-domain**, where the model is tested on double perturbations of cell states seen during training, and (2) **out-of-domain**, where the model is tested on double perturbations of cell states unseen during training. Metrics reported include Top-2, Top-5, and Top-10 accuracy, F1 score, area under the receiver operating characteristic curve (AUROC), and area under the precision-recall curve (AUPRC).

2. Matrix Optimization Baseline

Next, we attempted the **matrix optimization approach** used for single gene perturbations. Since our goal is to develop a method that applies to **unseen cell states**, rather than unseen genes, this approach necessitates some adjustments:

- For the training dataset, we calculated gene expression changes and performed pseudobulking (the same as in the reference paper), resulting in a matrix of $\Delta G \times P$.
- For testing, we computed the cosine similarity between $y_{test} - c_{test}$ and all columns in $\Delta G \times P$, then ranked the perturbation pairs by similarity.

Again, using the Norman dataset, we applied the train-test split with one cluster held out as test (the same as in CellNavi manuscript). Matrix optimization approach performed poorly in predicting single driver genes in unseen cell states (**Table R4**).

	Top-1 accuracy	Top-5 accuracy	Top-10 accuracy	F1 score	AUROC	AUPRC
Out of domain (cell state unseen)						
Matrix optimization	0.0019	0.0871	0.1657	0.2326	0.5230	0.0114
CellNavi	0.4900	0.4219	0.8884	0.4540	0.4900	0.4219

Table R4. Comparison of predictive performance between the matrix optimization approach and CellNavi for single perturbation prediction. out-of-domain: the model is tested on single perturbations of cell states unseen during training. Metrics reported include Top-1, Top-5, and Top-10 accuracy, F1 score, area under the receiver operating characteristic curve (AUROC), and area under the precision-recall curve (AUPRC).

3. Logistic Regression Baseline

A possible reason for the superior performance of our method is that **perturbation responses are cell-specific as implemented in CellNavi model, while the baseline methods assume that perturbation results are the same across cell types**. This makes the comparison unfair.

To make a fair comparison, we also explored a linear approach from another paper: <https://www.biorxiv.org/content/10.1101/2023.10.19.563100v1.full.pdf>, referenced in the paper you mentioned. This study demonstrates that L1-regularized logistic regression (LR) outperforms or is comparable to scBERT and scGPT, both of which are single-cell

foundation models, on cell type annotation tasks. Considering that our task is more akin to cell type annotation, where transcriptomes are used for classification and prioritization of discrete labels, we next adapted the LR approach for driver gene prediction.

To run this experiment, we used the following methodology:

For a pair of control and perturbed cells, we calculate the change in expression for each gene, concatenate them into a vector with a length of G , and input it into an L1-regularized logistic regression classifier (implemented with scikit-learn). We define the task as a multi-class classification problem, where the cross-entropy loss is used as the loss function.

We tested the LR model under two settings: 1) Out of domain: Same to the matrix optimization baseline, the model was tasked with predicting single driver genes in unseen cell states. 2) Random Splitting: Perturbation data were randomly split, with the test set containing the same number of cells as in the unseen cell state setting for a direct comparison.

The results (**Table R5**) demonstrate several key findings. First, linear models showed a significant improvement compared to the matrix optimization approach. This suggests that training-based models are more effective than similarity-based methods for the driver gene prediction task, as they can better leverage the training data to identify underlying patterns. Second, the random splitting setting yielded better performance than the OOD setting. Although this improvement is not rigorously quantified, it highlights that OOD generalization remains a major challenge for LR models, which are unable to handle the complexity of unseen cell states effectively. Third, in both settings, LR performed substantially worse than CellNavi. This indicates that linear models such as logistic regression are fundamentally limited in their ability to capture the intricate, cell-specific perturbation responses required for robust driver gene prediction.

	Top-1 accuracy	F1 score	AUROC	AUPRC
Out of domain (cell state unseen)				
Linear regression	0.2078	0.1711	0.7169	0.1736
CellNavi	0.4900	0.4219	0.8884	0.4540
Random split				
Linear regression	0.3085	0.2592	0.8024	0.2604

Table R5. Comparison of predictive performance between the linear regression approach and CellNavi for single perturbation prediction. Out-of-domain: The model is tested on single perturbations from cell states unseen during training. Random split: The test set is randomly selected from the perturbation dataset. Metrics reported include Top-1 accuracy, F1 score, area under the receiver operating characteristic curve (AUROC), and area under the precision-recall curve (AUPRC).

These limitations are notable even in the relatively homogeneous K562 cell line, where minimal variability in cell states is expected. This result suggests that LR's performance

would likely degrade further when applied to primary cells, where cellular heterogeneity is significantly higher. This observation is consistent with Figure 2C of the referenced paper, which highlights that while embeddings generated by foundation models are useful, they are insufficient for addressing more complex tasks, such as cross-cell state generalization. Together, these findings reinforce the necessity of more sophisticated approaches, such as CellNavi, to achieve accurate predictions in biologically diverse settings.

4. Conclusions

The experiments conducted in this study demonstrate that both model-free and linear approaches performed poorly in the task of cross-cell-state driver gene prediction. Two primary reasons underlie these results.

First, driver gene prediction differs fundamentally from perturbation response prediction. While perturbation response prediction often involves regression or continuous numeric outputs, driver gene prediction is inherently a classification task. In this context, consistent predictions across cell states provide limited utility, as the task requires identifying specific driver genes within a complex and cell-specific biological landscape.

Second, the task of cross-cell-state prediction introduces additional challenges that simple approaches fail to address. These challenges arise from the need to generalize across diverse cellular contexts, where perturbation responses are highly cell-specific. Embeddings learned by foundation models, such as those used in CellNavi, provide a potential advantage by capturing complex, non-linear relationships and cell-specific perturbation effects, enabling more robust predictions in these challenging scenarios.

The experimental results and additional discussions supporting these conclusions have been included in the Quantitative Assessment Section (highlighted in red) and the **Supplementary Note 3**.

9. Case studies shown in figures 3, 4 and 5 do not include competing methods; and are based on anecdotal evaluation. Authors could shorten this part, and focus on more benchmarking evaluation (see points above).

Reply: Thank you for your comments and suggestions. To address your concerns, we have made substantial efforts to bolster the benchmarking evaluation and provide more rigorous quantitative analyses throughout the manuscript.

As part of our revision, we conducted **extensive benchmarking comparisons with several alternative baseline methods**, including graph inference algorithms such as GENIE3, GRNBoost2, and RENGE, as well as linear models (Table S1-S3 in the revised

manuscript). These comparisons revealed that the alternative methods consistently underperform on our task.

In addition, we performed **ablation studies on key components of the CellNavi pipeline**, including the Cell State Manifold (CMM) module, the Driver Gene Predictor module, and the prior gene interaction graph (Figure S2.6 and Table S6 in the revised manuscript). To evaluate the impact of graph selection, we systematically replaced the NicheNet graph with alternative configurations, such as GRNs derived from GENIE3, GRNBoost2, or RENGE, fully connected graphs, sparsified graphs, and random graphs. In all cases, these substitutions led to a decline in performance.

Beyond these analyses, we defined quantitative metrics to demonstrate that the **CMM effectively learns meaningful cell representations** evaluation (Figure S1.2-S1.3 in the revised manuscript). Furthermore, we conducted **cross-validation** on benchmark datasets to ensure robust and reliable evaluation (Table S4-S5 in the revised manuscript). Lastly, we also incorporated **additional quantitative metrics** to comprehensively evaluate our method and its performance.

As this task—predicting driver genes across cell states—is relatively new, with few established methods or standardized benchmarks, we had to adapt tools designed for related tasks, such as in silico perturbation, for comparison. These adaptations may explain the relatively lower performance of competing methods compared to CellNavi, which is specifically designed for this problem. Additionally, the lack of widely accepted metrics or datasets beyond CRISPR datasets makes it challenging to perform comprehensive quantitative evaluations.

Nonetheless, we believe that the most exciting applications of CellNavi extend beyond CRISPR datasets to areas such as target identification and drug mechanism-of-action (MoA) analysis, as summarized in the case studies related to Figure 3, 4, and 5. These are critical problems that lack solid ground truth datasets but are supported by indirect evidence and biological plausibility. By presenting these case studies, we aim to demonstrate the potential of CellNavi in diverse scenarios and inspire further research in this domain. While this work is the first of its kind and can certainly be improved, we hope it highlights the importance of this emerging problem and encourages the development of more robust methodologies and datasets for benchmarking in the future.

We have integrated all additional quantitative benchmarks, analyses, and ablation studies into the manuscript, presenting them prior to the case studies. We believe these revisions provide a better balance between rigorous benchmarking and exploratory applications.

10. The cell type annotation evaluation, which is reported in the discussion, is out of place. Authors should remove; move somewhere else, or contextualize the need for this.

Reply: The cell type annotation evaluation was originally included to demonstrate the effectiveness of the embeddings learned by our model. However, we agree that it may be out of place in the context of the discussion. To address this, we have replaced this evaluation with more relevant and straightforward experiments, as mentioned earlier:

1. We now demonstrate that the Cell Manifold Model (CMM) learns meaningful cell embeddings by showing that it not only aligns cells of the same type across varying sequencing depths but also reconstructs developmental trajectories from single-cell data (**Figure. S1.2 and Figure. S1.3** in the revised manuscript). These evaluations provide a clear and biologically relevant assessment of the CMM's ability to capture cellular relationships.
2. We performed ablations of both the pretraining and fine-tuning components, including the Cell State Manifold module and the Driver Gene Predictor module, to evaluate their respective contributions to the overall model performance (**Figure. S2.6**).

These revised experiments provide a more direct evaluation of the model's design and functionality, ensuring that the results align more closely with the study's primary objectives.

Minor points

Line 171 indicates “we stratified the cells into distinct states, selecting a specific cell cluster for testing while training on the remaining clusters (Fig. 2a).” This does not fit the figure legend or the figure itself. Please consider improving this.

Reply: Our intention was to use Figure 2a as a conceptual illustration applicable to all CRISPR datasets. It reflects a nature that perturbed cells from different perturbations often appeared mixed rather than clustering together as expected. However, the figure may be misleading, as its visual shape resembles the previous dataset, and it does not explicitly depict double perturbations. To address this, we have generated a new figure (**Supplementary Figure S2.5**) that more clearly illustrates the data split, including the handling of double perturbations. We appreciate your suggestion and have made this clarification in the revised manuscript.

Figure S2.5 UMAP visualization of the train-test data split for the Norman dataset. (a) Using the unsupervised Leiden algorithm, we stratified cells into distinct states (colored). (b) A specific cell cluster was selected as the test set, while the remaining clusters were used for training. To ensure a rigorous evaluation, all multi-gene perturbations were excluded from the training set. Consequently, the training data (light gray) consisted only of single-gene perturbations within certain clusters, while the test set can be split into 1) single-gene perturbations from the held-out cluster (light blue), 2) double-gene perturbations from the held-out cluster (dark blue), and 3) double-gene perturbations from the training clusters (dark gray). The results reported in the main text are all derived from the held-out cluster (blue cluster on the UMAP visualization). To further ensure fairness, the test cell cluster was shuffled (similar to cross-validation) to obtain more robust and unbiased results (Table S5).

Abstract mentions casual connections but this is not explained and further supported in the main manuscript.

Reply: Thank you for pointing this out. To be more precise, we have revised the wording from “causal” to “directional” in the abstract and main text. Additionally, we have elaborated on the meaning of “directional” (or “causal”) in the Results section of the revised manuscript, and these changes have been highlighted for clarity.

Reviewer #1 (Remarks on code availability):

1. The software description was sparse, which make its execution difficult. I run into a system-specific error (Error: ValueError: ProcessGroupNCCL is only supported with GPUs, no GPUs found). Authors need to clarify hardware requirements for CellNavi; and include more details for uses. See suggestions below.

Reply: Thank you for your valuable feedback. We sincerely apologize for the lack of clarity in our previous description, which led to difficulties in running CellNavi. To clarify: both training and inference in CellNavi are developed specifically for NVIDIA GPUs, requiring CUDA version 12.1 or higher and Python version 3.12 or above.

To assist users in verifying their setup, we have implemented CUDA availability checks in `start_train.py`. Users will see messages indicating whether CUDA is available, if no GPUs are detected, or if there's an error in CUDA installation, along with detailed error information.

Additionally, we have emphasized our environment requirements in the GitHub repository (<https://github.com/DLS5-Omics/CellNavi>) and added detailed descriptions of installation and usage. We also packaged CellNavi for easy installation via `pip`, which automatically installs the required versions of dependencies. We sincerely hope these improvements address your concerns and make CellNavi more accessible and user-friendly.

2. Publish on PyPI/Conda: Simplifying the installation process by publishing the software on PyPI or Conda will make it more user-friendly and accessible. A single-command installation (`pip install` or `conda install`) would significantly improve usability and adoption.

Reply: To address this, we have packaged CellNavi into a Python package that can be installed using `pip install ..`. This allows users to set up the environment with a single command, significantly simplifying the installation process. Detailed installation instructions, including the use of `pip`, can be found on the installation section of our GitHub repository (<https://github.com/DLS5-Omics/CellNavi>). The repository provides comprehensive guidance to ensure a smooth setup experience.

3. Create Documentation and using Google Colab: Providing detailed and interactive documentation would help users better understand and use the software. To make the tutorial more engaging and to ensure results can be reproduced effectively, I suggest creating an interactive notebook using Google Colab. This approach allows users to experiment with the software directly in their browser without needing to set up the environment locally, making it more accessible and user-friendly. See <https://github.com/snap-stanford/GEARS?tab=readme-ov-file>.

Reply: We completely agree that such a feature would significantly enhance user accessibility and convenience. However, due to the distributed framework on which CellNavi is built, configuring the required environment on Google Colab is currently not easily feasible. Distributed training often relies on specific hardware setups and environment variables that are difficult to replicate in a Colab environment. As an alternative to improve accessibility, we have taken the following steps:

- Packaged CellNavi for Easy Installation: We have packaged CellNavi as a Python package that can be installed via `pip`. This allows users to quickly set up the software by creating a new virtual environment (e.g., using Conda) and installing all required dependencies with a single command.
- Provided a Detailed Tutorial Notebook: In our `tutorials` directory, we have included `tutorial.ipynb`, a Jupyter notebook that guides users through running example datasets. After downloading the example data, users can execute this notebook step-by-step, making it easier to understand and use CellNavi.

We hope these efforts help bridge the gap and make CellNavi more accessible and user-friendly for researchers. We will continue to explore ways to further enhance accessibility, including potential solutions for interactive notebooks in the future.

4. The current script for launching training (`launch_train.sh`) is highly system-specific and introduces unnecessary complexity for general users. It assumes certain configurations (e.g., NVIDIA GPUs, distributed training) and lacks flexibility, which can hinder usability.

Reply: We acknowledge that our current implementation has limitations. Due to resource constraints, we currently only support training and inference on NVIDIA GPU architectures. While we are not able to support all different configurations at the moment, we are committed to continuous improvement and expansion of our support in the future.

To address the flexibility concerns you've raised, we'd like to highlight that we offer two distinct training methods:

1. Single GPU: Users have the option to either run the Python file `start_train.py` directly (as demonstrated in our `tutorial.ipynb`) or use `launch_train.sh` for training.
2. Distributed GPU: For distributed architecture, which requires reading environment parameters, training can be initiated using `launch_train.sh` from the command line.

We've provided detailed explanations of these options in our tutorial to guide users through the process.

Importantly, for both training methods, users can easily modify training parameters and file paths in the `config.json` file. This approach enhances user convenience and intuitiveness, allowing for customization without altering the core scripts.

5. Key variables like RANK, WORLD_SIZE, and distributed settings are not explained. To improve this add clear documentation or comments to explain the purpose and usage of the script.

Reply: We apologize for the lack of clarity. In a distributed system, these parameters play a critical role in enabling communication and coordination among processes:

1. RANK: Represents the global rank of the process within the distributed setup. Each process is assigned a unique rank, which helps differentiate its role in the training process.
2. LOCAL_RANK: Denotes the rank of the process on the local machine. This is particularly useful when multiple GPUs are used per node, as it allows processes to identify their specific GPU allocation.
3. WORLD_SIZE: Indicates the total number of processes in the distributed setup. This value is essential for ensuring that all processes are aware of the overall system configuration.
4. MASTER_ADDR: Specifies the IP address or hostname of the master node (rank 0). All worker processes connect to this address to synchronize and exchange information during training.
5. MASTER_PORT: Defines the port number on which the master node listens for incoming connections from worker processes. This port must be free and available on the master node to avoid conflicts.

It is important to note that if users run CellNavi on a single GPU, these parameters do not impact program execution. In such cases, they are pre-defined, allowing users to directly run the Python file `start_train.py`. However, when using a multi-node distributed GPU architecture, these parameters must be read from environment variables, which requires running `launch_train.sh`.

To make this clearer for users, we have provided detailed explanations of these parameters and their usage in our tutorial. The tutorial also includes step-by-step instructions for both single-GPU and distributed training setups.

6. GPU detection relies on `nvidia-smi`, which is specific to NVIDIA systems. It would be better to use a general detection method like `torch.cuda.device_count()` for broader compatibility

Reply: You are absolutely correct that relying on `nvidia-smi` limits compatibility to NVIDIA systems. While NVIDIA GPUs are the most widely used and were the focus of our implementation due to resource limitations, we have now incorporated a more general detection method using a try/catch block around the `torch.cuda.device_count()` function to check whether the running system has the required GPU and CUDA support. Additionally, we would like to clarify that our model requires GPU usage for training to ensure optimal performance. For testing, we have added support for CPU execution to enhance compatibility; however, we still recommend using a GPU for better efficiency and performance.

Reviewer #2 (Remarks to the Author):

This manuscript, titled "Directing Cellular Transitions on Gene Graph-Enhanced Cell State Manifold" by Tianze Wang et al., presents CellNavi, an innovative deep learning framework designed to predict key driver genes governing cellular state transitions. Recognizing the challenges in identifying these genes due to the complexity and scale of transcriptomic data, the authors designed CellNavi to integrate single-cell transcriptomics with causal gene graphs, creating a biologically meaningful cell state manifold. This manifold reduces high-dimensional data into a lower-dimensional representation that preserves intrinsic biological features and relationships. CellNavi comprises two core components: the Cell Manifold Model (CMM) and the Driver Gene Predictor (DGP). The CMM captures and encodes cell states using a graph-based Transformer architecture that incorporates gene interaction networks, enhancing the biological relevance of the manifold. The DGP leverages this manifold to identify key driver genes by analyzing transcriptomic changes associated with transitions between source and target cell states. Then, the authors rigorously evaluated CellNavi across a diverse range of tasks and datasets, including cellular differentiation, disease progression, and drug response. Through testing on CRISPR perturbation datasets and benchmarking against established methods such as SCENIC+ and GEARS, the manuscript demonstrates CellNavi's superior predictive performance and generalization capabilities. Notably, the model excels in identifying critical genetic regulators in contexts as varied as T cell differentiation and neurodegenerative disease progression, underscoring its broad applicability. Furthermore, the framework's ability to infer mechanisms of action for drug compounds highlights its potential utility in therapeutic innovation and drug discovery. While the study presents notable advancements, some aspects of this work should be clarified/improved, as detailed below.

Reply: We sincerely appreciate your positive and construction feedback. We have made significant efforts to address the comments and improve the manuscript accordingly. Below, we summarize the key updates and clarifications based on your suggestions:

- We have performed additional ablation studies to evaluate the critical role of components in our model, including the Cell Manifold Model (CMM), the Driver Gene Predictor (DGP), and the gene graphs. These studies highlight the unique contributions of the NicheNet gene graph and demonstrate how alternative graph configurations and sparsity impact model performance.
- We have performed further analysis on the CMM, particularly on its ability to capture meaningful cell state manifold through trajectory reconstruction and sequencing depth alignment tasks.

- To provide a more rigorous and comprehensive evaluation, we have included additional metrics, while implementing cross-validation strategies across different datasets.
- Additionally, we have expanded the discussion around the impact of noise on model performance, the potential bias in gene graphs, and the dimensionality of the cell manifold.
- Finally, we have included details on model training and computational efficiency.

While we have addressed the majority of the reviewer's comments, we acknowledge that some questions, such as the potential biases in graph construction and the optimal dimensionality of the cell manifold, remain open for further exploration. These topics represent exciting directions for future work, as they are central to advancing the interpretability and generalizability of cell representation models. Thank you again for your constructive feedback.

Major:

1. The integration of causal gene graphs is a key innovation in this study. To highlight their importance, the authors could conduct ablation studies to assess the impact of gene graphs on model performance. This would provide clearer insights into their critical role and validate their contribution.

Reply: We agree that ablation study is an important point to highlight the critical role of gene graph. In the original manuscript, we demonstrated that integrating a gene graph allows the model to assign higher attention bias and weights to transcription factors (TFs) compared to non-TFs, relative to a model without a gene graph (Figure S6.2–6.3). To further validate the importance of gene graphs, we conducted two additional ablation studies: (1) replacing NicheNet with gene graphs constructed using alternative methods, and (2) exploring the impact of graph properties, such as connectivity and structure, on predictive performance.

First, we examined the effects of alternative gene graphs generated using three widely used GRN inference methods: GENIE3 [1], GRNBoost2 [2], and RENGE [3], a single-cell-specific method. GRNs were constructed using cells from the Schmidt dataset, adhering to default parameters from prior studies where applicable. For GENIE3 and RENGE, computational memory constraints required us to limit the analysis to the top 5,000 highly variable genes, which may have impacted the completeness of the inferred networks. Additionally, RENGE, designed for time-series scRNA-seq data, was adapted to static single-cell data since the Schmidt dataset lacks temporal information.

We evaluated these alternative networks, alongside the original NicheNet graph, on predictive accuracy, F1-score, AUROC, AUPRC, and Matthew Correlation Coefficient (MCC). The result demonstrates that NicheNet consistently outperforms these

alternatives (**Table R1**). We attribute this to the fact that, while alternative methods focus exclusively on transcription factor–target gene relationships, **NicheNet integrates both gene regulatory and signaling pathway information**, providing a broader and more nuanced representation of cellular states. This comprehensive description is particularly advantageous for our task, which involves capturing **subtle cellular state changes** induced by genetic or chemical perturbations that target non-transcription factor genes. Adaptation of the GRN methods may have also compromised their performance.

Second, we examined the impact of graph properties and connectivity on predictive performance by testing the following configurations: 1) **Fully Connected Graph**: Every two genes are connected by edges of equal weight, representing a maximally connected network. This is also an equivalent to no prior graphs. 2) **Sparsified Graphs**: We down-sampled the total number of edges to 1/10 and 1/20 of the original graph to evaluate the influence of reduced connectivity. 3) **Random Graphs**: These graphs preserved the number of nodes and certain structural characteristics of the original network, including self-loops, with edges introduced probabilistically to maintain consistency.

As shown in **Table R1**, the structured gene graphs, such as NicheNet and other GRNs, consistently outperformed random graphs or graphs with different connectivity. Collectively, these results emphasize the importance of using a biologically meaningful and integrative gene graph, such as NicheNet, to ensure predictive robustness and accuracy.

	Top-1 accuracy	Top-5 accuracy	AUROC	AUPRC	F1 score	MCC
NicheNet	0.6206	0.7326	0.8745	0.4991	0.6154	0.6155
GENIE3	0.4916	0.6149	0.8216	0.3642	0.5105	0.4917
GRNBoost2	0.4091	0.5201	0.8063	0.2850	0.3982	0.4111
RENGE	0.5600	0.6835	0.8443	0.4487	0.5800	0.5551
Fully connected graph	0.3467	0.4555	0.7484	0.2103	0.3034	0.3468
	0.3184	0.4435	0.7525	0.1827	0.3049	0.3147
1/20 sparsity	0.3806	0.5101	0.7794	0.2696	0.3667	0.3853
	0.3291	0.4664	0.7746	0.1736	0.3366	0.3358
	0.3338	0.4542	0.7721	0.2153	0.3193	0.3347
1/10 sparsity	0.3857	0.5045	0.7821	0.2394	0.3776	0.3857
	0.3248	0.4526	0.7690	0.2069	0.3010	0.3236

	0.3916	0.5010	0.7835	0.2483	0.3716	0.3893
Random graph	0.3546	0.4839	0.7633	0.2465	0.3511	0.3591
	0.3911	0.5287	0.7965	0.2619	0.3711	0.3893

Table R1. Predictive performance comparison of CellNavi using NicheNet and alternative graph configurations. The table presents a comparison of predictive performance for CellNavi using different gene graphs, including NicheNet and alternative configurations, on the Schmidt dataset. Performance metrics include Top-1 accuracy, Top-5 accuracy, area under the receiver operating characteristic curve (AUROC), area under the precision-recall curve (AUPRC), F1 score, and Matthews Correlation Coefficient (MCC). For sparsified and random graphs, three independent graphs were generated for each configuration.

However, the detailed reasons behind the observed performance differences between graph types remain challenging to fully dissect. The completeness or accuracy of the pathway information encoded in the NicheNet graph is also worth investigation. We will continue to explore these aspects in future studies, as they represent important directions for improving the interpretability and generalizability of cell representation models.

We have now included a new section in the Results (**Evaluating model components and graph configurations**), which presents ablation experiments using diverse graphs. In addition, we have added a corresponding paragraph in the Discussion to contextualize these findings. We believe these additions will better demonstrate the critical role of gene graphs in improving the predictive accuracy of the model.

References

1. Huynh-Thu, V. A., Irrthum, A., Wehenkel, L., & Geurts, P. Inferring regulatory networks from expression data using tree-based methods. *PLoS one*, 5(9), e12776 (2010).
2. Moerman, T., Aibar Santos, S., Bravo González-Blas, C. *et al.* GRNBoost2 and Arboreto: efficient and scalable inference of gene regulatory networks. *Bioinformatics*, 35(12), 2159-2161 (2019).
3. Ishikawa, M., Sugino, S., Masuda, Y. *et al.* RENGE infers gene regulatory networks using time-series single-cell RNA-seq data with CRISPR perturbations. *Commun Biol* 6, 1290 (2023)

2. The quality and completeness of the causal gene graphs likely impact performance. How do the authors address potential biases in model predictions, especially considering that gene graphs might encode biased relationships?

Reply: We acknowledge that quality, completeness, and potential biases in gene graphs could influence model performance. Below is our understanding and practice in treating potentially biased relationship.

It is important to note that gene graphs are not intrinsically observable given current experimental methods. They are typically constructed using a combination of high-

throughput datasets, curated knowledgebases, and computational predictions. Overrepresentation of well-studied pathways, transcription factors, or specific biological contexts, as well as incomplete coverage of less-characterized regions of the interactome may introduce quality concerns. Also, all methods for constructing gene graphs inherently rely on simplifying assumptions, which can introduce biases. For instance, in our cell-type-specific gene graphs, we adopt the simplifying assumption that edges are removed for genes with zero expression. While this approach has shown empirical utility, it requires further validation to ensure its biological accuracy. Evaluating the quality of such graphs is intrinsically challenging, especially for context-specific graphs. Validation of individual gene-gene connections using experimental methods such as FRET or other molecular assays represents one possible approach. However, due to the scale of these graphs, it is not yet feasible to systematically validate all connections experimentally.

One potential strategy for addressing these challenges in the future is to combine experimental validation with computational gene graph construction methods that provide confidence scores for each connection. This would allow for the prioritization of high-confidence interactions while also identifying areas of the graph that require further refinement. Nevertheless, this is a complex and long-term goal that lies beyond the scope of the current study.

At present, we rely on the best available gene graph, here specifically NicheNet, which has multiple types of gene/protein interaction information. This integration may help reduce biases associated with relying exclusively on transcriptional regulatory data. As shown in the ablation studies introduced in the response to question#1, we tested alternative graphs constructed using methods such as GENIE3, GRNBoost2, and RENGE, as well as simulated graphs with varying connectivity and structures. These experimental outcomes demonstrate that NicheNet consistently outperforms alternative graphs in predictive performance.

While NicheNet currently provides the best performance, we must acknowledge its limitations, such as potential biases in its underlying data sources and incompleteness in certain biological contexts. To address this, we have included a discussion paragraph in the revised manuscript to highlight these limitations and potential future directions:

However, it is worth noting that our approach to constructing cell-type-specific gene graphs involves a simplifying assumption: edges are removed for genes with zero expression. This approach is based on the premise that non-expressed genes are unlikely to participate in meaningful regulatory or signaling interactions. Filtering out these genes reduces noise and allows the model to focus on biologically active relationships. This principle has been applied in prior single-cell foundation models and cell-type-specific protein representation studies. Still, rigorous evaluation

should be conducted to determine whether the presented construction is the most optimal strategy. Comparing it to alternative approaches, such as constructing gene regulatory networks at the single-cell level, will provide additional insights. Future work could explore how different strategies for cell-type-specific network construction impact predictive performance and biological insights.

3. The construction of the cell state manifold is central to the model. The exact method for determining the dimensionality of the cell manifold is unclear. What criteria are used to determine the optimal dimensionality of the manifold?

Reply: This is indeed a compelling topic. In our study, we utilized a 2,048-dimensional coordinate space to represent the manifold of cell states. This dimensionality was chosen based on a combination of theoretical considerations and prior literature, and we presume that the precise dimensionality is not critical for constructing the cell state manifold itself.

The primary desiderata for constructing the manifold are (1) lower dimensionality than the original gene expression space and (2) biological interpretability of distances in the vector space. Lower dimensionality ensures the removal of redundant information, as the intrinsic dimensionality of cellular states is significantly lower than the ~20,000-dimensional gene expression space due to biological constraints and gene co-regulation. Biological interpretability ensures that distances in the manifold correspond to meaningful differences between cell states. For the first criterion, it is sufficient to select a dimensionality smaller than the original space, as the intrinsic dimensionality of the manifold is constrained by inter-gene relationships. For the second criterion, theoretical results such as the Nash embedding theorem (Nash, John (1954). "C1 isometric embeddings." *Annals of Mathematics*. 60(3): 383–396) demonstrate that any manifold of intrinsic dimensionality n can be faithfully embedded in a coordinate space of dimensionality no greater than $2n + 1$. Consequently, choosing a dimensionality larger than the intrinsic manifold dimensionality ensures that geometric properties are preserved. Furthermore, the use of down-sampling-reconstruction pretraining techniques in our model enforces biological meaning in the learned embeddings. Importantly, overestimating the dimensionality does not harm the representation, as redundant dimensions can correlate with the intrinsic ones, allowing for an isometric embedding without distorting biological information.

Previous single-cell foundation models, such as Geneformer, scGPT, and scFoundation, have successfully employed latent spaces of 256 or 512 dimensions to represent cell states across diverse biological systems. However, these models did not focus on recovering individual gene expression profiles from cell embeddings, a task central to our method. We hypothesize that a higher dimensionality is beneficial for achieving such

recovery, as it provides greater representational capacity. However, higher dimensionality may lead to GPU memory issue during training. The dimensionality of 2,048 was then chosen as a balance between computational efficiency and the ability to encode complex biological relationships.

We acknowledge that the choice of optimal dimensionality is an important question and that the manifold of cell states warrants further investigation. Due to resource limitations, we did not conduct a systematic evaluation of different dimensionalities in this study. This remains an exciting avenue for future exploration. Importantly, our framework is flexible and allows the dimensionality to be adjusted based on the dataset and specific application. This adaptability ensures that our approach can accommodate diverse biological scenarios.

To summarize, while the intrinsic dimensionality of the cell state manifold is not directly observable, our choice of 2,048 dimensions provides sufficient flexibility and capacity to capture its structure without compromising biological interpretability. This decision reflects a balance between theoretical considerations, prior insights, and practical constraints. The discussion on dimensionality is now included in **Supplementary Note 1**.

4. The manuscript mentions using down-sampling as part of the pretraining process. How does this affect the fidelity of the learned manifold, and does it introduce biases in the representation?

Reply: Bias in cell representation learning can arise from various sources. For instance, highly abundant cell types may be overrepresented, while rare populations are underrepresented. This is a universal issue inherent to most single-cell data analyses, and there is limited scope to completely eliminate it. Nonetheless, we propose and demonstrate that the down sampling-reconstruction strategy is effective to mitigate the bias introduced by differences in sequencing depth and data sparsity.

The motivation for the down sampling-reconstruction strategy stems from challenges inherent in single-cell sequencing data, where genes with lower expression levels are more likely to be lost due to differences in sequencing depth across batches. These dropouts can exacerbate batch effects, causing cells from datasets with varying depths to cluster separately rather than reflecting true biological differences [1-2]. To address this, we simulate depth variation by selectively masking low-expression genes through a controlled down sampling of sequencing depth and tasked the model to reconstruct the original expression. This approach enables the model to learn representations that are robust to data sparsity and better generalize across experimental conditions and sequencing depths.

To validate this approach, we utilized a human pancreas scRNA-seq dataset [3] that was not included in the pretraining corpus of our model. After down-sampling the expression profile 20-fold, major cell types remained distinguishable; however, a severe batch effect was observed, as cells from the down-sampled dataset failed to integrate with the original profile using standard methods. In contrast, **the Cell Manifold Model (CMM) successfully integrated embeddings from the down-sampled and original profiles**, preserving biological structure across sequencing depths (**Figure R1**).

Figure R1 (Figure S1.2). CellNavi can align the same cell types across varying sequencing depths. Upper panel: After 20-fold down-sampling on the expression profile, major cell types remained distinguishable, but a severe batch effect was observed. Lower panel: the Cell Manifold Model (CMM) in CellNavi can integrate embeddings from the down-sampled and original profiles, while preserving the biological structure. Integration quality was quantitatively assessed using iLISI (1: worst, 2: best) and cLISI (1: best, 2: worst), which evaluate batch mixing and cell type separation, respectively.

Furthermore, as detailed in response to question #5, we examined the fidelity of the constructed manifold using a trajectory reconstruction task. Trajectory reconstruction serves as a robust evaluation of the manifold's ability to capture smooth, biologically meaningful changes in cellular states. As shown in the following figure (**Figure R2**), **the manifold successfully recapitulates the transitions from Ngn3-low progenitor cells to mature endocrine populations during pancreatic endocrinogenesis**. This result

demonstrates that the manifold constructed by CellNavi effectively captures both global and local cell-state structures. We believe these additional analyses highlight the advantages of the pretraining strategy employed by CellNavi, even though there is still room for improvement. To provide a clearer description of our methods, we have included these results in the revised manuscript (Figure S1.2-S1.3).

References

1. Pratapa, A., Jaliha, A. P., Law, J. N., Bharadwaj, A., & Murali, T. M. (2020). Benchmarking algorithms for gene regulatory network inference from single-cell transcriptomic data. *Nature Methods*, 17, 147–154.
2. Korsunsky, I., et al. (2019). Fast, sensitive and accurate integration of single-cell data with Harmony. *Nature Methods*, 16, 1289–1296.
3. Baron, M., et al. (2016). A single-cell transcriptomic map of the human and mouse pancreas reveals inter-and intra-cell population structure. *Cell systems*, 3(4), 346-360.

5. The authors highlight that the model constructs a lower-dimensional manifold for cell states, capturing their intrinsic features while preserving relative similarities between cells. Could the authors provide both quantitative and qualitative evidence to support this claim?

Reply: Thank you for your suggestion. As highlighted in our responses to the previous question (#4), we propose two tasks to evaluate whether the cell state manifold effectively captures intrinsic cellular features, thereby preserving relative similarities and meaningful relationships between cells within the manifold:

1) As discussed earlier, CellNavi can **align the same cell types across datasets with varying sequencing depths (Figure R1 with both quantitative and qualitative evidence)**. This suggests that the model captures intrinsic features of cells, focusing on gene-gene relationships rather than relying solely on absolute expression values. This ability to integrate profiles with differing depths provides strong evidence of the robustness of the cell manifold.

2) We also examined if the cell manifold can **reconstruct developmental trajectory**. This task is well-suited to evaluate the fidelity of the learned manifold, because the associated data inherently represents continuous transitions between cell states, providing a natural benchmark for assessing whether the model captures both global progressions and local structure. In other words, unlike static or highly discrete datasets, trajectory datasets offer an opportunity to evaluate how well the manifold reflects smooth, biologically meaningful changes in cellular states.

To this end, we conducted analyses using a pancreatic endocrinogenesis dataset from Bastidas-Ponce et al. [1]. This dataset contains continuous developmental trajectories and are thus ideal for assessing the biological relevance of the learned embedding space.

For qualitative assessment, we visualized the cell representations obtained from the pretrained Cell Manifold Model (CMM) module within CellNavi using UMAP. The manifold reveals smooth transitions from Ngn3-low progenitor cells to mature endocrine populations, consistent with patterns observed in the gene expression UMAP and prior biological knowledge (**Figure R2a**).

Figure R2 (Figure S1.3). Trajectory reconstruction with the Cell Manifold Model. (a) UMAPs colored by cell type annotations (left) and diffusion pseudotime (DPT) (right, indicated by the colorbars), inferred from the original data (upper panel) and CellNavi-predicted data (lower panel). Cell type labels were assigned based on known markers, and pseudotime was computed using Scanpy's DPT implementation. (b) Spearman correlation between pseudotime inferred from the original data (x-axis) and from the CellNavi-predicted data (y-axis) with a quantification in c. (c) Summary of quantitative evaluation using Spearman correlation, along with clustering evaluation metrics including Adjusted Rand Index (ARI), Normalized Mutual Information (NMI), Average Silhouette Score (ASS), and Fowlkes-Mallows Score (FMS). Scores computed from random embeddings are included for comparison.

For quantitative assessment, we employed two complementary strategies to evaluate the quality of the learned manifold. First, we computed the Spearman correlation between diffusion pseudotime inferred from the original gene expression space and that obtained from the CellNavi embedding. This analysis assesses the preservation of global cell-state

progression, and our results show a high correlation, indicating that the model maintains the overall developmental structure (**Figure R2b-c**).

Second, we evaluated clustering consistency between the gene expression space and the CellNavi embedding. Using the Louvain algorithm with identical parameters in both spaces, we compared cluster assignments using Adjusted Rand Index (ARI), Normalized Mutual Information (NMI), Average Silhouette Score (ASS), and Fowlkes-Mallows Index (FMI) (**Figure R2c**). The clustering results in the CellNavi space were highly consistent with those from the original space, suggesting that the model preserves local similarity structure among cells. As a baseline, we compared against random embeddings and found that CellNavi performs substantially better across all metrics.

These results indicate that CellNavi effectively captures both global and local cell-state structure in a biologically meaningful low-dimensional manifold. However, we acknowledge that the performance could be further improved. Future work could investigate whether alternative strategies for pretraining or additional constraints on the manifold construction might enhance the interpretability and resolution of subtle cell-state transition.

References

1. Bastidas-Ponce, A., Tritschler, S., Dony, L., et al. Comprehensive single cell mRNA profiling reveals a detailed roadmap for pancreatic endocrinogenesis. *Development* 146.12 (2019): dev173849.

6. The CMM is pretrained with reconstruction tasks before joint training with the DGP. How do pretraining and fine-tuning contribute to final model performance? Is there a trade-off between these stages?

Reply: The roles of pretraining and fine-tuning in our framework are distinct and complementary, with no inherent trade-off between the two stages. They address different aspects of the modeling process, which together contribute to the final performance of the model.

Pretraining: Generalized Cell Representation Learning. Pretraining captures global cell state structures and relationships by learning a generalized cell manifold. This process encodes cell-cell relationships and creates a latent space that effectively represents cell identities and states. It enables the model to generalize across diverse biological contexts and perform downstream tasks, such as cell type annotation, without additional fine-tuning.

However, while pretraining provides a strong foundation by capturing global cell state manifolds, it does not inherently learn task-specific relationships, such as how genetic perturbations drive cell state transitions. These relationships are essential for tasks like

in silico perturbation and driver gene prediction, which go beyond the scope of pretraining. Pretraining, on the other hand, provides a good starting point for fine-tuning for specific applications.

Fine-Tuning: Task-Specific Adaptation. Fine-tuning is necessary to adapt the pretrained model to specific tasks. In the context of genetic perturbations, fine-tuning enables the model to learn causal relationships between genes and cellular transitions. By leveraging CRISPR perturbation datasets, fine-tuning focuses on connecting specific genes to cell state changes, which is critical for driver gene predictions and understanding disease-specific cell dynamics. Without fine-tuning, the pretrained model cannot accurately perform such complex tasks because these relationships are not part of the initial pretraining process.

Trade-Offs may present between fine-tuning tasks. While pretraining and fine-tuning are complementary, trade-offs may arise if the goal is to build a model that supports a wide range of tasks requiring fine-tuning. This is because different fine-tuning tasks may demand conflicting adaptations in the model. For instance, tasks focused on cell type annotation may require the model to emphasize global manifold structures, while tasks like driver gene prediction may demand more attention to causal gene-cell relationships. Managing these trade-offs would require careful task-specific fine-tuning strategies and possibly multi-task learning approaches.

Ablation Studies: Evaluating the Contribution of Pretraining (CMM) and Fine-Tuning (DGP). To evaluate the individual contributions of the pretraining and fine-tuning steps in CellNavi, we conducted ablation studies under two settings:

- **Ablation on CMM (Pretraining):** The pretraining step was removed, and the DGP was used directly. In this setup, gene name embeddings were skipped, and a one-dimensional gene expression vector was used as input, with each gene represented by its position.
- **Ablation on DGP (Fine-Tuning):** Since fine-tuning is essential for driver gene prediction, we replaced the DGP with a multinomial logistic regression model. This model ranked candidate driver genes based on the cell embeddings generated by the pretrained CMM, thereby bypassing the fine-tuning step.

We compared these two ablated models with the original CellNavi (incorporating both CMM and DGP) using two types of data splits: (1) **Out-of-domain split:** The Norman single perturbation split from Figure 2, where a cluster is held out from training to simulate a generalization scenario, and (2) **In-domain split:** Randomly selecting the same number of test cells as in (1), mimicking a setting without generalization. Removing either the pretraining step (CMM) or the fine-tuning module (DGP) led to performance drops in both scenarios, indicating that the full pipeline is necessary for optimal performance (**Figure**

R3, also as Figure S2.6 in the revised manuscript). Notably, the performance gap was more pronounced in the out-of-domain split. In particular, when testing on biologically distinct cases (out-of-domain split), the absence of pretraining caused a more significant decline in performance compared to the random split, suggesting that pretraining with CMM improves generalization across biological contexts.

Figure R3 (Figure S2.6). Performance evaluation of CellNavi with and without CMM pretraining and DGP fine-tuning modules. The figure compares the performance of CellNavi under three configurations: (1) CellNavi with both CMM pretraining and DGP fine-tuning, (2) using the DGP without gene embeddings from CMM or any pretrained model, instead directly using the gene expression vector, and (3) replacing the DGP with a simpler multinomial logistic regression model on top of the CMM. Performance is assessed under two evaluation settings: out-of-domain (the previously used Norman single perturbation split that holds one cluster out from training, red) and in-domain (a random split on the Norman dataset with the same train/test size, gray). Metrics include Top-1 accuracy, Top-5 accuracy, AUROC, AUPRC, F1 score, and MCC (Matthews Correlation Coefficient). Dashed lines indicate performance change between evaluation settings.

In summary, pretraining and fine-tuning serve distinct and essential purposes in CellNavi framework. Pretraining provides a strong foundation by capturing generalized cell state representations, while fine-tuning enables task-specific adaptation to capture gene-cell

relationships. Together, they synergistically contribute to the final model performance, with pretraining ensuring generalization across diverse biological contexts and fine-tuning enabling precise task-specific predictions. Thank you again for raising this important point!

7. Biological data are inherently noisy, which can significantly impact model performance. Evaluating how CellNavi performs under varying degrees of noise in single-cell gene expression data would provide valuable insights into the model's reliability and its capacity to handle noisy, real-world datasets effectively.

Reply: Thank you for raising this important point regarding the impact of noise on biological data on model performance. In the context of our task, we have identified and categorized two types of noise: 1) Noise due to technical variability in sequencing, such as dropout events and variations in sequencing depth. 2) Noise related to CRISPR efficiency, including fluctuating perturbation efficiency, off-target effects, and stochasticity from intrinsic cellular variability.

For the first type of noise, we demonstrated that our approach effectively overcomes sequencing noise caused by varying depths (**Figure R1**). To summarize the key results relevant to this point: by leveraging our down-sampling recovery pre-training, we align input data across different sequencing depths, achieve robust imputation of missing gene expressions, and reduce technical variability in single-cell datasets. This recovery capability is a crucial feature for addressing the challenges of real-world single-cell datasets, where sequencing depth can vary significantly.

For the second type of noise, we did observe challenges related to CRISPR perturbation variability. For instance, perturbed cells from distinct perturbations often appeared mixed rather than forming well-defined clusters. This observation is supported by the normalized Local Inverse Simpson's Index (LISI) scores, which quantify the degree of mixing or segregation in the dataset. A normalized LISI score of 1 indicates perfect mixing, whereas a score of 0 reflects complete segregation. For the Norman dataset, we found a mean

normalized LISI score of 0.67 and a median normalized LISI score of 0.75, indicating a high degree of mixing overall (**Figure R4**).

Figure R4. Distribution of normalized LISI across different gene perturbations in the dataset

We found that model performance is correlated with the heterogeneity of the training dataset (Figure 2e): perturbed genes with higher LISI scores (more mixed perturbation results) exhibited worse performance. This issue is inherently difficult to address, as it may reflect the stochastic nature of cellular behaviors, which are challenging to capture and delineate without sufficient data. We anticipate that including training data from multiple batches, sources, and sgRNAs could further reduce biases from specific experiments. By doing so, the model may effectively smooth out noise or biases introduced by individual perturbations, improving its robustness to CRISPR-specific variability.

As suggested, we have incorporated this discussion into the manuscript. Specifically, we now highlight that CellNavi is capable of handling a certain level of noise in real-world biological data, particularly noise due to technical variability in sequencing. Additionally, we discuss that noise related to CRISPR efficiency, such as fluctuating perturbation effects and off-target activities, poses a more complex challenge and may require further methodological advancements and experimental efforts to address comprehensively. In the Discussion section, we included the following:

Inherent noise in biological data presents a significant challenge for modeling. In this study, we addressed this challenge by employing a down-sampling recovery pretraining strategy with a mixed down-sampling rate to mitigate technical variability, such as dropout events and differences in sequencing depth. This strategy aligns input data of varying depths, enabling the model to process real-world single-cell datasets more effectively. This recovery capability is especially critical for ensuring robust performance across datasets with inconsistent sequencing quality. Another source of noise arises from variability in CRISPR perturbation efficiency, including fluctuating perturbation success rates and off-target effects caused by intrinsic cellular stochasticity. Although CRISPR screens provide a rich and diverse dataset for training CellNavi, this noise may lead to incorrect labels during training and biased predictions. To mitigate this, future efforts could incorporate data from multiple batches, sources, and sgRNAs to reduce biases associated with specific experimental conditions. Additionally, integrating diverse perturbation data, such as chemical treatments with well-characterized targets or direct introduction of mRNAs or proteins, could complement CRISPR-based data and further enhance model robustness.

8. As depicted in Figure 2a, the manuscript describes using a specific cell cluster as the test set while others are used for training. While this approach assesses generalization across clusters, relying on a single train-test split may introduce bias depending on the selected cluster. Incorporating cross-validation would offer a more comprehensive and reliable evaluation of the model's performance.

Reply: Thank you for your suggestion.

For the Schmidt dataset, the data is naturally split into resting and re-stimulated T cells. To evaluate the model's robustness across different conditions, we employed a reverse train-test strategy: training the model on resting T cells and testing on re-stimulated T cells, and vice versa (training on re-stimulated and testing on resting). This approach ensures evaluation across biologically distinct states and highlights the model's ability to generalize.

For the Norman dataset, we adopted a cluster-based holdout strategy. Specifically, we held out one cluster for testing while using single perturbation data from the remaining clusters to fine-tune the model. To achieve cross-validation, we rotated the test cluster across different clusters, ensuring thorough and unbiased evaluation. This setup allows us to assess the model's performance on unseen cell clusters while leveraging perturbation data from other clusters for fine-tuning, yielding a total of 5 train-test split datasets.

The results for these cross-validation experiments are currently included in the revised manuscript (Table S4-S5) for a more comprehensive and rigorous evaluation of our methods. For your convenience, we also attached the results below (**Table R2-R3**).

	Top-1 acc (↑)	Top-5 acc (↑)	AUROC (↑)	AUPRC (↑)	F1 score (↑)	MCC (↑)
Train: Re-stimulated T cells. Test: Resting T cells						
CellNavi	0.6206	0.7326	0.8745	0.4991	0.6154	0.6155
SCENIC+	0.0930	0.3482	0.6274	0.0689	0.0493	0.0667
GEARS	0.0520	0.2110	0.5170	0.0307	0.0276	0.0317
DEGlogFC	0.0693	0.2104	0.6295	0.0790	0.0483	0.0578
Train: Resting T cells. Test: Re-stimulated T cells						
CellNavi	0.4020	0.5313	0.8021	0.2809	0.4223	0.4013
SCENIC+	0.0672	0.2366	0.5936	0.0509	0.0599	0.0492
GEARS	0.1021	0.2747	0.5388	0.0362	0.0554	0.0739
DEGlogFC	0.0722	0.2021	0.6237	0.0345	0.0343	0.0598

Table R2 (Table S4). The table presents cross-validation results comparing CellNavi with alternative on the Schmidt dataset. Performance metrics include Top-1 accuracy, Top-5 accuracy, area under the receiver operating characteristic curve (AUROC), area under the precision-recall curve (AUPRC), F1 score, and Matthews Correlation Coefficient (MCC). Two experimental conditions are shown: (1) training on re-stimulated T cells and testing on resting T cells, and (2) training on resting T cells and testing on re-stimulated T cells.

	Top-1 accuracy	Top-5 accuracy	AUROC	AUPRC	F1 score	MCC
Split 1 (used in Figure 2)						
CellNavi	0.4900	0.6330	0.8884	0.4540	0.4219	0.4816
SCENIC	0.1140	0.2970	0.5713	0.0228	0.1005	0.0963
GEARS	0.0562	0.1420	0.5535	0.0267	0.0437	0.0439
DGE	0.0319	0.0931	0.5474	0.0137	0.0103	0.0098
Split 2						
CellNavi	0.8632	0.9259	0.9723	0.8496	0.1384	0.7074
SCENIC	0.0019	0.0061	0.4304	0.0108	0.0005	0.0063
GEARS	0.0226	0.0600	0.5416	0.0203	0.0331	0.0147
DGE	0.7691	0.7769	0.9164	0.6053	0.8002	0.5616
Split 3						
CellNavi	0.4902	0.6997	0.8960	0.4438	0.3821	0.4834
SCENIC	0.0529	0.1252	0.5456	0.0249	0.0409	0.0509
GEARS	0.1647	0.3425	0.5335	0.0183	0.1262	0.1500
DGE	0.0412	0.1309	0.5969	0.0205	0.0129	0.0341
Split 4						
CellNavi	0.4609	0.6405	0.8676	0.3668	0.4224	0.4547
SCENIC	0.0221	0.0831	0.5265	0.0182	0.0175	0.0336
GEARS	0.1429	0.3007	0.5582	0.0221	0.1332	0.1259
DGE	0.0238	0.0699	0.5427	0.0204	0.0223	0.0167
Split 5						
CellNavi	0.5944	0.7857	0.9118	0.5336	0.1671	0.5581
SCENIC	0.0734	0.1335	0.7148	0.0341	0.0478	0.0786
GEARS	0.2019	0.3270	0.5429	0.0202	0.1562	0.1605
DGE	0.2564	0.4126	0.8245	0.1345	0.1632	0.1981

Table R3 (Table S5). The table presents cross-validation results comparing CellNavi with alternative on the Norman dataset, using single perturbation experiments. Performance metrics include Top-1 accuracy, Top-5 accuracy, area under the receiver operating characteristic curve (AUROC), area under the precision-

recall curve (AUPRC), F1 score, and Matthews Correlation Coefficient (MCC). Split 1 is the same split as used in Figure 2 in the manuscript.

9. The manuscript evaluates model performance using AUROC and F1 score (e.g., Figure 2b-c), which offer valuable insights into overall performance. However, including additional metrics, such as precision-recall curves or Matthews Correlation Coefficient (MCC), could provide a more nuanced and comprehensive evaluation of the model's effectiveness.

Reply: We have thoroughly updated the evaluation metrics used in our study to provide a more comprehensive assessment of model performance. We now report the following performance metrics: macro F1 score, area under the receiver operating characteristic curve (AUROC, macro), area under the precision-recall curve (AUPRC, macro), Matthew's correlation coefficient (MCC), top-1 accuracy, and top-5 accuracy.

10. The manuscript provides insights into the performance of CellNavi but does not appear to include details about computational efficiency. What are the runtimes for training and inference, along with the hardware specifications (e.g., GPU, memory) used?

Reply: We apologize for not providing sufficient details earlier and thank the reviewer for highlighting this point. The pretraining phase of CellNavi was conducted on a large dataset of approximately 20 million cells, requiring 32 A100 around 16 days to complete. The fine-tuning phase, performed on task-specific datasets ranging from 28,453 to 111,445 cells, required 14 to 40 hours depending on the dataset size, with a batch size of 128. For inference, the model achieved a processing speed of approximately 0.38 seconds per cell. All experiments were conducted using an NVIDIA A100 GPU (80 GB PCIe) and an Intel Xeon Gold 6342 CPU (24 cores, 2.80 GHz). We have included these details in the revised manuscript to improve its transparency and reproducibility.

Minor:

1. In line 64, the word "that" following the colon in "that Cell Manifold Model" should be revised to "the" for grammatical accuracy.

2. In line 298, there is a minor spelling error in the figure caption: "cekers" should be corrected to "whiskers," the appropriate term for describing elements of a box plot.

Reply: Thank you for your thorough review. We have corrected the typos accordingly in the revised manuscript highlighted the changes in red.

Reviewer #3 (Remarks to the Author):

This manuscript presents CellNavi, a novel deep learning framework for predicting driver genes in cell state transitions. The authors address a crucial and challenging problem, employing a sophisticated approach that integrates manifold learning with prior knowledge from gene interaction graphs. Incorporating these graphs into the Cell Manifold Model is a notable innovation, likely improving the model's ability to capture complex gene-state relationships. The self-supervised pretraining task for the CMM is also a strength, leveraging the abundance of unlabeled single-cell data. The demonstration of CellNavi's performance across diverse biological contexts, from T cell differentiation to drug responses, showcases its potential utility. The framework's ability to generalize across cell types and predict driver genes even in scenarios not explicitly present in the training data, such as drug treatments, is particularly promising.

Reply: We sincerely appreciate your recognition of CellNavi and your constructive feedback, which has greatly helped us refine and strengthen our manuscript. Below, we outline the key revisions we made to address your comments and improve the presentation of our work.

First, in response to your inquiries regarding the down-sampling rate, we demonstrated the empirical rationale behind our choice of the current rates and the pretraining strategy, which employs a mix of 1–20 down-sampling, by analyzing over 1,000 single-cell RNA-seq datasets. Further, comparative experiments on three models showed that the mixed strategy achieved the best overall performance, by balancing sequencing depth variation and aligning datasets with differing sparsity levels.

Second, we expanded our benchmarking experiments to compare CellNavi with additional state-of-the-art gene regulatory network (GRN) inference methods, including GENIE3, GRNBoost2, and RENGE. These methods, while highly effective in their respective domains, performed poorly when applied to our perturbation-driven driver gene prediction task. We believe this limitation stems from their reliance on TF-target gene relationships, which are insufficient for capturing the broader regulatory and signaling dynamics required for this application.

Additionally, we have extended the discussion to address the inherent noise in biological data, particularly in CRISPR datasets, as well as limitations related to the reliance on such data for training and testing. We discussed challenges such as off-target effects and incomplete perturbation coverage, which can introduce variability and bias, and proposed strategies to mitigate these issues. These include integrating diverse data modalities and improving the model's ability to generalize beyond the training dataset.

Finally, we acknowledged the importance of temporal dynamics and stochasticity in cell state transitions, which are not fully captured by CellNavi's current reliance on static snapshots. To address this, we outlined future directions for incorporating time-series single-cell data and dynamic modeling approaches to better capture intermediate states and temporal dependencies in cellular processes. This discussion has been added to emphasize the potential for further development of CellNavi in this area.

These revisions and new analyses have been incorporated into the manuscript to provide a more comprehensive evaluation and presentation of CellNavi.

1.Sensitivity to Downsampling: The CMM's pretraining incorporates a downsampling step. Could the authors provide a more detailed analysis of the model's sensitivity to the choice of downsampling rate? How do varying rates influence the learned manifold and the downstream driver gene predictions?

Reply: The motivation for down-sampling-reconstruction strategy is to address the issue of batch effects caused by varying sequencing depths in single-cell RNA-seq datasets. Differences in sequencing depth often result in the loss of low-expression genes, exacerbating technical variability and causing cells with similar biological identities to cluster separately [1-2]. To mitigate this, we simulate depth variation by selectively masking low-expression genes at controlled down-sampling rates and train the model to reconstruct the original gene expression profiles. We expect this strategy enables the model to learn embeddings that are robust to data sparsity and align cell representations across datasets with differing depths. Below, we elaborate on the rationale behind the selection of the current down-sampling rate and present results exploring the sensitivity of the model to different down-sampling rates.

Choice of down-sampling rate

The down-sampling rate (ds) determines the fraction of UMI counts retained, with higher rates leading to more aggressive sparsity. Given the computational resources and time required for pre-training, we empirically selected the down-sampling rate based on an analysis of real-world single-cell RNA-seq datasets.

First, we analyzed the distribution of median UMI counts per cell across 1,487 human single-cell RNA-seq datasets collected from the Human Cell Atlas. As shown in the histogram and empirical cumulative distribution (ECDF) in **Figure R1**, **the majority (80%) of datasets exhibit a median UMI count between 1,000 and 10,000, while half the datasets have a median UMI below 3,184.** A down-sampling rate of 20 simulates a scenario where datasets with median UMI counts near 10,000 are reduced to approximately 500 UMIs per cell, a level at which meaningful biological interpretation becomes increasingly challenging [3-4]. This analysis suggests a maximum down-

sampling rate of 20, a realistic sparsity level that approaches the threshold where data may become biologically uninformative in certain scenarios.

Figure R1. Distribution of median UMI counts per cell across 1,487 human single-cell RNA-seq datasets. Left: the histogram of median total UMI counts, highlighting the variability in sequencing depth across datasets. Right: the empirical cumulative distribution (ECDF) of median UMI counts. 91% of datasets have a median UMI count below 10,000, while 11% fall below 1,000.

To further examine the impact of down-sampling rates on cell-type resolution, we used a human pancreas scRNA-seq dataset as a representative example [5]. We tested various rates ($ds=5, 10, 20, 30, 50, 100$) and found that dominant cell populations remained distinguishable even at high down-sampling rates, likely due to high-expression marker genes. However, **moderate cell types, such as quiescent and activated stellate cells and endothelial cells, began to mix at rates above $ds=20$** , as shown in **Figure R2**. Based on these observations, we constrained the maximum down-sampling rate to 20 in our experiments to preserve the resolution of rare populations while simulating realistic sparsity levels.

Mixed Down sampling Strategy (ds1-20)

For the final pretraining procedure, **we employed a mixed down-sampling strategy (ds1-20) with rates ranging from $ds=1$ (original dataset) to $ds=20$** . This approach simulates a continuous variation in sequencing depth across datasets, enhancing the model's ability to generalize while mitigating out-of-distribution challenges between the input (down-sampled data) and the output (original expression profiles). This strategy aligns with principles from robust learning frameworks [6].

Figure R2. UMI counts distribution and cell-type clustering at varying down-sampling rates. (a) UMI count distribution. Center line, median; box limits, upper and lower quartiles; whiskers, 1.5x interquartile range. (b) UMAP visualization of the dataset after different down-sampling rates. Cells are colored by original cell-type annotations. Black circles highlight cell types that start to mix at higher down-sampling rates.

Experiments

To investigate how different down-sampling rates affect cell manifold recovery, we explored three settings: ds1-20 (the mixed down-sampling strategy), ds2 (a down-sampling rate of 2 only), and ds20 (a down-sampling rate of 20 only). Due to resource and time constraints, we selected these three strategies, with ds2 and ds20 representing two extreme cases of down-sampling. Evaluations were performed on the previously described human pancreas dataset, which was held out from training. For comparison, we simulated datasets with down-sampling rates of 2, 5, 10, and 20.

The mixed strategy (ds1-20) demonstrated superior overall recovery performance compared to single-rate models (ds2 or ds20) (**Figure R3**). As expected, the ds2 model excelled at aligning 2-fold down-sampled data to the original expression profiles but performed poorly on 20-fold data. Conversely, the ds20 model showed strong performance on 20-fold data but exhibited relatively suboptimal performance when tested on 2-fold data.

These results indicate that the mixed down-sampling strategy (ds1-20) achieves the best balance and generalization across datasets with varying levels of sparsity. While this initial exploration offers meaningful insights and addresses the posed question, it remains

a preliminary effort. A more systematic and comprehensive investigation will provide deeper understanding and further validate these findings in the future.

Figure R3. Evaluation of different down-sampling strategies on cell manifold recovery. (a) Quantitative performance metrics, including NMI (normalized mutual information), ARI (adjusted Rand index), ASW (average silhouette width), cLISI (cell-level local inverse Simpson's index), and iLISI (integration-level local inverse Simpson's index), were used to compare recovery across down-sampling strategies (ds1-20, ds2, ds20) under simulated down-sampling rates (ds=2, 5, 10, 20, x-axis). Arrows indicate the desired direction of improvement for each metric. (b) UMAP visualization of cell-type and batch organization for the three models (ds1-20, ds20, ds2) on simulated datasets with 2-fold and 20-fold down-sampling rates. The ds1-20 model demonstrated superior alignment and separation of cell types across both rates, while ds2 struggled with 20-fold data, and ds20 is less satisfying on aligning 2-fold data. Cell types are color-coded, and batches are distinguished as original profiles (blue) and down-sampled profiles (gray).

We have now included these details in the revised manuscript (**Supplementary Note 2**). Thank you for your feedback and for helping us clarify these points.

References

1. Pratapa, A., Jaliha, A. P., Law, J. N., Bharadwaj, A., & Murali, T. M. (2020). Benchmarking algorithms for gene regulatory network inference from single-cell transcriptomic data. *Nature Methods*, 17, 147–154.
2. Korsunsky, I., et al. (2019). Fast, sensitive and accurate integration of single-cell data with Harmony. *Nature Methods*, 16, 1289–1296.
3. Zheng, G. X., et al. (2017). Massively parallel digital transcriptional profiling of single cells. *Nature communications*, 8(1), 14049.
4. Svensson, V., da Veiga Beltrame, E., & Pachter, L. (2020). A curated database reveals trends in single-cell transcriptomics. *Database*, 2020, baaa073.
5. Baron, M., et al. (2016). A single-cell transcriptomic map of the human and mouse pancreas reveals inter-and intra-cell population structure. *Cell systems*, 3(4), 346-360.
6. Goodfellow, I. J., et al. (2014). Generative Adversarial Networks. Preprint at <https://arxiv.org/abs/1406.2661>.

2. Comparison with other GRN inference methods: While the comparison to SCENIC+ and GEARS is relevant, benchmarking CellNavi against a wider range of established gene regulatory network inference methods would be valuable. Including methods like GENIE3, GRNBoost2, and single-cell specific methods like RENGINE (<https://www.nature.com/articles/s42003-023-05594-4?fromPaywallRec=false>) would provide a more comprehensive performance assessment relative to the state-of-the-art.

Reply:

Following your suggestions, we conducted additional benchmarking experiments using three representative gene regulatory network (GRN) inference methods: GENIE3 [1], GRNBoost2 [2], and the recently developed single-cell-specific method RENGINE [3]. These comparisons were designed to provide a more comprehensive evaluation of CellNavi's performance relative to state-of-the-art GRN inference methods.

We began by constructing GRNs using cells from Norman dataset and Schmidt dataset, respectively, maintaining the same experimental settings as outlined in the original manuscript (see Methods section). Due to computational memory constraints, we limited the analysis for GENIE3 and RENGINE to the top 5,000 highly variable genes, which may have impacted the completeness of the inferred networks. For RENGINE, which was developed to infer GRNs using time-series scRNA-seq data, we adapted the method to work with static single-cell RNA-seq data, as the Norman and Schmidt datasets lack time-series information. While this adaptation allowed for benchmarking, it eliminated key features of RENGINE.

Next, we conducted two types of experiments leveraging these GRNs. First, we directly tested the GRN inference methods, following the same downstream analysis pipeline used for SCENIC+ as described in the Methods section for a fair comparison. Second, we replaced NicheNet with the inferred GRNs in the CellNavi framework to assess their utility.

Our results indicate that CellNavi with NicheNet outperforms all alternative approaches (**Table R1**). Especially, inference with GRNs performed worse compared to leveraging GRNs as the prior graph in CellNavi. This is likely due to the limitations of GRN-based approaches, which rely heavily on predefined transcription factor-target gene structures and may fail to capture key “driver genes” or indirect regulatory relationships critical for cell state dynamics, such as those involved in drug responses. In contrast, NicheNet and CellNavi integrate both signaling pathways and transcriptional regulatory information, enabling them to account for more comprehensive and subtle gene-gene interactions.

	Top-1 acc	Top-5 acc	AUROC	AUPRC	F1 score	MCC
Schmidt Dataset						
CellNavi(with NicheNet)	0.6206	0.7326	0.8745	0.4991	0.6154	0.6155
GENIE3	0.1389	0.4941	0.5785	0.1000	0.0084	0.0842
GRNBoost2	0.1049	0.2718	0.6106	0.0788	0.0787	0.0843
RENGE	0.0358	0.2738	0.5133	0.0468	0.0034	0.0126
CellNavi(with GENIE3)	0.4916	0.6149	0.8216	0.3642	0.5105	0.4917
CellNavi(with GRNBoost2)	0.4091	0.5201	0.8050	0.2850	0.3982	0.4111
CellNavi(with RENGINE)	0.5600	0.6835	0.8443	0.4487	0.5800	0.5551
Norman Dataset						
CellNavi(with NicheNet)	0.4900	0.6330	0.8884	0.4540	0.4219	0.4816
GENIE3	0.0865	0.1911	0.5909	0.0318	0.0693	0.0739
GRNBoost2	0.0313	0.0902	0.5301	0.0183	0.0239	0.0363
RENGE	0.0516	0.3107	0.5836	0.0636	0.0160	0.0062
CellNavi(with GENIE3)	0.1390	0.3819	0.7935	0.0941	0.1107	0.1271
CellNavi(with GRNBoost2)	0.2533	0.4495	0.8210	0.2012	0.2295	0.2399
CellNavi(with RENGINE)	0.4184	0.5974	0.8532	0.3518	0.3733	0.4074

Table R1. The influence of GRNs on CellNavi performance. Top-1 accuracy, Top-5 accuracy, F1-score, AUROC, AUPRC, and Matthews Correlation Coefficient (MCC) for various GRN inference methods and GRN graphs utilized in combination with the CellNavi framework, evaluated on the Norman and Schmidt datasets.

We have included these new results and the corresponding discussion in the revised manuscript (Table S1 and Table S6) for a more comprehensive and rigorous evaluation of CellNavi.

References

1. Huynh-Thu, V. A., Irrthum, A., Wehenkel, L., & Geurts, P. Inferring regulatory networks from expression data using tree-based methods. *PLoS one*, 5(9), e12776 (2010).
2. Moerman, T., Aibar Santos, S., Bravo González-Blas, C. *et al.* GRNBoost2 and Arboreto: efficient and scalable inference of gene regulatory networks. *Bioinformatics*, 35(12), 2159-2161 (2019).
3. Ishikawa, M., Sugino, S., Masuda, Y. *et al.* RENGE infers gene regulatory networks using time-series single-cell RNA-seq data with CRISPR perturbations. *Commun Biol* **6**, 1290 (2023)

3. Limitations of the CRISPR approach: While the CRISPR screens provide a more controlled setting for evaluation than purely observational data, certain limitations should be explicitly addressed. Specifically, the incompleteness of typical CRISPR screens (targeting only a subset of genes) and the potential for off-target effects can influence the apparent accuracy of the method. The authors should discuss these limitations and how they might affect their interpretation of CellNavi's performance.

Reply: Thank you for highlighting this critical point. We used CRISPR screen data for CellNavi training, as it provides a large and diverse dataset with rich biological information. However, this approach has inherent limitations, as you mentioned. First, it is impractical to exhaustively target all genes with CRISPR experiments, and since CellNavi is not designed to generalize to unseen genes, this fact limits its broader application. Second, off-target effects in CRISPR screens can introduce noise, potentially leading to incorrect labels during model training and biased prediction. These issues can be mitigated in future work through three complementary strategies: 1) Enabling the ability to propose unseen genes by better capturing gene networks and their effects on cellular states through improved model designs and the integration of other data modalities. 2) Mitigating the effect of off-targets on model training by using data from multiple batches, sources, and sgRNAs to reduce biases from specific experiments. 3) Incorporating diverse perturbation data, such as chemical perturbations with well-characterized targets or direct introduction of mRNAs or proteins, to complement CRISPR-based data.

As suggested, we have now included discussions in the manuscript addressing these limitations and potential mitigation strategies:

“Another source of noise arises from variability in CRISPR perturbation efficiency, including fluctuating perturbation success rates and off-target effects caused by intrinsic cellular stochasticity. Although CRISPR screens provide a rich and diverse dataset for training CellNavi, this noise may lead to incorrect labels during training and biased predictions. To mitigate this...”

“CellNavi is not designed to generalize to novel genes, which limits its broader applicability. Expanding the model’s capacity to predict unseen genes would require capturing gene networks and representations that enable extrapolation beyond the training dataset. While a comprehensive single-cell CRISPR experiment encompassing a broader range of target genes and cell types is highly desirable...”

“Finally, CellNavi currently lacks the ability to accurately model long-range transitions due to its reliance on CRISPR perturbations and static snapshots of transcriptomic data...”

We believe these points will clarify how the limitations of CRISPR data influence CellNavi and outline potential approaches to strengthen its robustness in the future. Thank you for your valuable feedback!

4. Addressing Temporal Dynamics and Stochasticity: CellNavi primarily focuses on predicting driver genes based on static snapshots of cell states. However, cell state transitions are dynamic processes often influenced by intrinsic noise and stochasticity. How might these temporal aspects and inherent randomness affect the model's performance? The authors should discuss these limitations and potential future directions for incorporating temporal information and stochasticity into the CellNavi framework.

Reply: Regarding **stochasticity**, we did observe a high level of stochasticity in CRISPR perturbation response. For instance, perturbed cells from distinct perturbations often appeared mixed rather than forming well-defined clusters. This observation is supported by the normalized Local Inverse Simpson’s Index (LISI) scores, which quantify the degree of mixing or segregation in the dataset. A normalized LISI score of 1 indicates perfect mixing, whereas a score of 0 reflects complete segregation. For the Norman dataset, we found a mean normalized LISI score of 0.67 and a median normalized LISI score of 0.75, indicating a high degree of mixing overall (**Figure R4**).

Figure R4. Distribution of normalized LISI across different gene perturbations in the dataset

We did find that model performance is correlated with the heterogeneity of the training dataset (**Figure 2e**): datasets with more mixed perturbation results exhibited worse performance. Similarly to the response to the previous comment, we believe that including training data from multiple batches, sources, and sgRNAs may effectively smooth out the stochasticity introduced by individual perturbations, improving its robustness to CRISPR-specific variability. Also, extending the model to include stochastic components, such as Gaussian processes or latent variable models, could allow us to better capture variability and uncertainty in cellular transitions, thereby improving predictions.

Cell state transitions are inherently **dynamic** processes that unfold over time. Static snapshots may miss critical intermediate states and fail to capture the temporal ordering of gene regulatory events. This limitation could reduce the accuracy of driver gene predictions for transitions that involve sequential or time-dependent regulatory mechanisms, especially for long-range transitions in cell identities or fates that likely involve multiple waves of signaling changes. We have demonstrated that the Cell Manifold Model (CMM) in the CellNavi framework can reconstruct developmental trajectories (**Figure R5**, Figure S1.3 in the manuscript), which is a promising starting point. However, it is very challenging to establish the molecular patterns that drive the transitions of cell fates.

Future directions to address this stochasticity issue include integrating time-series single-cell transcriptomic data into the training of CellNavi framework. By incorporating temporal data, the model could learn not only the endpoints of transitions but also the intermediate cell states, enabling a more comprehensive understanding of the dynamic progression of cellular transitions at molecular level. For example, RNA velocity data could be incorporated to infer pseudotemporal trajectories. These trajectories could then be used to construct dynamic manifolds that reflect the progression of cell states over time, enabling the model to account for temporal dependencies.

Figure R5 (Figure S1.3) Trajectory reconstruction with the Cell Manifold Model. (a) UMAPs colored by cell type annotations (left) and diffusion pseudotime (DPT) (right), inferred from the original data (upper panel) and CellNavi-predicted data (lower panel). Cell type labels were assigned based on known markers, and pseudotime was computed using Scanpy’s DPT implementation. **(b)** Spearman correlation between pseudotime inferred from the original data (x-axis) and from the CellNavi-predicted data (y-axis) with a quantification in **c**. **(c)** Summary of quantitative evaluation using Spearman correlation, along with clustering evaluation metrics including Adjusted Rand Index (ARI), Normalized Mutual Information (NMI), Average Silhouette Score (ASS), and Fowlkes-Mallows Score (FMS). Scores computed from random embeddings are included for comparison. We used a mouse pancreatic endocrinogenesis dataset from Bastidas-Ponce et al.⁸⁴. Therefore, we enhanced our model with mouse scRNA-seq for this task.

We have added the following content to the discussion section to address this point:

“Finally, CellNavi currently lacks the ability to accurately model long-range transitions due to its reliance on CRISPR perturbations and static snapshots of transcriptomic data. Many biological processes, such as differentiation and disease progression, involve gradual changes or transient intermediate states that are not captured in steady-state data. Incorporating time-resolved single-cell data to construct dynamic manifolds that reflect temporal progression could enable CellNavi to capture meaningful intermediate states and provide deeper insights into the dynamics of cellular transitions.”

5. Evaluation with Synthetic Data: While the CRISPR data provides some level of ground truth, a more rigorous evaluation using synthetic data generated from a known mathematical model would greatly benefit the manuscript. Simulations based on stochastic dynamical systems models, where the true driver genes are predefined, would allow for a precise quantification of CellNavi’s accuracy. This

approach would also enable systematic exploration of the model's performance under different conditions (such as varying the level of noise) and parameter settings, offering valuable insights into its strengths and limitations. This would address the limitations of real CRISPR data, such as incomplete perturbation coverage and off-target effects.

This manuscript presents a promising new tool with clear potential for advancing the study of cell state transitions. Addressing the points above would significantly strengthen the work and increase confidence in the capabilities of the CellNavi framework.

Reply: Thank you for this valuable suggestion. We agree that using synthetic datasets generated from well-defined mathematical models can provide a controlled evaluation framework, allowing for precise quantification of CellNavi's accuracy and systematic exploration of its performance under varying conditions.

While our current evaluation relies on real-world CRISPR perturbation data, which offers biologically relevant insights, it does have limitations, such as incomplete perturbation coverage, potential off-target effects, and inherent biological stochasticity. Synthetic data would allow us to address these limitations by providing a fully controlled ground truth for benchmarking. Simulations based on stochastic dynamical systems models or other computational frameworks could generate data with predefined TRUE driver genes, enabling us to rigorously evaluate the model's accuracy and robustness.

That said, incorporating synthetic evaluations was not included in this round of revisions due to time constraints and the significant computational and methodological effort required to design and validate realistic synthetic datasets that accurately reflect the complexity of real single-cell perturbation data. Developing biologically meaningful synthetic datasets that include both noise and realistic cell state transitions would require careful parameterization and validation to ensure they are reflective of real-world scenarios. Given the focus of this study on demonstrating the applicability of CellNavi to real, experimentally derived perturbation datasets, we prioritized improvements to our benchmarking using CRISPR data and additional ablation studies as requested by the reviewers.

We fully agree, however, that integrating synthetic data evaluation in future work will add significant value. While this effort is beyond the scope of the current revision, we are excited to pursue this direction in future studies. We appreciate this insightful suggestion, which will undoubtedly enhance the robustness and generalizability of CellNavi in subsequent work. Thank you again for pointing out this important opportunity for further development.

We sincerely thank all reviewers for the time and effort you dedicated to helping us improve our manuscript, as well as for recognizing the value of our work. In finalizing the revision, we have made the following changes.

Reviewer #1:

Remarks to the Author:

I am mostly happy with the current version of the manuscript and with the efforts of the authors to address my requests. I still have some requests for clarifications and suggestions to better balance some of the author's statements.

1. In line 532, authors state. "However, it is worth noting that our approach to constructing cell-type-specific gene graphs involves a simplifying assumption: edges are removed for genes with zero expression. This approach is based on the premise that non-expressed genes are unlikely to participate in meaningful regulatory or signaling interactions. Filtering out these genes reduces noise and allows the model to focus on biologically active relationships." Authors need to consider that zero entries might also be related to expression dropouts / lower sequencing depth of a particular cell. This aspect should be also discussed in the text.

We mentioned the limitation on removing genes with zero expression (Lines 399-401): "Yet, we recognize that zero expression values may also stem from technical artifacts such as dropout or low sequencing depth, rather than true biological absence."

2. While results regarding interpretability are interesting, i.e. TFs have more attention, the authors do not really address the interpretability request in full. Authors should stress the explainability limitations of their work in the discussions, by for example adopting a more critical view in their discussion as presented in the reply letter (see below).

Current text in the discussion line 562 "This explicit focus on regulatory elements provides CellNavi with a distinct advantage, enabling it to better model complex biological processes, and highlights the value of graph-based learning in improving model interpretability and biological relevance".

Reply to my request: "We acknowledge that explainability remains a critical challenge in deep learning models, including CellNavi, while our current work focuses on benchmarking performance and demonstrating the utility of pathway-based graphs. This is an area for future exploration, where we aim to develop

methods to better visualize and interpret the graph attention mechanisms and how they relate to pathway connectivity and driver gene predictions.”

We stressed the explainability limitations of our work in the discussions (Lines 390-394): “However, we caution that attention mechanisms do not equate to mechanistic interpretability. The explainability remains a critical challenge for deep learning models, including CellNavi. Future work should develop tools to visualize and interpret how graph structures and attention dynamics shape predictions of driver genes.”

Reviewer #3:

Remarks to the Author:

I believe the authors have addressed my main concerns comprehensively and adequately. I think the comparison with synthetic datasets produced by mathematical models may be a little tangential to their study and perhaps they can follow this up in future work.

Remarks on code availability:

One small suggestion here is in the "About" section, i'd suggest the authors add some keywords to help the scientific community find their code.

As suggested by Reviewer 3, we have added appropriate keywords to the GitHub repository to improve accessibility and discoverability.